

# Exact large-scale correlations
# in integrable systems out of equilibrium

**Benjamin Doyon**

Department of Mathematics, King's College London, Strand, London WC2R 2LS, U.K.

## Abstract

Using the theory of generalized hydrodynamics (GHD), we derive exact Euler-scale dynamical two-point correlation functions of conserved densities and currents in inhomogeneous, non-stationary states of many-body integrable systems with weak space-time variations. This extends previous works to inhomogeneous and non-stationary situations. Using GHD projection operators, we further derive formulae for Euler-scale two-point functions of arbitrary local fields, purely from the data of their homogeneous one-point functions. These are new also in homogeneous generalized Gibbs ensembles. The technique is based on combining a fluctuation-dissipation principle along with the exact solution by characteristics of GHD, and gives a recursive procedure able to generate $n$-point correlation functions. Owing to the universality of GHD, the results are expected to apply to quantum and classical integrable field theory such as the sinh-Gordon model and the Lieb-Liniger model, spin chains such as the XXZ and Hubbard models, and solvable classical gases such as the hard rod gas and soliton gases. In particular, we find Leclair-Mussardo-type infinite form-factor series in integrable quantum field theory, and exact Euler-scale two-point functions of exponential fields in the sinh-Gordon model and of powers of the density field in the Lieb-Liniger model. We also analyse correlations in the partitioning protocol, extract large-time asymptotics, and, in free models, derive all Euler-scale $n$-point functions.



# 1 Introduction

The nonequilibrium dynamics of integrable many-body systems has received a large amount of attention recently, especially in view of experimental realizations in cold atomic gases [1–3]. It is known that in situations with slow, large-scale variations in space and time, the principles of hydrodynamics hold [4–6]. The recently developed generalized hydrodynamics (GHD) [7,8] applies these principles to the presence of infinitely-many conservation laws afforded by integrability. The original works [7,8] strongly suggest that GHD, in the quasi-particle formulation, has wide applicability within quantum systems, including quantum chains and quantum field theory (QFT), requiring only a restricted set of dynamical and kinematical data. These data arise from the thermodynamic Bethe ansatz (TBA) [9–11]. In the quantum context (and omitting the simple cases of free particles), GHD has been explicitly worked out in

general integrable QFT with diagonal scattering (such as the sinh-Gordon and Lieb-Liniger models) [7, 12, 13], in the XXZ quantum chain [8, 14], and in the Hubbard model, which displays "nested Bethe ansatz" [15], and is expected to apply to all known integrable QFT and Bethe-ansatz integrable models. The structure of GHD, however, transcends its origin from the Bethe ansatz, and GHD can be shown to apply to an even larger variety of models, including classical integrable field theory [16] and classical gases such as the hard rod model [13, 17–19] and soliton gases [20–25]. The theory has been quite successful, see for instance [26–33]. GHD, as developed until now, is valid at the Euler scale, but viscous and other corrections have been considered, see [19, 29, 34–38]. In the present paper, we restrict to the Euler scale.

An important problem is that of evaluating dynamical correlations. For definiteness, let an initial state $\langle \cdots \rangle_{\text{ini}}$ be of the form

$$\langle O \rangle_{\text{ini}} = \frac{\text{Tr}\left(e^{-\int_{\mathbb{R}} dx \sum_i \beta_i(x) \mathfrak{q}_i(x)} O\right)}{\text{Tr}\left(e^{-\int_{\mathbb{R}} dx \sum_i \beta_i(x) \mathfrak{q}_i(x)}\right)} \tag{1.1}$$

(for any observable $O$). Here $\mathfrak{q}_i(x)$, $i \in \mathbb{N}$ form a basis of local and quasi-local densities [39] of homogeneous, extensive conserved quantities $Q_i = \int dx\, \mathfrak{q}_i(x)$ in involution, and $\beta_i(x)$ are parameters, which can be interpreted as generalized local temperatures or local chemical potentials of the integrable hierarchy. We use a continuous space notation $x$, and the trace notation Tr. This is for convenience, and the problem is posed in its most general setting, for classical (where the trace means a summation over classical configurations) or quantum models, on a one-dimensional infinite space that can be continuous or discrete.

The state (1.1) is an inhomogeneous version of a generalized Gibbs ensemble [40–42]. Let the evolution of a local observable $\mathcal{O}(x)$ be generated by some homogeneous dynamics that is integrable, for instance with Hamiltonian $H$,

$$\mathcal{O}(x, t) = e^{iHt} \mathcal{O}(x) e^{-iHt}. \tag{1.2}$$

Then one would like to evaluate the set of dynamical connected correlation functions[1]

$$\langle \mathcal{O}_1(x_1, t_1) \cdots \mathcal{O}_n(x_n, t_n) \rangle_{\text{ini}}^{\text{c}} \tag{1.3}$$

for local observables $\mathcal{O}_k(x_k, t_k)$.

The problem can be divided into two classes. First, if all $\beta_i(x) = \beta_i$ are independent of position, then the initial state is (homogeneous) GGE. The evaluation of exact correlations functions within GGEs is a difficult problem, and in classical models has been little studied. One-point functions of conserved densities $\mathfrak{q}_i$ are directly accessible from the TBA, and those of conserved currents $\mathfrak{j}_i$ (with $\partial_t \mathfrak{q}_i + \partial_x \mathfrak{j}_i = 0$) were obtained as part of the development of GHD [7,8]. There is also the Leclair-Mussardo formula for GGE one-point functions of generic local fields in integrable QFT [43,44], based on form factors [45–47], and formulae for certain one-point functions in the Lieb-Liniger model [48, 49] and the sinh-Gordon model [50–52]. For GGE two-point functions, various types of spectral expansions exist [53–57], including new results of the Leclair-Mussardo type [58], as well as exact results in free-particle models based on integrable partial differential equations [59–63] (mostly Gibbs states are considered, but the techniques are extendable to GGEs). In integrable quantum spin chains, expressions for correlation functions in Gibbs states [64, 65] and in GGEs [66–68] have been obtained, but large space-time asymptotics are still to be fully addressed. Stronger results exist in the hydrodynamic regime: Lieb-Liniger particle density correlations from form factors [69], and more generally a set of efficient formulae for two-point functions of all local densities and

---

[1] Here and below, the superscript $^{\text{c}}$ means "connected".

currents in any integrable model [13, 15], obtained by combining GHD with hydrodynamic projection methods [70, 71].

Second, more interestingly, let $\beta_i(x)$ depend on the position $x$ in a weak enough fashion. This may arise, in good approximation, as initial ground states or finite-temperature states of quantum or classical systems in weakly varying potentials, or after a (short) local-relaxation time in the partitioning protocol of non-equilibrium steady states [72]. In this case, much less is known. GHD gives direct access to local GGEs describing the mesoscopic fluid cells, hence to all space-time dependent one-point functions of observables whose GGE averages can already be evaluated. However, for two- and higher-point functions, results only exist in the context of free field theory. Importantly, this includes Luttinger Liquids, and gives access, using the local density approximation and related hydrodynamic ideas, to the low-temperature limit of inhomogeneous integrable models, such as the Lieb-Liniger model in inhomogeneous potentials or the Heisenberg chain with in homogeneous interaction coupling, see for instance [73–79]. Inhomogeneous two- and higher-point functions have never been studied in more general interacting integrable systems.

In this paper, we provide both a first step in the study of correlation functions in inhomogeneous situations, and further develop the theory of correlation functions in (homogeneous) GGEs. We evaluate Euler-scaled dynamical connected correlation functions in inhomogeneous, non-stationary states, in the generality of GHD (without inhomogeneous force fields). The results apply not only to conserved densities and currents, but also to more general local fields where correlation function formulae are obtained purely from the knowledge of GGE one-point functions. The latter are new also when specialized to GGEs.

More precisely, the objects we study are as follows. Consider the scaled initial state

$$\langle O \rangle_{\text{ini},\lambda} = \frac{\text{Tr}\left(e^{-\int_{\mathbb{R}} dx \sum_i \beta_i(\lambda^{-1}x) q_i(x)} O\right)}{\text{Tr}\left(e^{-\int_{\mathbb{R}} dx \sum_i \beta_i(\lambda^{-1}x) q_i(x)}\right)}, \tag{1.4}$$

for smooth functions $\beta_i(x)$. The scaling in $\lambda$ guarantees that the Lagrange parameters of the initial state depend weakly on the position. Let us denote by $\mathcal{N}_\lambda(x,t)$ a mesoscopic fluid cell: this can be taken as a space-time region whose extent scales as $\lambda^\nu$ for some $\nu_0 < \nu < 1$, around the scaled point $\lambda x$, say $\mathcal{N}_\lambda(x,t) = \{(y,s) : \sqrt{(y-\lambda x)^2 + (s-\lambda t)^2} < \lambda^\nu\}$. The value of $\nu_0$ depends on the subleading corrections to Euler hydrodynamics; if they are diffusive, then we would expect $\nu_0 = 1/2$. Let us also denote by $|\mathcal{N}_\lambda| = \int_{\mathcal{N}_\lambda(x,t)} dy\,ds$ its volume. The "Eulerian scaling limit" for correlation functions is defined as the limit

$$\langle \mathcal{O}_1(x_1,t_1)\cdots\mathcal{O}_N(x_N,t_N)\rangle^{\text{Eul}}_{[n_0]} \tag{1.5}$$

$$= \lim_{\lambda\to\infty} \lambda^{N-1} \int_{\mathcal{N}_\lambda(x_1,t_1)} \frac{dy_1 ds_1}{|\mathcal{N}_\lambda|} \cdots \int_{\mathcal{N}_\lambda(x_N,t_N)} \frac{dy_N ds_N}{|\mathcal{N}_\lambda|} \langle \mathcal{O}_1(y_1,s_1)\cdots\mathcal{O}_N(y_N,s_N)\rangle^{\text{c}}_{\text{ini},\lambda}$$

for fixed $x_k$'s and $t_k$'s. Here the superscript c means that we take connected correlation functions, and $n_0$ represents the initial GHD occupation function, which characterizes the initial state at the Euler scale (the GGEs of the initial fluid cells). Fluid-cell averaging, $\int_{\mathcal{N}_\lambda(x_k,t_k)} \frac{dy_k ds_k}{|\mathcal{N}_\lambda|}\cdots$, is necessary in order to avoid non-Eulerian oscillations, and averaging can be performed in various ways (see [16] for a discussion of fluid-cell averaging and oscillations). For one-point functions, numerical observations and exact calculations in free models suggest that fluid-cell averaging is not necessary, and one has $\langle \mathcal{O}(x,t)\rangle^{\text{Eul}}_{[n_0]} = \lim_{\lambda\to\infty}\langle \mathcal{O}(\lambda x,\lambda t)\rangle_{\text{ini},\lambda}$.

We propose a generating function method in order to evaluate (1.5), based on combining an Euler-scale fluctuation-dissipation principle with the "nonlinear method of characteristics" introduced in [33]. We expect the generating function method to be valid whenever equal-time correlations vanish fast enough in space. It is expected to work in all quantum and classical

systems that have been shown to be accessible by GHD, and applies to conserved densities $q_i$ and currents $j_i$. In the cases of two-point functions, we show that the method provides explicit nonlinear integral equations which can in principle be solved numerically, and from which various special cases can be extracted. The results on two-point functions agree with the GHD projection operators derived in [13], and in homogeneous states, reproduce the formulae found in [13, 15].

Further, using hydrodynamic projections, we find formulae for Euler-scale two-point functions of arbitrary local fields, expressed purely in terms of their homogeneous GGE averages. To every local field we associate a hydrodynamic spectral function obtained from its GGE averages, which enters the two-point function formula. Combining with the Leclair-Mussardo expansion in integrable QFT (or its counterpart in classical field theory [80]), we obtain form factor series for Euler-scale dynamical two-point functions for any local field. Using the Bertini-Piroli-Calabrese simplification of the Negro-Smirnov formula [50–52] we also obtain explicit results for two-point functions of exponential fields in the sinh-Gordon model, and using Pozsgay's formula [81], of powers of the density operator in the Lieb-Liniger model. These constitute the first such exact results not only in inhomogeneous, non-stationary states, but also in homogeneous GGEs.

Finally, we obtain all Euler-scale $n$-point functions in free models, study two-point functions of conserved densities in the partitioning protocol, obtaining a number of new results for its solution by characteristics, and study the large-time asymptotics of two-point functions from arbitrary inhomogeneous initial conditions.

The paper is organized as follows. In Section 2, we review the basics of GHD, with emphasis both on the general framework accounting for all known examples, and on aspects which are important for the study of dynamical correlation functions. In Section 3, we present the main results about correlation functions, including the generating function method, the two-point functions of conserved densities and currents, the hydrodynamic projection interpretation, and the extension to generic local observables. In Section 4, we give examples of the main formulae, in the sinh-Gordon and Lieb-Liniger models, and in free-particle models. In Section 5 we provide some discussion and analysis of the results, including a study of two-point functions in the partitioning protocol, and a precise analysis of the large-time asymptotics of two-point functions for a large class of initial states. Finally, we conclude in Section 6. The details of the computations are reported in appendices.

## 2   Review of GHD

Making full sense of the state (1.1) is not a trivial matter. If the infinite sum in the exponential truncates, then – at least in classical and quantum chains – there is a well developed mathematical theory [82–84]. In the case of homogeneous states, $\beta_i(x) = \beta_i$, there are many studies that discuss the precise terms that must be included within the infinite series $\sum_i \beta_i Q_i$ in various situations, and its convergence in terms of averages of local observables, see the review [41]. A mathematically rigorous framework has been given [85] showing that the infinite sum can be interpreted as a decomposition in a basis of the Hilbert space of pseudolocal charges; in particular, the infinite series itself is a pseudolocal conserved charge. Later, it was understood how GGEs connect to the quasi-particle description of TBA [86], and an in-depth analysis of finite-series truncations and convergence of local averages was given [67].

Here we concentrate on the quasi-particle description of GHD as originally developed [7,8]. The generality of GHD has been claimed in various works and the same basic ingredients extracted, see e.g. [13, 15, 33]. In order to establish the notation, which follows [7], we recall these ingredients. We further provide general notions concerning correlation functions, and

we make a full account of situations with non-symmetric differential scattering phase (or TBA kernel), making apparent the invariance under quasi-particle reparametrization. It has been noted that this general framework needs small adjustments in order to deal with spin-carrying quantities in the massive regime of the XXZ Heisenberg chain, see [14]; we will not consider this subtlety here.

## 2.1 GGEs in the quasi-particle formulation

We denote by $\mathcal{S}$ the spectral space of the model. The space $\mathcal{S}$ can be seen roughly as the space of all quasi-particle characteristics admitted in the thermodynamics of the model; it is the space of excitations emerging after diagonalizing the scattering in the thermodynamic limit. In general, $\mathcal{S}$ is decomposed into disconnected components: each component represents a quasi-particle type, and is a continuum representing the allowed momenta for this quasi-particle type. The spectral space, therefore, has the form of a disjoint union $\mathcal{S} = \cup_{a \in A} I_a$, where $\mathcal{A}$ is the set of quasi-particle types, and $I_a$ are continuous subsets of copies $\mathbb{R}$ representing the continua of momenta for each particle type. We will parametrise each continuum by a variable $\theta \in I_a$, which we will refer to as the rapidity[2]. One may write a spectral parameter as $\boldsymbol{\theta} = (\theta, a)$ with $\theta \in I_a$ and $a \in \mathcal{A}$. We will use the notation

$$\int_{\mathcal{S}} \mathrm{d}\boldsymbol{\theta} = \sum_{a \in \mathcal{A}} \int_{I_a} \mathrm{d}\theta. \tag{2.1}$$

Besides the set $\mathcal{S}$, the model is specified by giving the momentum and energy functions $p(\boldsymbol{\theta})$ and $E(\boldsymbol{\theta})$ respectively, and the differential scattering phase (or more generally the TBA kernel occurring after diagonalization of the scattering) $\varphi(\boldsymbol{\theta}, \boldsymbol{\alpha})$, a function of two spectral parameters. The momentum function $p(\boldsymbol{\theta})$ defines physical space and specifies the parametrisation used. Without loss of generality, by faithfulness of the parametrisation we assume that it satisfies

$$p'(\boldsymbol{\theta}) > 0,$$

where $p'(\boldsymbol{\theta}) = \mathrm{d}p(\boldsymbol{\theta})/\mathrm{d}\theta$ (here and below the prime $'$ denotes a rapidity derivative). The energy function, on the other hand, defines physical time, and equals the "one-particle eigenvalue" (or the equivalent in classical systems) of the conserved charge that generates time translations (the Hamiltonian), see for instance [12]. The differential scattering phase, of course, specifies the interaction.

All equations below are independent of the momentum parametrisation $\theta$ used. This invariance involves certain transformation properties of the objects introduced, which are either scalar fields or vector fields. Under rapidity reparametrisations, the differential scattering phase $\varphi(\boldsymbol{\theta}, \boldsymbol{\alpha})$ transforms as a vector field (i.e. as $\partial/\partial\theta$) in $\theta$, and a scalar field in $\alpha$, that is

$$\varphi(\boldsymbol{\theta}, \boldsymbol{\alpha})\mathrm{d}\theta \qquad \text{is invariant under reparametrisation } \theta \mapsto f(\theta),\ \alpha \mapsto f(\alpha). \tag{2.2}$$

For instance, the differential scattering phase is defined, in diagonal scattering models, as $\varphi(\boldsymbol{\theta}, \boldsymbol{\alpha}) = -\mathrm{i}\,\mathrm{d}S(\boldsymbol{\theta}, \boldsymbol{\alpha})/\mathrm{d}\theta$ where $S(\boldsymbol{\theta}, \boldsymbol{\alpha})$ is the two-body scattering matrix. The momentum and energy functions are scalar fields, while their derivatives, $p'(\boldsymbol{\theta})$ and $E'(\boldsymbol{\theta})$, are vector fields.

Also given is a set of one-particle eigenvalues, scalar fields $h_i(\boldsymbol{\theta})$ for $i \in \mathbb{N}$ associated to the conserved charges $Q_i$. The space spanned by these functions is assumed to be in bijection with a dense subspace of the Hilbert space of pseudolocal conserved charges (this Hilbert space is

---

[2]Note however that this is not necessarily any of the rapidities that may appear in natural ways in Bethe ansatz solutions, it is simply some faithful parametrisation of the continua of momenta.

induced by the inner product defined via integrated correlations, see [85] and the Remark in Subsection 2.2).

The important dynamical quantities, which specify the GGE in the TBA quasi-particle formulation, are an occupation function $n(\boldsymbol{\theta})$, a pseudo-energy $\epsilon(\boldsymbol{\theta})$, a particle density $\rho_{\mathrm{p}}(\boldsymbol{\theta})$ and a state density $\rho_{\mathrm{s}}(\boldsymbol{\theta})$, which are all related to each other [10, 11]. The former two are scalar fields, the latter vector fields. Associated to these is the dressing map $h \mapsto h^{\mathrm{dr}}_{[n]}$, which is a functional of $n(\boldsymbol{\theta})$ and a linear operator on (an appropriate space of) spectral functions $h$. We define it, in general, differently for its action on vector fields and on scalar fields: it is defined by solving the linear integral equations

$$h^{\mathrm{dr}}_{[n]}(\boldsymbol{\theta}) = h(\boldsymbol{\theta}) + \int_{\mathcal{S}} \frac{\mathrm{d}\boldsymbol{\alpha}}{2\pi} \varphi(\boldsymbol{\theta}, \boldsymbol{\alpha}) n(\boldsymbol{\alpha}) h^{\mathrm{dr}}_{[n]}(\boldsymbol{\alpha}) \qquad \text{(if } h(\boldsymbol{\theta}) \text{ is a vector field)}$$

$$h^{\mathrm{dr}}_{[n]}(\boldsymbol{\theta}) = h(\boldsymbol{\theta}) + \int_{\mathcal{S}} \frac{\mathrm{d}\boldsymbol{\alpha}}{2\pi} \varphi(\boldsymbol{\alpha}, \boldsymbol{\theta}) n(\boldsymbol{\alpha}) h^{\mathrm{dr}}_{[n]}(\boldsymbol{\alpha}) \qquad \text{(if } h(\boldsymbol{\theta}) \text{ is a scalar field).}$$

(2.3)

The dressing operation preserves the transformation property under rapidity reparametrization. For lightness of notation in this paper, omitting the index $[n]$ means dressing with respect to the occupation function denoted $n(\boldsymbol{\theta})$, that is $h^{\mathrm{dr}} = h^{\mathrm{dr}}_{[n]}$.

It will be convenient to employ an integral-operator notation. We introduce the scattering operator $T$, with kernel $T(\boldsymbol{\theta}, \boldsymbol{\alpha}) = \varphi(\boldsymbol{\theta}, \boldsymbol{\alpha})/(2\pi)$, acting on spectral functions $h$ as

$$(Th)(\boldsymbol{\theta}) = \int_{\mathcal{S}} \frac{\mathrm{d}\boldsymbol{\alpha}}{2\pi} \varphi(\boldsymbol{\theta}, \boldsymbol{\alpha}) h(\boldsymbol{\alpha}), \qquad (2.4)$$

as well as its transposed $T^{\mathrm{T}}$ with kernel $T^{\mathrm{T}}(\boldsymbol{\theta}, \boldsymbol{\alpha}) = \varphi(\boldsymbol{\alpha}, \boldsymbol{\theta})/(2\pi)$. By a slight abuse of notation, we also sometimes use $n$ for the diagonal operator acting as multiplication by $n(\boldsymbol{\theta})$. In these terms,

$$h^{\mathrm{dr}} = (1 - Tn)^{-1} h \qquad \text{(if } h(\boldsymbol{\theta}) \text{ is a vector field),}$$

$$h^{\mathrm{dr}} = (1 - T^{\mathrm{T}} n)^{-1} h \qquad \text{(if } h(\boldsymbol{\theta}) \text{ is a scalar field).}$$

(2.5)

Both the occupation function and the particle density may be taken as characterising a thermodynamic state (a GGE). Other state quantities are related to them:

$$2\pi \rho_{\mathrm{s}} = (p')^{\mathrm{dr}}, \qquad \rho_{\mathrm{p}} = n \rho_{\mathrm{s}}, \qquad (2.6)$$

where $p'(\boldsymbol{\theta})$ is a vector field[3]. The relation between the pseudo-energy $\epsilon(\boldsymbol{\theta})$ and the occupation function $n(\boldsymbol{\theta})$ depends on the type of excitation mode considered: it is different for quantum fermionic or bosonic degrees of freedom (as discussed in [10]), for classical particle-like modes such as solitons (as discussed in [87,88]) or hard rods (as discussed in [13,29]), and for classical radiative modes occurring for instance in classical field theory (the GHD of classical field theory is developed in [16] based on [87,88]). We have $n(\theta, a) = \partial \mathsf{F}_a(\epsilon)/\partial \epsilon |_{\epsilon = \epsilon(\theta, a)}$ where the free energy function $\mathsf{F}_a$ is given by

$$\mathsf{F}_a(\epsilon) = \begin{cases} -\log(1 + e^{-\epsilon}) \\ \log(1 - e^{-\epsilon}) \\ -e^{-\epsilon} \\ \log \epsilon \end{cases} \Rightarrow n(\boldsymbol{\theta}) = \begin{cases} 1/(e^{\epsilon(\boldsymbol{\theta})} + 1) & (a \text{ is a fermion}) \\ 1/(e^{\epsilon(\boldsymbol{\theta})} - 1) & (a \text{ is a boson}) \\ e^{-\epsilon(\boldsymbol{\theta})} & (a \text{ is a classical particle}) \\ 1/\epsilon(\boldsymbol{\theta}) & (a \text{ is a radiative mode}) \end{cases}$$

(2.7)

---

[3]Note that if the state density is given by some other means – for instance via its fundamental geometric interpretation [33] – then the first equation in (2.6) can be seen as a definition of the momentum function for the chosen spectral parametrisation.

(recall that the mode type is encoded within the particle type $a$ of the spectral parameter $\boldsymbol{\theta} = (\theta, a)$). Note that the free energy function determines the "generalized free energy" of the GGE, given by $\int d\boldsymbol{\theta}\, p'(\boldsymbol{\theta}) F_a(\epsilon(\boldsymbol{\theta}))$.

Averages in GGEs will be denoted by $\langle O \rangle_{[n]}$, functionals of the state variable $n(\boldsymbol{\theta})$. Averages of conserved densities and currents are found to be [7, 8]

$$\langle \mathsf{q}_i \rangle_{[n]} = \int_{\mathcal{S}} d\boldsymbol{\theta}\, \rho_{\mathrm{p}}(\boldsymbol{\theta}) h_i(\boldsymbol{\theta}) = \int_{\mathcal{S}} \frac{dp(\boldsymbol{\theta})}{2\pi} n(\boldsymbol{\theta}) h_i^{\mathrm{dr}}(\boldsymbol{\theta}) \tag{2.8}$$

$$\langle \mathsf{j}_i \rangle_{[n]} = \int_{\mathcal{S}} d\boldsymbol{\theta}\, v^{\mathrm{eff}}(\boldsymbol{\theta}) \rho_{\mathrm{p}}(\boldsymbol{\theta}) h_i(\boldsymbol{\theta}) = \int_{\mathcal{S}} \frac{dE(\boldsymbol{\theta})}{2\pi} n(\boldsymbol{\theta}) h_i^{\mathrm{dr}}(\boldsymbol{\theta}). \tag{2.9}$$

The effective velocity is [7, 8, 90]

$$v^{\mathrm{eff}}(\boldsymbol{\theta}) = \frac{(E')^{\mathrm{dr}}(\boldsymbol{\theta})}{(p')^{\mathrm{dr}}(\boldsymbol{\theta})}. \tag{2.10}$$

Here we recall that $h_i(\boldsymbol{\theta})$ are scalar fields and $E'(\boldsymbol{\theta})$ and $p'(\boldsymbol{\theta})$ are vector fields.

The Lagrange parameters $\{\beta_i\}$ of a GGE fix the state, formally, via the trace expression

$$\langle O \rangle_{[n]} = \frac{\mathrm{Tr}\left(e^{-\sum_i \beta_i Q_i} O\right)}{\mathrm{Tr}\left(e^{-\sum_i \beta_i Q_i}\right)}. \tag{2.11}$$

One can recover the occupation function $n(\boldsymbol{\theta})$ from the set $\{\beta_i : i \in \mathbb{N}\}$, and vice versa, via a set of nonlinear integral equations: one defines the GGE driving term $w(\boldsymbol{\theta}) = \sum_i \beta_i h_i(\boldsymbol{\theta})$, which involves the one-particle eigenvalues $h_i(\boldsymbol{\theta})$ associated to the conserved charges $Q_i$, and one solves $\epsilon(\boldsymbol{\theta}) = w(\boldsymbol{\theta}) + \int (d\boldsymbol{\gamma}/2\pi)\, \varphi(\boldsymbol{\gamma}, \boldsymbol{\theta}) F_b(\epsilon(\boldsymbol{\gamma}))$ (where $\boldsymbol{\gamma} = (\gamma, b)$). For our purposes, we mainly need the derivative of $n(\boldsymbol{\theta})$ with respect to $\beta_i$. Again the result depends on the type of excitation mode considered, and may be written as

$$\frac{\partial}{\partial \beta_i} n(\boldsymbol{\theta}) = -h_i^{\mathrm{dr}}(\boldsymbol{\theta}) n(\boldsymbol{\theta}) f(\boldsymbol{\theta}), \tag{2.12}$$

where the statistical factor of the mode is $f(\theta, a) = -\partial_\epsilon^2 F_a(\epsilon)/\partial_\epsilon F_a(\epsilon)|_{\epsilon=\epsilon(\theta,a)}$, giving

$$f(\boldsymbol{\theta}) = \begin{cases} 1 - n(\boldsymbol{\theta}) & \text{(fermions)} \\ 1 + n(\boldsymbol{\theta}) & \text{(bosons)} \\ 1 & \text{(classical particles)} \\ n(\boldsymbol{\theta}) & \text{(radiative modes)}. \end{cases} \tag{2.13}$$

The quantities $\epsilon(\boldsymbol{\theta})$, $\rho_s(\boldsymbol{\theta})$, $\rho_{\mathrm{p}}(\boldsymbol{\theta})$ $v^{\mathrm{eff}}(\boldsymbol{\theta})$ and $f(\boldsymbol{\theta})$ are all functionals of an occupation function; below we use these symbols for the quantities associated to the occupation function denoted $n(\boldsymbol{\theta})$.

## 2.2 Generalized fluids in space-time

Recall that the Eulerian scaling limit (1.5) is defined as a large-scale limit, with fluid cell averaging, of connected correlation functions. This exactly extracts the information about the correlations that is present in the physics of Euler fluids. In order to describe it, we need to construct fluid configurations where at every Euler-scale space-time position $(x, t) \in \mathbb{R} \times \mathbb{R}$ lies a GGE. We thus need a family of state functions, which we denote equivalently as

$$n_{x,t}(\boldsymbol{\theta}) \equiv n_t(x; \boldsymbol{\theta}),$$

with $\boldsymbol{\theta} \in \mathcal{S}$ the spectral parameter. The function $n_{x,t}(\boldsymbol{\theta})$, as a function of $\boldsymbol{\theta}$ for $x, t$ fixed, is the occupation function describing the GGE in the fluid cell at $(x, t)$. Below we will use the index $[n_{x,t}]$ for averages in the GGE at the space-time point $(x, t)$, which are functionals of this function of $\boldsymbol{\theta}$. On the other hand, $n_t(x; \boldsymbol{\theta})$ seen as a function of the doublet $(x, \boldsymbol{\theta})$ for $t$ fixed, is the fluid state on the time slice $t$. We will use the index $[n_t]$ for functionals that depend on this function of $(x; \boldsymbol{\theta})$. For instance, the Eulerian scaling limit (1.5) is a functional of the initial state $n_0$, while by definition, evolving for a (Euler-scale) time $t$ gives

$$\left\langle \prod_k \mathcal{O}_k(x_k, t_k + t) \right\rangle_{[n_0]}^{\text{Eul}} = \left\langle \prod_k \mathcal{O}_k(x_k, t_k) \right\rangle_{[n_t]}^{\text{Eul}}. \tag{2.14}$$

Recall that the dressing operation (2.3) as well as the various TBA quantities are all functionals of an occupation function. For readability, we will use the notation $h^{\text{dr}}(x, t; \boldsymbol{\theta}) = h^{\text{dr}}_{[n_{x,t}]}(\boldsymbol{\theta})$, as well as $\rho_s(x, t; \boldsymbol{\theta})$, $\rho_p(x, t; \boldsymbol{\theta})$, $v^{\text{eff}}(x, t; \boldsymbol{\theta})$ and $f(x, t; \boldsymbol{\theta})$ for the quantities associated to the occupation function $n_{x,t}(\boldsymbol{\alpha})$ (as a function of $\boldsymbol{\alpha}$ for $(x, t)$ fixed).

The fluid state on any time slice $t$ takes a factorized form, where on each fluid cell lies a GGE. That is, at large scales correlation functions factorize as

$$\lim_{\lambda \to \infty} \left\langle \prod_{k=1}^N \mathcal{O}_k(\lambda x_k, \lambda t) \right\rangle_{\text{ini}, \lambda} = \prod_{k=1}^N \langle \mathcal{O}_k(x_k) \rangle_{[n_{x_k,t}]} \qquad (x_j \neq x_k \text{ for } j \neq k). \tag{2.15}$$

Here $\langle \mathcal{O}_k(x_k) \rangle_{[n_{x_k,t}]}$ is the average of the local (Schrödinger-picture) operator $\mathcal{O}_k(x_k)$, in the GGE $n_{x_k,t}$ which lies at Euler-scale space-time position $(x_k, t)$. In order for the results below to be valid, we in fact require that equal-time, space-separated connected correlation functions vanish fast enough[4],

$$\lim_{\lambda \to \infty} \lambda^{N-1} \left\langle \prod_{k=1}^N \mathcal{O}_k(\lambda x_k, \lambda t) \right\rangle_{\text{ini}, \lambda}^c = 0 \qquad (x_j \neq x_k \text{ for } j \neq k). \tag{2.16}$$

Thus the Eulerian scaling limit (1.5) is zero whenever all times are the same and no two positions coincide. Relation (2.16) is expected to hold for all conserved densities and currents, and for most other local observables, in a large family of states; for instance, it holds in any homogeneous, nonzero-temperature Kubo-Martin-Schwinger state of local quantum chains.

The initial fluid state $n_0(x; \boldsymbol{\theta})$ is the Euler scale version of the state (1.1). According to (2.15), it factorizes into local GGEs. The local GGE at space-time position $(x, 0)$ is determined by the parameters $\{\beta_i(x) : i \in \mathbb{N}\}$ which appear in (1.1) as per (2.11):

$$\langle O \rangle_{[n_{x,0}]} = \frac{\text{Tr}\left(e^{-\sum_i \beta_i(x) Q_i} O\right)}{\text{Tr}\left(e^{-\sum_i \beta_i(x) Q_i}\right)}. \tag{2.17}$$

In particular, according to (2.12), it satisfies the functional derivative equation

$$\frac{\delta}{\delta \beta_i(y)} n_0(x; \boldsymbol{\theta}) = -\delta(x - y) h^{\text{dr}}_i(x, 0; \boldsymbol{\theta}) n_0(x; \boldsymbol{\theta}) f(x, 0; \boldsymbol{\theta}). \tag{2.18}$$

In accordance with the factorized form (2.15) and especially (2.16), equal-time scaled connected correlation functions have support only at coinciding points. In fact, taking the Eulerian scaling limit, they can be written in the form

$$\left\langle \prod_{k=1}^N \mathcal{O}_k(x_k) \right\rangle_{[n_t]}^{\text{Eul}} = C_{[n_{x_1,t}]}^{\mathcal{O}_1, \ldots, \mathcal{O}_N} \prod_{j=2}^N \delta(x_1 - x_j). \tag{2.19}$$

---

[4]In non-equilibrium steady states emerging form the partitioning protocol, this requirement is broken by certain fields, see e.g. [89, Eq.33].

By integration, one can identify the pre-factor as the full integral of the connected correlation function in the homogeneous local state at $x_1$,

$$C^{\mathcal{O}_1,\dots,\mathcal{O}_N}_{[n_{x_1,t}]} = \int_{\mathbb{R}^{N-1}} dx_2 \cdots dx_N \Big\langle \prod_{k=1}^{N} \mathcal{O}_k(x_k) \Big\rangle^c_{[n_{x_1,t}]}. \tag{2.20}$$

Note that the scaling factor $\lambda^{N-1}$ exactly cancels that coming from the re-scaling of the integration variables, and that thanks to the space integration, it is not necessary anymore to average over fluid cells.

**Remark.** For every GGE $n(\boldsymbol{\theta})$, there is a Hilbert space formed by the completion, under the natural topology, of the space of local observables with the $(x, t)$-dependent inner "hydrodynamic inner product"

$$\langle \mathcal{O}_1 | \mathcal{O}_2 \rangle_{[n]} = C^{\mathcal{O}_1^\dagger, \mathcal{O}_2}_{[n]} = \int_{\mathbb{R}} dx \, \langle \mathcal{O}_1^\dagger(x) \mathcal{O}_2(0) \rangle^c_{[n]}. \tag{2.21}$$

There is a sub-Hilbert space formed by the set of conserved densities $\mathcal{O}_1, \mathcal{O}_2 \in \{\mathfrak{q}_i : i \in \mathbb{N}\}$ within this Hilbert space. The space spanned by $h_i(\boldsymbol{\theta})$, $i \in \mathbb{N}$ is required to be dense within this sub-Hilbert space, and this, for all $n = n_t(x)$. This, generically, imposes the inclusion of quasi-local conserved densities. See the review [39] for quasi-local densities, and [85] for a rigorous description of these Hilbert spaces and the way they are involved in generalized thermalization.

## 2.3 Time evolution

Consider a generalized fluid in space-time that is obtained, after the Eulerian scaling limit, by evolving an initial state (1.1) using a homogeneous dynamics as in (1.2), (1.3). This satisfies an Eulerian fluid equation [7, 8]. This is the main equation of GHD, which can be written as the convective evolution equation

$$\partial_t n_t(x; \boldsymbol{\theta}) + v^{\text{eff}}(x, t; \boldsymbol{\theta}) \partial_x n_t(x; \boldsymbol{\theta}) = 0. \tag{2.22}$$

Its "solution by characteristics" was discovered in [33]. Given the initial condition $n_0(x; \boldsymbol{\theta})$, one introduces the characteristics, a function $u(x, t; \boldsymbol{\theta})$, which one evaluates along with the evolved state $n_t(x; \boldsymbol{\theta})$ by solving the following set of equations:

$$n_t(x; \boldsymbol{\theta}) = n_0(u(x, t; \boldsymbol{\theta}); \boldsymbol{\theta})$$
$$\int_{x_0}^{x} dy \, \rho_s(y, t; \boldsymbol{\theta}) = \int_{x_0}^{u(x,t;\boldsymbol{\theta})} dy \, \rho_s(y, 0; \boldsymbol{\theta}) + v^{\text{eff}}(x_0, 0; \boldsymbol{\theta}) \rho_s(x_0, 0; \boldsymbol{\theta}) \, t. \tag{2.23}$$

In these equations, $x_0$ is an "asymptotically stationary point": it must be chosen far enough on the left in such a way that $n_s(x; \boldsymbol{\theta}) = n_0(x; \boldsymbol{\theta})$ for all $x < x_0$ and $s \in [0, t]$ (typically, one should think of it as $x_0 = -\infty$). This provides the evolution from the initial condition $n_0$ for a time $t$.

It is worth noting that the function $u(x, t; \boldsymbol{\theta})$ has the simple interpretation as the position, at time 0, from where a quasi-particle trajectory of spectral parameter $\boldsymbol{\theta}$ would reach the position $x$ at time $t$. Indeed, it solves

$$\partial_t u(x, t; \boldsymbol{\theta}) + v^{\text{eff}}(x, t; \boldsymbol{\theta}) \partial_x u(x, t; \boldsymbol{\theta}) = 0, \qquad u(x, 0; \boldsymbol{\theta}) = x. \tag{2.24}$$

Thus, defining the trajectory $x(t)$, starting at $x(0) = y$, via

$$u(x(t), t; \boldsymbol{\theta}) = y, \tag{2.25}$$

we find

$$\frac{\mathrm{d}x(t)}{\mathrm{d}t}\partial_x u(x,t;\boldsymbol{\theta})|_{x=x(t)} + \partial_t u(x,t;\boldsymbol{\theta})|_{x=x(t)} = 0 \ \Rightarrow \ \frac{\mathrm{d}x(t)}{\mathrm{d}t} = v^{\mathrm{eff}}(x(t),t;\boldsymbol{\theta}). \quad (2.26)$$

Below we assume the following: (i) the state density $\rho_s(\boldsymbol{\theta})$ is positive for all $\boldsymbol{\theta}$, and (ii) the equations (2.23) have a unique solution. Thanks to these assumptions, differentiating with respect to $x$ the second equation in (2.23), we have the inequality

$$\partial_x u(x,t;\boldsymbol{\theta}) > 0, \quad (2.27)$$

which imply that the function $u(x,t;\boldsymbol{\theta})$ is invertible with respect to the position.

**Remark.** Note that if we assume that the effective velocity $v^{\mathrm{eff}}(\boldsymbol{\theta})$ is a monotonically increasing function of the rapidity $\theta$, then (Appendix A)

$$u'(x,t;\boldsymbol{\theta}) < 0 \quad (2.28)$$

so that that the function $u(x,t;\boldsymbol{\theta})$ is invertible with respect to the rapidity. The latter condition is satisfied for instance in Galilean of relativistic field theories. This condition slightly simplifies some of the considerations, and in particular it guarantees that $(v^{\mathrm{eff}})'(\boldsymbol{\theta}) \neq 0$. In fact, if the latter inequality is not satisfied, then some of the asymptotic results below do not apply. Yet, we will not make use of the monotonicity assumption, but we will implicitly assume that $(v^{\mathrm{eff}})'(\boldsymbol{\theta}) \neq 0$ when it appears in denominators, keeping the discussion of how a vanishing derivative of the effective velocity may change some results for the conclusion.

## 3 Correlation functions

Despite the factorization properties (2.15) and (2.16) on equal-time slices, scaled connected correlation functions (1.5) are nontrivial when fields do not all lie on the same time slice. That is, a connected dynamical $N$-point function vanishes, at the Euler scale, as $\lambda^{1-N}$ with a generically nonzero coefficient, which is extracted (after fluid-cell average) by taking the Eulerian scaling limit (1.5).

In this section, we develop a recursive procedure that generates all scaled dynamical correlation functions (1.5). The procedure is based on linear responses and an extension of the fluctuation-dissipation theorem to Euler scale correlations. We identify the *propagator,* propagating from time 0 to time $t$, as (simply related to) the linear response of $n_t$ to variations of the initial condition $n_0$. We explain how, in the cases of two-point functions involving conserved densities $\mathsf{q}_i(x,t)$ and currents $\mathsf{j}_i(x,t)$, one can obtain from this procedure explicit integral equations. We also explain how one can extend these formulae, combining hydrodynamic projection principles with the Leclair-Mussardo formula, to two-point functions involving other local fields. We finally state the general results for scaled $n$-point functions in free models.

It is worth noting that in general, correlation functions depend on much more than the information present in the Euler hydrodynamics. For instance, although the knowledge of the GGE equations of states is sufficient to determine the full thermodynamics and Euler hydrodynamics, it cannot be sufficient to determine correlation functions of the type $\langle \mathsf{q}_i(x,t)\mathsf{q}_j(0)\rangle^{\mathrm{c}}_{\mathrm{ini}}$. Indeed, GGE equations of state give information about conserved charges $Q_i = \int \mathrm{d}x\, \mathsf{q}_i(x)$, but conserved densities $\mathsf{q}_i(x)$ are defined from these only up to total spatial derivatives of local fields. Thus any result from GHD for two-point correlation function $\langle \mathsf{q}_i(x)\mathsf{q}_j(0)\rangle^{\mathrm{c}}_{\mathrm{ini}}$ cannot depend on the precise definition of $\mathsf{q}_i(x)$. The Eulerian scaling limit (1.5) only probes large wavelengths, and derivative corrections to $\mathsf{q}_i(x)$ are expected to give vanishing contributions. This is why it is possible to obtain exact results purely from GHD for this scaling limit. Any

correction to the Eulerian scaling limit necessitates additional information, hence cannot lie entirely within the present GHD framework.

Euler-scaled dynamical correlations can be seen as being produced by "waves" of conserved quantities ballistically propagating in the fluid between the fields involved in the correlation function. The problem can thus be seen as that of propagating Euler-scale waves from the initial delta-function correlation (2.19), essentially using the evolution equation (2.22). This form of the problem is made more explicit in the case of two-point functions in Subsection 3.3 using hydrodynamic projection theory.

## 3.1 Generating higher-point correlation functions

The main idea of the method is to use responses to local (in the Euler sense) disturbance in order to generate dynamical correlations. Indeed, consider the state (1.1). The response to a small change of the local potential $\beta_i(x)$ at the point $x$ should provide information about the correlation between the observable $O$ (which can be a product of local observables) and the local conserved density $\mathsf{q}_i(x)$. At the Euler scale (1.5), the functional differentiation with respect to $\beta_i(x)$ brings down the density $\mathsf{q}_i(x)$, and does nothing else. This is clear in classical models as it follows from differentiation of the exponential function. In quantum models, terms coming from nontrivial commutators between local conserved densities are negligible at the Euler scale: they only give rise to derivatives of local operators, see [12, eqs. 91-93], which can be neglected in Eulerian correlation functions[5]. Therefore,

$$\Big\langle \mathsf{q}_i(x,0) \prod_k \mathcal{O}_k(x_k, t_k) \Big\rangle^{\text{Eul}}_{[n_0]} = -\frac{\delta}{\delta\beta_i(x)} \Big\langle \prod_k \mathcal{O}_k(x_k, t_k) \Big\rangle^{\text{Eul}}_{[n_0]}. \tag{3.1}$$

We see that Eulerian dynamical correlation functions are related to response functions. This constitutes a generalisation, both out of equilibrium and to the presence of the higher conserved charges of integrable models, of the fluctuation-dissipation theorem.

Let us consider Euler scale correlation functions (1.5) involving charge densities and currents. The one-point functions are given by (2.8), (2.9). Evolving in time and taking the Eulerian scaling limit is simple,

$$\langle \mathsf{q}_i(x,t)\rangle^{\text{Eul}}_{[n_0]} = \langle \mathsf{q}_i(x)\rangle^{\text{Eul}}_{[n_t]} = \int_{\mathcal{S}} \frac{d\boldsymbol{\theta}}{2\pi} p'(\boldsymbol{\theta}) n_t(x;\boldsymbol{\theta}) h^{\text{dr}}_i(x,t;\boldsymbol{\theta}) \tag{3.2}$$

$$\langle \mathsf{j}_i(x,t)\rangle^{\text{Eul}}_{[n_0]} = \langle \mathsf{j}_i(x)\rangle^{\text{Eul}}_{[n_t]} = \int_{\mathcal{S}} \frac{d\boldsymbol{\theta}}{2\pi} E'(\boldsymbol{\theta}) n_t(x;\boldsymbol{\theta}) h^{\text{dr}}_i(x,t;\boldsymbol{\theta}). \tag{3.3}$$

Higher-point functions with many insertions of conserved densities are obtained recursively as follows. Let $\prod_{k=1}^N \mathcal{O}_k(x_k, t_k)$ be a product of local observables at various space-time positions. It is convenient to assume that $t_N = 0$, without loss of generality as we can always evolve in time using (2.23). Assume that $\langle \prod_{k=1}^N \mathcal{O}_k(x_k, t_k)\rangle^{\text{Eul}}_{[n_0]}$ is known as a functional of $n_0(x;\theta)$. This is the case for $N = 1$ with $\mathcal{O}_1$ being a conserved density or current (see below for other one-point functions). From this, we may obtain correlation functions $\langle \prod_{k=1}^N \mathcal{O}_k(x_k, t_k + t) \mathcal{O}_{N+1}(x_{N+1}, 0)\rangle^{\text{Eul}}_{[n_0]}$ with $\mathcal{O}_{N+1} = \mathsf{q}_j$ for any $j$. This is of the same form as the correlation at order $N$: it contains $N + 1$ observables, where all $N$ previous local observables have been evolved for a time $t$, and a new conserved density has been inserted at time

---

[5]Note that at the Euler scale, $\mathsf{q}_i(x,t)$ is completely characterised by the corresponding conserved charge $Q_i$, hence only defined up to a total derivative.

$t_{N+1} = 0$. We obtain:

$$
\left\langle \prod_{k=1}^{N} \mathcal{O}_k(x_k, t_k + t)\, \mathfrak{q}_j(y, 0) \right\rangle_{[n_0]}^{\mathrm{Eul}} = -\frac{\partial}{\partial \beta_j(y)} \left\langle \prod_{k=1}^{N} \mathcal{O}_k(x_k, t_k) \right\rangle_{[n_t]}^{\mathrm{Eul}} \tag{3.4}
$$

$$
= \int_{\mathcal{S}} \mathrm{d}\boldsymbol{\alpha} \int_{\mathcal{S}} \mathrm{d}\boldsymbol{\theta} \int_{\mathbb{R}} \mathrm{d}z\, n_0(y; \boldsymbol{\alpha})\, f(y, 0; \boldsymbol{\alpha})\, h_j^{\mathrm{dr}}(y, 0; \boldsymbol{\alpha}) \times
$$

$$
\times \frac{\delta n_t(z; \boldsymbol{\theta})}{\delta n_0(y; \boldsymbol{\alpha})} \frac{\delta}{\delta \tilde{n}(z; \boldsymbol{\theta})} \left\langle \prod_{k=1}^{N} \mathcal{O}_k(x_k, t_k) \right\rangle_{[\tilde{n}]}^{\mathrm{Eul}} \Bigg|_{\tilde{n} = n_t}.
$$

We have used (3.1), (2.14) and (2.18). In this expression, $\delta n_t(z; \boldsymbol{\theta})/\delta n_0(y; \boldsymbol{\alpha})$ is the functional derivative of the time-evolved occupation function $n_t(z; \boldsymbol{\theta})$ with respect to variations of the initial condition $n_0(y; \boldsymbol{\alpha})$ from which it is evolved.

Density-density two-point functions take a particularly simple form thanks to the general formula

$$
\partial_\mu \int_{\mathcal{S}} \frac{\mathrm{d}\boldsymbol{\theta}}{2\pi} g(\boldsymbol{\theta}) n(\boldsymbol{\theta}) h^{\mathrm{dr}}(\boldsymbol{\theta}) = \int_{\mathcal{S}} \frac{\mathrm{d}\boldsymbol{\theta}}{2\pi} g^{\mathrm{dr}}(\boldsymbol{\theta}) \partial_\mu n(\boldsymbol{\theta}) h^{\mathrm{dr}}(\boldsymbol{\theta}) \tag{3.5}
$$

obtained in [13], where $\mu$ is any parameter on which a GGE state $n(\boldsymbol{\theta})$ may depend, and $g(\boldsymbol{\theta})$, $h(\boldsymbol{\theta})$ are any spectral functions (either $g$ is a vector field and $h$ is a scalar field, or vice versa). The functional derivative on the right-hand side in (3.4) may be evaluated using this along with (3.2) (specialized to $t = 0$), giving

$$
\langle \mathfrak{q}_i(x, t) \mathfrak{q}_j(y, 0) \rangle_{[n_0]}^{\mathrm{Eul}}
$$
$$
= \int_{\mathcal{S}} \mathrm{d}\boldsymbol{\alpha} \int_{\mathcal{S}} \mathrm{d}\boldsymbol{\theta}\, n_0(y; \boldsymbol{\alpha})\, f(y, 0; \boldsymbol{\alpha})\, h_j^{\mathrm{dr}}(y, 0; \boldsymbol{\alpha}) \frac{\delta n_t(x; \boldsymbol{\theta})}{\delta n_0(y; \boldsymbol{\alpha})} \rho_s(x, t; \boldsymbol{\theta}) h_i^{\mathrm{dr}}(x, t; \boldsymbol{\theta}). \tag{3.6}
$$

Recall that $\rho_s(x, t; \boldsymbol{\theta})$ is the state density (2.6) evaluated with respect to the occupation function at space-time position $(x, t)$. The density-current two-point function can be obtained similarly. Higher-point functions are obtained using (3.4) by further functional differentiation, using similar techniques.

The crucial objects in these formulae are the functional derivatives of the time-evolved occupation function $n_t(x; \boldsymbol{\theta})$ with respect to its initial condition $n_0(y; \boldsymbol{\alpha})$. These describe the dynamical responses of the fluid at time $t$ to a change of initial condition. The two-point function only involves the first derivative, while higher-point functions will involve higher derivatives.

Below it will be convenient to define the *propagator* as a simple conjugation of the first derivative of the evolution operator:

$$
\Gamma_{(y,0) \to (x,t)}(\boldsymbol{\theta}, \boldsymbol{\alpha}) = \left( n_t(x; \boldsymbol{\theta}) f(x, t; \boldsymbol{\theta}) \right)^{-1} \frac{\delta n_t(x; \boldsymbol{\theta})}{\delta n_0(y; \boldsymbol{\alpha})} n_0(y; \boldsymbol{\alpha}) f(y, 0; \boldsymbol{\alpha}). \tag{3.7}
$$

In terms of the propagator, the density-density two-point function takes the form

$$
\langle \mathfrak{q}_i(x, t) \mathfrak{q}_j(y, 0) \rangle_{[n_0]}^{\mathrm{Eul}}
$$
$$
= \int_{\mathcal{S}} \mathrm{d}\boldsymbol{\theta} \int_{\mathcal{S}} \mathrm{d}\boldsymbol{\alpha}\, \Gamma_{(y,0) \to (x,t)}(\boldsymbol{\theta}, \boldsymbol{\alpha}) \rho_p(x, t; \boldsymbol{\theta}) f(x, t; \boldsymbol{\theta}) h_i^{\mathrm{dr}}(x, t; \boldsymbol{\theta}) h_j^{\mathrm{dr}}(y, 0; \boldsymbol{\alpha}). \tag{3.8}
$$

Note that the propagator is a vector field as a function of its first argument, and a scalar field as a function of its second.

In the following, we concentrate on two-point functions: we explain how to evaluate the propagator via integral equations, and how to go beyond correlation functions involving conserved densities. It turns out that the propagator, as defined in (3.7), satisfies a linear integral

equation whose source term and kernel stay well defined even at points where the occupation function vanish. We leave for future studies the developments of expressions for higher-point functions and the evaluation of higher-derivatives of the time evolved occupation function.

## 3.2 Exact two-point functions of densities and currents

The derivation of the following formulae, based on the techniques introduced above, is presented in Appendix B.1. Here we describe the main results.

In order to express the results, it is convenient to introduce the "star-dressing" operation, which for a vector field $g(\boldsymbol{\theta})$ and a GGE occupation function $n(\boldsymbol{\theta})$ is defined by

$$g^{*\mathrm{dr}}(\boldsymbol{\theta}) = \left( T n g \right)^{\mathrm{dr}}(\boldsymbol{\theta}) = g^{\mathrm{dr}}(\boldsymbol{\theta}) - g(\boldsymbol{\theta}). \tag{3.9}$$

Note that without interaction, we have $g^{*\mathrm{dr}} = 0$. We will also need the effective acceleration $a^{\mathrm{eff}}_{[n_0]}(x; \boldsymbol{\theta})$ introduced in [12]. This is a functional of $n_0(x; \boldsymbol{\theta})$ (seen as a function of $(x, \boldsymbol{\theta})$). It is defined as $a^{\mathrm{eff}}_{[n_0]}(x; \boldsymbol{\theta}) = -(\partial_x w(x))^{\mathrm{dr}}(x, 0; \boldsymbol{\theta})/(p')^{\mathrm{dr}}(x, 0; \boldsymbol{\theta})$ where $w(x; \boldsymbol{\theta}) = \sum_i \beta_i(x) h_i(\boldsymbol{\theta})$ is a scalar field, the TBA driving term of the GGE $n_0(x; \boldsymbol{\theta})$ (see (2.17)). For our purpose, we may write it in the equivalent forms

$$a^{\mathrm{eff}}_{[n_0]}(x; \boldsymbol{\theta}) = \frac{\partial_x n_0(x; \boldsymbol{\theta})}{2\pi \rho_{\mathrm{p}}(x, 0; \boldsymbol{\theta}) f(x, 0; \boldsymbol{\theta})} = -\frac{\partial_x \epsilon(x, 0; \boldsymbol{\theta})}{2\pi \rho_{\mathrm{s}}(x, 0; \boldsymbol{\theta})}. \tag{3.10}$$

The effective acceleration encodes the inhomogeneity of the fluid state $n_0(x; \boldsymbol{\theta})$.

It will be convenient to see the propagator $\Gamma_{(y,0) \to (x,t)}(\boldsymbol{\theta}, \boldsymbol{\alpha})$ as the kernel of a linear integral operator acting on scalar fields via contraction on the spectral parameter $\boldsymbol{\alpha}$:

$$\left( \Gamma_{(y,0) \to (x,t)} g \right)(\boldsymbol{\theta}) = \int_{\mathcal{S}} \mathrm{d}\boldsymbol{\alpha} \, \Gamma_{(y,0) \to (x,t)}(\boldsymbol{\theta}, \boldsymbol{\alpha}) g(\boldsymbol{\alpha}). \tag{3.11}$$

This can be interpreted as bringing the spectral function $g$ from the point $(y, 0)$ to the point $(x, t)$ starting in the initial state $n_0$. We show in Appendix B.1 that the propagator satisfies, for $x, y > x_0$ (recall (2.23) for the quantity $x_0$), the following integral equation:

$$\begin{aligned}
\left( \Gamma_{(y,0) \to (x,t)} g \right)(\boldsymbol{\theta}) &- 2\pi a^{\mathrm{eff}}_{[n_0]}(u; \boldsymbol{\theta}) \int_{x_0}^x \mathrm{d}z \left( \rho_{\mathrm{s}}(z, t) f(z, t) \Gamma_{(y,0) \to (z,t)} g \right)^{*\mathrm{dr}}(z, t; \boldsymbol{\theta}) \\
&= \delta(y - u) g(\boldsymbol{\theta}) - 2\pi a^{\mathrm{eff}}_{[n_0]}(u; \boldsymbol{\theta}) \Theta(u - y) \left( \rho_{\mathrm{s}}(y, 0) f(y, 0) g \right)^{*\mathrm{dr}}(y, 0; \boldsymbol{\theta}) \\
&\text{with} \quad u = u(x, t; \boldsymbol{\theta}),
\end{aligned} \tag{3.12}$$

where $\Theta(\dots)$ is Heavyside's Theta-function. This defines $\Gamma_{(y,0) \to (x,t)}(\boldsymbol{\theta}, \boldsymbol{\alpha})$. In this and other equations below, functions such as $\rho_{\mathrm{s}}(x, t; \boldsymbol{\theta})$ and $f(x, t; \boldsymbol{\theta})$ with *omitted* spectral argument $\boldsymbol{\theta}$, are to be seen as diagonal integral operators, acting simply by multiplication by the associated quantity.

Remark that if the initial state is homogeneous, in which case the evolution is trivial $n_t(x; \boldsymbol{\theta}) = n(\boldsymbol{\theta})$, then we have $a^{\mathrm{eff}}_{[n_0]}(u; \boldsymbol{\theta}) = 0$ and $u = x - v^{\mathrm{eff}}(\boldsymbol{\theta})t$, and we find

$$\Gamma_{(y,0) \to (x,t)}(\boldsymbol{\theta}, \boldsymbol{\alpha}) = \delta(x - y - v^{\mathrm{eff}}(\boldsymbol{\theta})t) \delta_{\mathcal{S}}(\boldsymbol{\theta} - \boldsymbol{\alpha}) \qquad \text{(homogeneous states).} \tag{3.13}$$

Here and below, $\delta_{\mathcal{S}}(\boldsymbol{\theta} - \boldsymbol{\alpha}) = \delta(\theta - \alpha)\delta_{b,a}$ for $\boldsymbol{\alpha} = (\alpha, a)$ and $\boldsymbol{\theta} = (\theta, b)$. In the absence of interaction, we have $u = x - v^{\mathrm{gr}}(\boldsymbol{\theta})t$ where $v^{\mathrm{gr}}(\boldsymbol{\theta}) = E'(\boldsymbol{\theta})/p'(\boldsymbol{\theta})$ is the group velocity, and

$$\Gamma_{(y,0) \to (x,t)}(\boldsymbol{\theta}, \boldsymbol{\alpha}) = \delta(x - y - v^{\mathrm{gr}}(\boldsymbol{\theta})t) \delta_{\mathcal{S}}(\boldsymbol{\theta} - \boldsymbol{\alpha}) \qquad \text{(without interactions).} \tag{3.14}$$

Further, at $t = 0$, one obtains

$$\Gamma_{(y,0)\to(x,0)}(\boldsymbol{\theta}, \boldsymbol{\alpha}) = \delta(x - y)\,\delta_{\mathcal{S}}(\boldsymbol{\theta} - \boldsymbol{\alpha}) \qquad \text{(vanishing time difference)}. \tag{3.15}$$

The propagator (3.12) allows one to evaluate two-point functions of conserved densities in inhomogeneous states as per (3.8). For two-point functions involving currents, the results are simple modifications of the above, where the effective velocity multiplies the dressed one-particle eigenvalues. The results are

$$\langle \mathsf{q}_i(x,t)\mathsf{q}_j(y,0)\rangle_{[n_0]}^{\text{Eul}} = \int_{\mathcal{S}} d\boldsymbol{\theta} \int_{\mathcal{S}} d\boldsymbol{\alpha}\, \Gamma_{(y,0)\to(x,t)}(\boldsymbol{\theta}, \boldsymbol{\alpha})\rho_{\text{p}}(x,t;\boldsymbol{\theta})f(x,t;\boldsymbol{\theta}) \times \tag{3.16}$$
$$\times h_i^{\text{dr}}(x,t;\boldsymbol{\theta})h_j^{\text{dr}}(y,0;\boldsymbol{\alpha})$$

$$\langle \mathsf{j}_i(x,t)\mathsf{q}_j(y,0)\rangle_{[n_0]}^{\text{Eul}} = \int_{\mathcal{S}} d\boldsymbol{\theta} \int_{\mathcal{S}} d\boldsymbol{\alpha}\, \Gamma_{(y,0)\to(x,t)}(\boldsymbol{\theta}, \boldsymbol{\alpha})\rho_{\text{p}}(x,t;\boldsymbol{\theta})f(x,t;\boldsymbol{\theta}) \times \tag{3.17}$$
$$\times v^{\text{eff}}(x,t;\boldsymbol{\theta})h_i^{\text{dr}}(x,t;\boldsymbol{\theta})h_j^{\text{dr}}(y,0;\boldsymbol{\alpha})$$

$$\langle \mathsf{q}_i(x,t)\mathsf{j}_j(y,0)\rangle_{[n_0]}^{\text{Eul}} = \int_{\mathcal{S}} d\boldsymbol{\theta} \int_{\mathcal{S}} d\boldsymbol{\alpha}\, \Gamma_{(y,0)\to(x,t)}(\boldsymbol{\theta}, \boldsymbol{\alpha})\rho_{\text{p}}(x,t;\boldsymbol{\theta})f(x,t;\boldsymbol{\theta}) \times \tag{3.18}$$
$$\times h_i^{\text{dr}}(x,t;\boldsymbol{\theta})v^{\text{eff}}(y,0;\boldsymbol{\alpha})h_j^{\text{dr}}(y,0;\boldsymbol{\alpha})$$

$$\langle \mathsf{j}_i(x,t)\mathsf{j}_j(y,0)\rangle_{[n_0]}^{\text{Eul}} = \int_{\mathcal{S}} d\boldsymbol{\theta} \int_{\mathcal{S}} d\boldsymbol{\alpha}\, \Gamma_{(y,0)\to(x,t)}(\boldsymbol{\theta}, \boldsymbol{\alpha})\rho_{\text{p}}(x,t;\boldsymbol{\theta})f(x,t;\boldsymbol{\theta}) \times \tag{3.19}$$
$$\times v^{\text{eff}}(x,t;\boldsymbol{\theta})h_i^{\text{dr}}(x,t;\boldsymbol{\theta})v^{\text{eff}}(y,0;\boldsymbol{\alpha})h_j^{\text{dr}}(y,0;\boldsymbol{\alpha}).$$

These expressions are similar to those obtained in [13], except for the nontrivial propagator $\Gamma_{(y,0)\to(x,t)}(\boldsymbol{\theta}, \boldsymbol{\alpha})$. In the homogeneous case, using (3.13), we indeed recover the result of [13]. Formulae (3.16), (3.17), (3.18) and (3.19), with (3.12), are the main results of this paper.

It is natural to separate the propagator into two terms,

$$\Gamma_{(y,0)\to(x,t)}(\boldsymbol{\theta}, \boldsymbol{\alpha}) = \delta(y - u(x,t;\boldsymbol{\theta}))\delta_{\mathcal{S}}(\boldsymbol{\theta} - \boldsymbol{\alpha}) + \Delta_{(y,0)\to(x,t)}(\boldsymbol{\theta}, \boldsymbol{\alpha}). \tag{3.20}$$

We will refer to the first term as the *direct propagator*, and the last as the *indirect propagator*. As explained in Appendix B.2, the indirect propagator satisfies the following linear integral equation:

$$\frac{\left(\Delta_{(y,0)\to(x,t)}g\right)(\boldsymbol{\theta})}{2\pi a_{[n_0]}^{\text{eff}}(u(x,t;\boldsymbol{\theta});\boldsymbol{\theta})} = \left(\mathsf{W}_{(y,0)\to(x,t)}g\right)(\boldsymbol{\theta}) + \int_{x_0}^x dz \left(\rho_{\text{s}}(z,t)f(z,t)\Delta_{(y,0)\to(z,t)}g\right)^{*\text{dr}}(z,t;\boldsymbol{\theta}), \tag{3.21}$$

where the source term is

$$\left(\mathsf{W}_{(y,0)\to(x,t)}g\right)(\boldsymbol{\theta}) = \int_{x_0}^x dz \sum_{\boldsymbol{\gamma}\in\boldsymbol{\theta}_\star(z,t;y)} \frac{\rho_{\text{s}}(z,t;\boldsymbol{\gamma})n_0(y;\boldsymbol{\gamma})f(y,0;\boldsymbol{\gamma})}{|u'(z,t;\boldsymbol{\gamma})|} T^{\text{dr}}(z,t;\boldsymbol{\theta},\boldsymbol{\gamma})g(\boldsymbol{\gamma})$$
$$- \Theta(u(x,t;\boldsymbol{\theta}) - y)\left(\rho_{\text{s}}(y,0)f(y,0)g\right)^{*\text{dr}}(y,0;\boldsymbol{\theta}). \tag{3.22}$$

Here the dressed scattering operator is

$$T^{\text{dr}} = (1 - Tn)^{-1}T, \tag{3.23}$$

and $T^{\text{dr}}(z,t;\boldsymbol{\theta},\boldsymbol{\gamma})$ is, as a function of $\boldsymbol{\theta},\boldsymbol{\gamma}$, the kernel of $T^{\text{dr}}(z,t)$ (with dressing with respect to the state $[n_{z,t}]$). $T^{\text{dr}}(z,t;\boldsymbol{\theta},\boldsymbol{\gamma})$ is a vector field as function of the $\theta$, and a scalar field as function of $\gamma$. The root set is

$$\boldsymbol{\theta}_\star(x,t;y) = \{\boldsymbol{\theta} : u(x,t;\boldsymbol{\theta}) = y\}. \tag{3.24}$$

If the effective velocity is monotonic with respect to the rapidity, by virtue of (2.28), the function $u(x, t; \boldsymbol{\theta})$ is locally invertible on the rapidity $\theta$, wherefore the set $\boldsymbol{\theta}_\star(x, t; y)$ contains at most one element $\boldsymbol{\theta} = (\theta_a, a)$ per particle type $a \in \mathcal{A}$. In general, however, the set may contain more solution per particle type. In terms of (3.20), we have for instance

$$\langle q_i(x, t) q_j(y, 0) \rangle_{[n_0]}^{\text{Eul}} \tag{3.25}$$
$$= \sum_{\boldsymbol{\gamma} \in \boldsymbol{\theta}_\star(x, t; y)} \frac{\rho_s(x, t; \boldsymbol{\gamma}) n_0(y; \boldsymbol{\gamma}) f(y, 0; \boldsymbol{\gamma})}{|u'(x, t; \boldsymbol{\gamma})|} h_i^{\text{dr}}(x, t; \boldsymbol{\gamma}) h_j^{\text{dr}}(y, 0; \boldsymbol{\gamma}) +$$
$$+ \int_{\mathcal{S}} \mathrm{d}\boldsymbol{\theta} \int_{\mathcal{S}} \mathrm{d}\boldsymbol{\alpha} \, \Delta_{(y, 0) \to (x, t)}(\boldsymbol{\theta}, \boldsymbol{\alpha}) \rho_p(x, t; \boldsymbol{\theta}) f(x, t; \boldsymbol{\theta}) h_i^{\text{dr}}(x, t; \boldsymbol{\theta}) h_j^{\text{dr}}(y, 0; \boldsymbol{\alpha})$$

(and remark that $n_0(y; \boldsymbol{\gamma}) f(y, 0; \boldsymbol{\gamma}) = n_t(x; \boldsymbol{\gamma}) f(x, t; \boldsymbol{\gamma})$ in the first term on the right-hand side).

## 3.3 Connection with hydrodynamic projections

The main ideas of hydrodynamic projections, in the cases of two-point functions at the Euler scale, can be gathered within two statements. First, correlations are transported solely by (ballistically propagating) conserved densities. Second, the overlap between a local observable and such a propagating conserved density, in the fluid cell containing the local observable, is obtained by the hydrodynamic inner product (2.21) within this cell. Using this, the expressions (3.17)-(3.19), involving currents, are in fact consequences of (3.16) using hydrodynamic projection theory. Here we first re-write the expressions obtained above in the hydrodynamic-projection form. We then show how taking this form implies the Euler-scale fluctuation-dissipation principle we have used to derive the expressions (3.17)-(3.19).

### 3.3.1 Re-writing in hydrodynamic-projection form

First, the integral operator $S_{(y, 0) \to (x, t)}$, acting on scalar fields and giving vector fields, that generates the charge two-point function as

$$\langle q_i(x, t) q_j(y, 0) \rangle_{[n_0]}^{\text{Eul}} = \int_{\mathcal{S}} \mathrm{d}\boldsymbol{\theta} \, h_i(\boldsymbol{\theta}) \big( S_{(y, 0) \to (x, t)} h_j \big)(\boldsymbol{\theta}) \tag{3.26}$$

is given by

$$S_{(y, 0) \to (x, t)} = (1 - n_{x, t} T)^{-1} \rho_p(x, t) f(x, t) \Gamma_{(y, 0) \to (x, t)} (1 - T^{\text{T}} n_{y, 0})^{-1}. \tag{3.27}$$

Relation (3.27) is simply a re-writing of (3.16). Note that by symmetry of the correlation functions, $S_{(y, 0) \to (x, t)}^{\text{T}} = S_{(x, t) \to (y, 0)}$ where T denotes transpose. From hydrodynamic projection, it is known that correlation functions involving currents are obtained by using the linearized Euler operator, which, in a GGE state $n(\theta)$, is given by [13]

$$A_{[n]} = (1 - nT)^{-1} v^{\text{eff}} (1 - nT); \tag{3.28}$$

it acts on vector fields and gives vector fields. Re-writing (3.17)-(3.19), currents correlations are obtained as:

$$A_{[n_{x, t}]} S_{(y, 0) \to (x, t)} \quad \text{(for the current on the left)}$$
$$S_{(y, 0) \to (x, t)} A_{[n_{y, 0}]}^{\text{T}} \quad \text{(for the current on the right)} \tag{3.29}$$
$$A_{[n_{x, t}]} S_{(y, 0) \to (x, t)} A_{[n_{y, 0}]}^{\text{T}} \quad \text{(for both observables being currents).}$$

These are indeed expressions that are expected form hydrodynamic projection principles [13]. In the homogeneous case we have that $n_t = n_0$ and, using (3.13), that $S_{(y,0)\to(x,t)} = S_{(x,0)\to(y,-t)} = S_{(x,t)\to(y,0)}$. In this case one usually denotes the operator as $S(x-y,t)$, and we recover $S(x-y,t)^{\mathrm{T}} = S(x-y,t)$.

Further, according to hydrodynamic projection principles, one would expect $S_{(y,0)\to(x,t)}$ to solve the evolution equation

$$\partial_t S_{(y,0)\to(x,t)} + \partial_x \big(A_{[n_{x,t}]} S_{(y,0)\to(x,t)}\big) = 0, \tag{3.30}$$

with the initial condition $S_{(y,0)\to(x,0)} = \delta(x-y)C_{[n_{x,0}]}$. Here $C_{[n]}$, for a GGE state $n(\theta)$, is the correlation operator. Its matrix elements, in the space of conserved densities, are the connected integrated two-point functions $C_{[n]}^{q_i q_j}$ (see (2.20) and (2.21)), and as an operator it is [13]

$$C_{[n]} = (1-nT)^{-1} \rho_{\mathrm{p}} f (1-T^{\mathrm{T}}n)^{-1}. \tag{3.31}$$

Eq. (3.30) is the generalisation to space-time dependent states of the equation that was solved in [13] in order to obtain Euler-scale correlations in homogeneous states. It is an explicit expression of the problem of propagating Euler-scale waves from the initial delta-function correlation (2.19), in the case of two-point correlations. It is simple to verify that indeed (3.30) follows from the results obtained: the initial condition holds by using (3.15), and (3.30) follows from (3.16) and (3.17) and the conservation law for local densities.

### 3.3.2 From hydrodynamic projections to Euler-scale fluctuation-dissipation principle

Above, we saw the hydrodynamic projection evolution problem emerging as a consequence of defining space-time correlation functions using an Euler-scale fluctuation-dissipation principle. Let us now reverse the logic: let us take (3.26) with (3.30), along with the appropriate correct initial condition as stated after (3.30), as a definition of the scaled dynamical two-point functions of conserved densities. From this, let us show that the Euler-scale fluctuation-dissipation principle (3.1) holds for two-point functions of conserved densities. The relation $\partial_t \langle q_i(x,t)\rangle_{[n_0]}^{\mathrm{Eul}} + \partial_x \langle j_i(x,t)\rangle_{[n_0]}^{\mathrm{Eul}} = 0$ follows from the basic GHD results. Taking functional derivatives with respect to $\beta(y)$, it implies

$$\partial_t \frac{\delta \langle q_i(x,t)\rangle_{[n_0]}^{\mathrm{Eul}}}{\delta\beta(y)} + \partial_x \frac{\delta \langle j_i(x,t)\rangle_{[n_0]}^{\mathrm{Eul}}}{\delta\beta(y)} = 0.$$

The result for these functional derivatives are the expressions on the right-hand sides of (3.16) and (3.17). Using these in the above equation, we indeed find that (3.27) satisfies (3.30) with the correct initial condition. Hence, we conclude that $\delta\langle q_i(x,t)\rangle_{[n_0]}^{\mathrm{Eul}}/\delta\beta(y) = \langle q_i(x,t)q_j(y,0)\rangle_{[n_0]}^{\mathrm{Eul}}$, as it should.

### 3.4 Two-point correlations of generic local observables

Let $n(\boldsymbol{\theta})$ be a GGE. We consider the inner product $\langle\mathcal{O}|q_i\rangle_{[n]}$, and it is assumed that spatial correlations within the state $n(\boldsymbol{\theta})$ decay faster than the inverse distance. Without loss of generality we assume $\mathcal{O}$ to be hermitian. If the average of $\mathcal{O}$ is known in a generic GGE $n(\boldsymbol{\theta})$, then we can use

$$\langle\mathcal{O}|q_i\rangle_{[n]} = -\frac{\partial}{\partial\beta_i}\langle\mathcal{O}\rangle_{[n]}. \tag{3.32}$$

In integral operator form, by linearity of the result in $h_i$, there exists a scalar field $V^{\mathcal{O}}(\boldsymbol{\theta})$ (a hydrodynamic spectral function associated to the local field $\mathcal{O}$), which is also a functional of $n(\boldsymbol{\theta})$, such that

$$\langle\mathcal{O}|q_i\rangle_{[n]} = \int_{\mathcal{S}} d\boldsymbol{\theta}\, \rho_{\mathrm{p}}(\boldsymbol{\theta}) f(\boldsymbol{\theta}) V^{\mathcal{O}}(\boldsymbol{\theta}) h_i^{\mathrm{dr}}(\boldsymbol{\theta}). \tag{3.33}$$

Here for later convenience we introduced the factors $\rho_{\mathrm{p}}(\boldsymbol{\theta})f(\boldsymbol{\theta})$ and used $h_i^{\mathrm{dr}}(\boldsymbol{\theta})$ instead of $h_i(\boldsymbol{\theta})$. For instance, according to results of [13], we have

$$V^{\mathsf{q}_i} = h_i^{\mathrm{dr}}, \qquad V^{\mathsf{j}_i} = v^{\mathrm{eff}} h_i^{\mathrm{dr}}. \tag{3.34}$$

We denote by $(C_{[n]})_{ij} = C_{[n]}^{\mathsf{q}_i\mathsf{q}_j} = \langle \mathsf{q}_i|\mathsf{q}_j\rangle_{[n]}$ the overlap, within a fluid cell of GGE $n(\boldsymbol{\theta})$, between conserved densities, as per (2.21) – this is the correlation matrix of conserved densities, the case $N = 2$ of (2.20). In integral operator form, this is the correlation operator (3.31) introduced above. Let us now consider a generalized fluid, with space-time state described by $n_{x,t}(\boldsymbol{\theta}) \equiv n_t(x;\boldsymbol{\theta})$. According to hydrodynamic projection principles, Euler-scale correlation functions can be written as

$$
\begin{aligned}
&\langle \mathcal{O}(x,t)\mathcal{O}'(y,0)\rangle_{[n_0]}^{\mathrm{Eul}} \\
&= \sum_{i,j,k,l} \langle \mathcal{O}|\mathsf{q}_i\rangle_{[n_{x,t}]} (C_{[n_{x,t}]})_{ij}^{-1} \langle \mathsf{q}_j(x,t)\mathsf{q}_k(y,0)\rangle_{[n_0]} (C_{[n_{y,0}]})_{kl}^{-1} \langle \mathsf{q}_l|\mathcal{O}\rangle_{[n_{y,0}]} \\
&= \rho_{\mathrm{p}}(x,t)f(x,t)V^{\mathcal{O}}(x,t)(1 - T^{\mathrm{T}}n_{x,t})^{-1} \times \\
&\quad \times C_{[n_{x,t}]}^{-1} S_{(y,0)\to(x,t)} C_{[n_{y,0}]}^{-1} \times \\
&\quad \times (1 - n_{y,0}T)^{-1}\rho_{\mathrm{p}}(y,0)f(y,0)V^{\mathcal{O}}(y,0) \\
&= \int_{\mathcal{S}} \mathrm{d}\boldsymbol{\theta}\, \rho_{\mathrm{p}}(x,t;\boldsymbol{\theta})f(x,t;\boldsymbol{\theta})V^{\mathcal{O}}(x,t;\boldsymbol{\theta}) \Big(\Gamma_{(y,0)\to(x,t)}V^{\mathcal{O}'}(y,0)\Big)(\boldsymbol{\theta}). \quad (3.35)
\end{aligned}
$$

The first equality is explained as follows. Reading from the right to the left, we first overlap the observable $\mathcal{O}'$ with a complete set of conserved quantities $\mathsf{q}_l$, with respect to the inner product (2.21) for the state $n_{y,0}$ (at the space-time point $(y,0)$, where the observable $\mathcal{O}'$ lies). Because the conserved quantities $\mathsf{q}_l$'s don't necessarily form an orthonormal set, we introduced the inverse correlation matrix $C_{[n_{y,0}]}$ at the space-time point $(y,0)$. These two factors represent the amplitude for $\mathcal{O}'$ to produce Euler propagating waves of conserved quantities. We then "transport" these waves from $(y,0)$ to $(x,t)$ by using the dynamical two-point function $\langle \mathsf{q}_j(x,t)\mathsf{q}_k(y,0)\rangle_{[n_0]}$ between conserved densities. Finally, we represent the amplitude for the transported wave to correlate with $\mathcal{O}$ by overlapping with $\mathcal{O}$ with respect to the inner product at $x,t$, introducing the inverse correlation matrix $C_{[n_{x,t}]}$ for orthonormality. The second equality is a re-writing in terms of integral operators, using (3.33) and (3.26). Finally, the last equality is obtained by replacing with the expressions (3.27) and (3.31). As a check, note that using (3.34), the above indeed reproduces the formulae (3.16)-(3.19). In particular, in homogeneous states, we use (3.13) and find

$$
\begin{aligned}
\langle \mathcal{O}(x,t)\mathcal{O}'(0,0)\rangle_{[n]}^{\mathrm{Eul}} &= \int_{\mathcal{S}} \mathrm{d}\boldsymbol{\theta}\, \delta(x - v^{\mathrm{eff}}(\boldsymbol{\theta})t)\rho_{\mathrm{p}}(\boldsymbol{\theta})f(\boldsymbol{\theta})V^{\mathcal{O}}(\boldsymbol{\theta})V^{\mathcal{O}'}(\boldsymbol{\theta}) \\
&= t^{-1} \sum_{\boldsymbol{\theta}\in\boldsymbol{\theta}_\star(\xi)} \frac{\rho_{\mathrm{p}}(\boldsymbol{\theta})f(\boldsymbol{\theta})}{|(v^{\mathrm{eff}})'(\boldsymbol{\theta})|} V^{\mathcal{O}}(\boldsymbol{\theta})V^{\mathcal{O}'}(\boldsymbol{\theta}), \quad (3.36)
\end{aligned}
$$

where $\xi = x/t$ and $\boldsymbol{\theta}_\star(\xi)$ is the set of solutions to $v^{\mathrm{eff}}(\boldsymbol{\theta}) = \xi$.

It turns out that, in integrable QFT, there exists a formula for the averages $\langle \mathcal{O}\rangle_{[n]}$ in GGEs, for any local field $\mathcal{O}$ [43, 44]. This formula, called the Leclair-Mussardo formula, involves an infinite summation over multiple integrals of form factors of the field $\mathcal{O}$. Nevertheless, its truncations can be used to numerically approximate expectation values. The Leclair-Mussardo formula was proven in [81], and it was used in [7] in order to provide further evidence for the proposed one-point averages of currents.

The formula has the structure of a sum over all numbers of particles $k$ of "connected" diagonal matrix elements $F_k^{\mathcal{O}}(\boldsymbol{\theta}_1,\ldots,\boldsymbol{\theta}_k) = \langle \boldsymbol{\theta}_1,\ldots\boldsymbol{\theta}_k|\mathcal{O}|\boldsymbol{\theta}_1,\ldots,\boldsymbol{\theta}_k\rangle_{\text{conn.}}$ of the field $\mathcal{O}$ (this is defined in [43]). Consider a GGE $n(\boldsymbol{\theta})$. Then the formula is

$$\langle \mathcal{O}\rangle_{[n]} = \sum_{k=0}^{\infty} \frac{1}{k!} \int_{\mathcal{S}^{\times k}} \prod_{j=1}^{k}\left(\frac{\mathrm{d}\boldsymbol{\theta}_j}{2\pi}\, n(\boldsymbol{\theta}_j)\right) F_k^{\mathcal{O}}(\boldsymbol{\theta}_1,\ldots,\boldsymbol{\theta}_k). \tag{3.37}$$

Recall that in our simplified notation, $\boldsymbol{\theta}$ represents the combination of a rapidity and any particle type the model may admit.

Importantly, in this formula, the information about the state is fully contained within the integration measure $\mathrm{d}\boldsymbol{\theta}\, n(\boldsymbol{\theta})$. Using (3.32) and (2.12), we therefore find

$$\langle \mathcal{O}|\mathsf{q}_i\rangle_{[n]} = \sum_{k=1}^{\infty} \frac{1}{k!} \int_{\mathcal{S}^{\times k}} \prod_{j=1}^{k}\left(\frac{\mathrm{d}\boldsymbol{\theta}_j}{2\pi}\, n(\boldsymbol{\theta}_j)\right) \sum_{j=1}^{k} h_i^{\mathrm{dr}}(\boldsymbol{\theta}_j) f(\boldsymbol{\theta}_j) F_k^{\mathcal{O}}(\boldsymbol{\theta}_1,\ldots,\boldsymbol{\theta}_k). \tag{3.38}$$

The function $F_k^{\mathcal{O}}$ is symmetric in all its arguments, and we may identify

$$V^{\mathcal{O}}(\boldsymbol{\theta}) = \sum_{k=0}^{\infty} \frac{1}{k!} \int_{\mathcal{S}^{\times k}} \prod_{j=1}^{k}\left(\frac{\mathrm{d}\boldsymbol{\theta}_j}{2\pi}\, n(\boldsymbol{\theta}_j)\right) (2\pi\rho_s(\boldsymbol{\theta}))^{-1} F_{k+1}^{\mathcal{O}}(\boldsymbol{\theta}_1,\ldots,\boldsymbol{\theta}_k,\boldsymbol{\theta}), \tag{3.39}$$

where $\rho_s(\boldsymbol{\theta})$ is the density of state, given in (2.6). The state dependence is within the integration measure and the density of state; the regularized diagonal matrix element $F_{k+1}^{\mathcal{O}}(\boldsymbol{\theta}_1,\ldots,\boldsymbol{\theta}_k,\boldsymbol{\theta})$ is purely a property of the field $\mathcal{O}$. It is interesting to re-specialize to $\mathcal{O}$ being a conserved density or current in order to verify that one indeed recovers (3.34) from (3.39). This is done in Appendix C.

Using (3.27) and reverting to the explicit notation $\Gamma_{(y,0)\to(x,t)}(\boldsymbol{\theta},\boldsymbol{\alpha})$ for the propagator via (3.11), we thus obtain

$$\langle \mathcal{O}(x,t)\mathcal{O}'(y,0)\rangle_{[n_0]}^{\mathrm{Eul}} \tag{3.40}$$

$$= \int_{\mathcal{S}} \mathrm{d}\boldsymbol{\theta} \int_{\mathcal{S}} \mathrm{d}\boldsymbol{\alpha}\, \Gamma_{(y,0)\to(x,t)}(\boldsymbol{\theta},\boldsymbol{\alpha})\, \rho_{\mathrm{p}}(x,t;\boldsymbol{\theta}) f(x,t;\boldsymbol{\theta}) \times$$

$$\times \sum_{k,k'=0}^{\infty} \frac{1}{k!(k')!} \int_{\mathcal{S}^{\times k}} \prod_{j=1}^{k}\left(\frac{\mathrm{d}\boldsymbol{\theta}_j}{2\pi}\, n_t(x;\boldsymbol{\theta}_j)\right) \int_{\mathcal{S}^{\times k'}} \prod_{j=1}^{k'}\left(\frac{\mathrm{d}\boldsymbol{\theta}_j'}{2\pi}\, n_0(y;\boldsymbol{\theta}_j')\right) \times$$

$$\times \left(2\pi\rho_s(x,t;\boldsymbol{\theta})\right)^{-1} F_{k+1}^{\mathcal{O}}(\boldsymbol{\theta}_1,\ldots,\boldsymbol{\theta}_k,\boldsymbol{\theta}) \left(2\pi\rho_s(y,0;\boldsymbol{\alpha})\right)^{-1} F_{k'+1}^{\mathcal{O}'}(\boldsymbol{\theta}_1',\ldots,\boldsymbol{\theta}_{k'}',\boldsymbol{\alpha}).$$

It is remarkable that such a complete formula exists in integrable field theory, for very general dynamical Euler-scale two-point correlation functions of local fields in inhomogeneous, non-stationary states. This formula is new both in the inhomogeneous case, and in the case of a homogeneous GGE; in the latter case, recall that the propagator simplifies to (3.13).

**Remark.** It is very likely that the form factor series (3.40), in homogeneous GGEs, can be obtained directly using an appropriate spectral expansion of the two-point function. Indeed, the structure of this series is extremely suggestive of the techniques introduced in [53, 56], based on the ideas of the Gelfand-Naimark-Segal construction. From these techniques, the trace expression representing the GGE average of a product of local fields is expressed as an expansion in "GGE form factors" very similar to the form factor expansion of vacuum two-point functions, in which each GGE form factor is itself a GGE trace of a single local field with additional particle creation / annihilation operator inserted. The leading term at the Euler

scale is that with one particle and its hole at the same rapidity, so that, pictorially,

$$\frac{\mathrm{Tr}\left(e^{-\sum_i \beta_i Q_i}\mathcal{O}(x,t)\mathcal{O}'(y,0)\right)}{\mathrm{Tr}\left(e^{-\sum_i \beta_i Q_i}\right)}$$

$$\sim \int d\boldsymbol{\theta}\, \frac{\mathrm{Tr}\left(e^{-\sum_i \beta_i Q_i}\mathcal{O}(x,t)A(\boldsymbol{\theta})A^\dagger(\boldsymbol{\theta})\right)}{\mathrm{Tr}\left(e^{-\sum_i \beta_i Q_i}\right)}\frac{\mathrm{Tr}\left(e^{-\sum_i \beta_i Q_i}\mathcal{O}'(y,0)A(\boldsymbol{\theta})A^\dagger(\boldsymbol{\theta})\right)}{\mathrm{Tr}\left(e^{-\sum_i \beta_i Q_i}\right)}. \quad (3.41)$$

Each single-field trace can be evaluated using the Leclair-Mussardo formula, giving a right-hand side similar to that of (3.40). We hope to come back to this problem in a future work.

## 4 Examples

### 4.1 Sinh-Gordon and Lieb-Liniger models

#### 4.1.1 Sinh-Grodon model

The sinh-Gordon model is an integrable relativistic QFT with Lagrangian density

$$\mathcal{L} = \frac{1}{2}\partial_\mu \Phi \partial^\mu \Phi - \frac{M^2}{g^2}(\cosh(g\Phi) - 1) \quad (4.1)$$

for a real scalar field $\Phi$, where $g$ is a coupling parameter and $M$ is a mass scale. Its TBA description contains a single particle of Fermionic type, so that $\mathcal{S} = \mathbb{R}$, $\boldsymbol{\theta} = \theta$ and $f(\theta) = 1 - n(\theta)$. We may choose $\theta$ as the rapidity, with $p(\theta) = m\sinh\theta$ and $E(\theta) = m\cosh\theta$, and the physical mass and differential scattering phase are given by

$$m^2 = \frac{\sin \pi a}{\pi a}M^2, \qquad \varphi(\theta, \alpha) = \frac{2\sin \pi a}{\sinh^2(\theta - \alpha) + \sin^2 \pi a}, \qquad \text{with} \quad a = \frac{g^2}{8\pi + g^2}. \quad (4.2)$$

As a set of natural local fields, one may consider the local conserved densities and currents of the model. They correspond to the spectral functions

$$h_s(\theta) = e^{s\theta}, \qquad s = \pm 1, \pm 3, \pm 5, \dots \quad (4.3)$$

This includes the density of momentum ($h_1 - h_{-1}$) and the density of energy ($h_1 + h_{-1}$), as well as higher-spin local conserved densities. The formulae derived in subsections 3.1 and 3.2 immediately give correlation functions for these densities in inhomogeneous, non-stationary situations (generalizing the homogeneous, stationary two-point function formulae found in [13]).

One may also obtain two-point correlation function formulae for other local fields that are not local conserved densities and currents, using the results of subsection 3.4. There exist explicit results for one-point function of certain exponential fields in GGEs, avoiding the complicated LM series, which thus can be used to extract $V^{\mathcal{O}}(\theta)$ as defined in (3.32), (3.33). It was found in [50–52] that

$$\frac{\langle e^{(k+1)g\Phi}\rangle_{[n]}}{\langle e^{kg\Phi}\rangle_{[n]}} = 1 + 4\sin(\pi a(2k+1)) \int \frac{d\theta}{2\pi} e^\theta n(\theta)[e^{-1}]^{\mathrm{dr}}_k(\theta), \quad (4.4)$$

where

$$[e^{-1}]^{\mathrm{dr}}_k(\theta) = e^{-\theta} + \int \frac{d\alpha}{2\pi} \chi_k(\theta, \alpha)n(\alpha)[e^{-1}]^{\mathrm{dr}}_k(\alpha) \quad (4.5)$$

is (in our interpretation) the $k$-dressing of the function $e^{-1}(\theta) = e^{-\theta}$ seen as a vector field: the dressing with respect to a different, $k$-dependent scattering kernel given by

$$\chi_k(\theta, \alpha) = 2\,\mathrm{Im}\left(\frac{e^{2ki\pi a}}{\sinh(\theta - \alpha - i\pi a)}\right). \tag{4.6}$$

Defining

$$H_k = 1 + 4\sin(\pi a(2k+1))\int \frac{d\theta}{2\pi} e^\theta n(\theta)[e^{-1}]^{dr}_k(\theta), \tag{4.7}$$

and using $\langle 1\rangle_{[n]} = 1$, the one-point function can be obtained for all $k \in \mathbb{N}$ as $\langle e^{kg\Phi}\rangle_{[n]} = \prod_{j=0}^{k-1} H_j$. Differentiating $H_k$ with respect to $\beta_i$ can be done using (2.12) and (3.5), giving

$$-\frac{\partial}{\partial \beta_i} H_k = 4\sin(\pi a(2k+1))\int \frac{d\theta}{2\pi} e^{dr}_k(\theta)n(\theta)f(\theta)h^{dr}_i(\theta)[e^{-1}]^{dr}_k(\theta), \tag{4.8}$$

where

$$e^{dr}_k(\theta) = e^\theta + \int \frac{d\alpha}{2\pi} \chi_k(\alpha, \theta)n(\alpha)e^{dr}_k(\alpha) \tag{4.9}$$

is the $k$-dressing of the function $e(\theta) = e^\theta$ seen as a scalar field. Using (3.33), we then find, for all $k \in \mathbb{N}$,

$$V^k(\theta) = \frac{2}{\pi\rho_s(\theta)}\sum_{j=0}^{k-1}\sin(\pi a(2j+1))\,e^{dr}_j(\theta)[e^{-1}]^{dr}_j(\theta)\prod_{\substack{l=0 \\ l\neq j}}^{k-1} H_l. \tag{4.10}$$

By the $\mathbb{Z}_2$ symmetry[6] $\Phi \mapsto -\Phi$ we have $V^{-k}(\theta) = V^k(\theta)$. Further [52], there is a symmetry $\langle e^{kg\Phi}\rangle_{[n]} = \langle e^{(k+a^{-1})g\Phi}\rangle_{[n]}$, and thus $V^k(\theta) = V^{k+a^{-1}}(\theta)$, which, for irrational couplings $a$, allows us to reach arbitrary values of $k \in \mathbb{R}$. The resulting $V^k(\theta)$ can be inserted into (3.35) and (3.36) in order to get Euler-scale two-point correlation functions of fields $e^{kg\Phi}$ and $e^{k'g\Phi}$ for any $k, k'$.

The generalized hydrodynamics of classical limit of the sinh-Gordon model was investigated in [16], where the classical limit of $V^k(\theta)$ was derived. Euler-scale two-point functions obtained by (3.36) were verified to agree with direct numerical simulations.

### 4.1.2 Lieb-Liniger model

The repulsive Lieb-Liniger model is defined (for mass equal to $1/2$) by the second-quantized Hamiltonian

$$H = \int dx\left(\partial_x\Psi^\dagger\partial_x\Psi(x) + c\Psi^\dagger\Psi^\dagger\Psi\Psi\right) \tag{4.11}$$

for a single complex bosonic field $\Psi$, where $c > 0$ is a coupling parameter. It is Galilean invariant, and its TBA description contains a single quasi-particle type, so we take $\mathcal{S} = \mathbb{R}$ and write $\boldsymbol{\theta} = \theta$. One may choose the parametrization given by the momentum, $\theta = p \in \mathbb{R}$ (so that $p'(\theta) = 1$ and $\rho_s = 1^{dr}/(2\pi)$). There are various TBA descriptions possible, but in one convenient description, the quasi-particle is of Fermionic type, hence $f(p) = 1 - n(p)$. In this description, the differential scattering phase is given by

$$\varphi(p) = \frac{2c}{p^2 + c^2}. \tag{4.12}$$

---

[6]We exclude states which are not $\mathbb{Z}_2$ symmetric, as they require an extension of the present formalism.

Again, as a set of natural local fields, one may consider the local conserved densities and currents of the model; they correspond to the spectral functions

$$h_r(p) = p^{r-1}, \qquad r = 1, 2, 3, \ldots \tag{4.13}$$

This includes the density of particles ($r = 1$), the density of momentum ($r = 2$) and the density of energy ($r = 3$). Again, the formulae derived in subsections 3.1 and 3.2 give correlation functions for these densities in inhomogeneous, non-stationary situations.

One may also obtain two-point correlation formulae for other local fields that are not local conserved densities and currents, using the results of subsection 3.4. Consider the $K^{\text{th}}$ power of the particle density,

$$\mathcal{O}_K = \frac{1}{(K!)^2} (\Psi^\dagger)^K (\Psi)^K. \tag{4.14}$$

It was shown in [49] that in a homogeneous state characterized by the occupation function $n(p)$, its average takes the form

$$\langle \mathcal{O}_K \rangle_{[n]} = \int_{\mathbb{R}^K} \left( \prod_{r=1}^{K} \frac{\mathrm{d}p_1}{2\pi} n(p_r) h_r^{\text{dr}}(p_r) \right) \prod_{j \geq l} \frac{p_j - p_l}{(p_j - p_l)^2 + c^2}. \tag{4.15}$$

Taking the $\beta_i$-derivative is simple, by using (2.12) and the general formula (3.5):

$$-\frac{\partial}{\partial \beta_i} \langle \mathcal{O}_K \rangle_{[n]} = \int_{\mathbb{R}^K} \left( \prod_{r=1}^{K} \frac{\mathrm{d}p_r}{2\pi} n(p_r) h_r^{\text{dr}}(p_r) \right) \sum_{s=1}^{K} g_s^{\text{dr}}(p_s) f(p_s) h_i^{\text{dr}}(p_s), \tag{4.16}$$

where

$$g_s(p_s) = \prod_{j \geq l} \frac{p_j - p_l}{(p_j - p_l)^2 + c^2} \tag{4.17}$$

is defined as a function of $p_s$ with $p_{r \neq s}$ fixed parameters. From this we identify

$$V^{\mathcal{O}_K}(p) = \sum_{s=1}^{K} \int_{\mathbb{R}^{K-1}} \left( \prod_{\substack{r=1 \\ r \neq s}}^{K} \frac{\mathrm{d}p_r}{2\pi} n(p_r) h_r^{\text{dr}}(p_r) \right) \frac{h_s^{\text{dr}}(p) g_s^{\text{dr}}(p)}{1^{\text{dr}}(p)}, \tag{4.18}$$

where $1^{\text{dr}}(p)$ is the dressed constant function 1. This gives two-point functions by insertion in (3.35) and, in the homogeneous case, in (3.36).

We note finally that in the very recent paper [91] new expressions for expectation values of the fields $\mathcal{O}_K$ are obtained using the non-relativistic limit of the sinh-Gordon model and the results of [50–52] recalled above. These appear to be more efficient. By using the methods shown here, this can in turn be used to obtain different expressions for $V^{\mathcal{O}_K}(\theta)$. This is worked out in [92].

## 4.2 Free particle models

In free particle models, formulae (3.16) - (3.19) simplify. Using (3.14), the fact that the dressing operator is trivial, and $n_t(x; \theta) = n_0(x - v^{\text{gr}}(\theta)t, \theta)$, one obtains the simple expression

$$
\begin{aligned}
\langle \mathsf{q}_i(x,t) \mathsf{q}_j(y,0) \rangle_{[n_0]}^{\text{Eul}} &= \int_{\mathcal{S}} \mathrm{d}\theta \, \rho_{\mathrm{p}}(x,t;\theta) f(x,t;\theta) h_i(\theta) h_j(\theta) \delta(x - y - v^{\text{gr}}(\theta)t) \\
&= \sum_{\theta \in (v^{\text{gr}})^{-1}(\frac{x-y}{t})} \frac{\rho_{\mathrm{p}}(y,0;\theta) f(y,0;\theta)}{|(v^{\text{gr}})'(\theta)| \, t} h_i(\theta) h_j(\theta). \tag{4.19}
\end{aligned}
$$

Here $(v^{\mathrm{gr}})^{-1}(\xi) = \{\boldsymbol{\theta} : v^{\mathrm{gr}}(\boldsymbol{\theta}) = \xi\}$. The integral form has the clear physical interpretation of a correlation coming from the ballistically propagating particles on the ray connecting the two fields. It is evaluated by summing over all solutions to $v^{\mathrm{gr}}(\boldsymbol{\theta}) = (x - y)/t$, of which there is at most one for every particle type. One similarly obtains current correlations by multiplying by factors of $v^{\mathrm{gr}}(\boldsymbol{\theta})$.

For instance, in the quantum Ising model (a free Majorana fermion), where there is a single particle type, one has $p(\theta) = m\sinh\theta$, $E(\theta) = m\cosh\theta$, $v(\theta) = \tanh\theta$ and $f(y, 0; \theta) = 1 - 2\pi\rho_{\mathrm{p}}(y, 0; \theta)/(m\cosh\theta)$. If the initial state is locally thermal with local inverse temperature $\beta(y)$, then

$$2\pi\rho_{\mathrm{p}}(y, 0; \theta) = \frac{m\cosh\theta}{1 + \exp[-\beta(y)m\cosh\theta]}. \tag{4.20}$$

In this case, the energy density dynamical two-point function (writing $\mathfrak{q}_1 = T^{00}$, the time-time component of the stress-energy tensor) is zero outside the lightcone, and otherwise is

$$\langle T^{00}(x, t)T^{00}(y, 0)\rangle^{\mathrm{Eul}}_{[n_0]} \tag{4.21}$$

$$= \frac{m^3\cosh^5\theta}{2\pi t\,(1 + \exp[-\beta(y)m\cosh\theta])(1 + \exp[\beta(y)m\cosh\theta])}\bigg|_{\theta = \mathrm{arctanh}\,((x-y)/t)}$$

$$= \frac{m^3 t^4}{8\pi s^5\cosh^2\left(\frac{\beta(y)mt}{2s}\right)} \qquad \text{(Ising model)},$$

where $s = \sqrt{t^2 - (x - y)^2}$ is the relativistic time-like distance between the fields.

Similarly, consider the correlation function of particle densities (writing the density as $\mathfrak{q}_0 = \mathfrak{n}$) in the free nonrelativistic, spinless fermion, evolved from a state with space-dependent temperature $\beta(y)$ and chemical potential $\mu(y)$. For instance, this describes the Tonks-Girardeau limit of the Lieb-Liniger model. We find

$$\langle \mathfrak{n}(x, t)\mathfrak{n}(y, 0)\rangle^{\mathrm{Eul}}_{[n_0]} = \frac{m}{8\pi t\cosh^2\left(\frac{\beta(y)}{2}\left(\frac{m(x-y)^2}{2t^2} - \mu(y)\right)\right)} \qquad \text{(Tonks-Girardeau)}. \tag{4.22}$$

In free particle models, it is possible to develop the full program outlined in Subsection 3.1, and to obtain explicit expressions for every $N$-point correlation functions, at least for conserved densities. The procedure is quite straightforward, and the results can be expressed as follows. Let $w(x; \boldsymbol{\theta}) = \sum_{j=0}^{\infty}\beta_j(x)h_j(\boldsymbol{\theta})$ be the TBA driving term of the GGEs (2.17). Then, for $N = 2, 3, 4, \ldots$ we have

$$\left\langle\prod_{k=1}^{N}\mathfrak{q}_{i_k}(x_k, t_k)\right\rangle^{\mathrm{Eul}}_{[n_0]} \tag{4.23}$$

$$= \sum_{\boldsymbol{\theta}\in(v^{\mathrm{gr}})^{-1}\left(\frac{x_2-x_1}{t_2-t_1}\right)}\frac{p'(\boldsymbol{\theta})(t_2 - t_1)^{-1}}{2\pi|(v^{\mathrm{gr}})'(\boldsymbol{\theta})|}g_N\left(\frac{x_1t_2 - x_2t_1}{t_2 - t_1}; \boldsymbol{\theta}\right)\times$$

$$\times\prod_{k=3}^{N}\delta\left(\frac{(x_k - x_1)(t_2 - t_1) - (x_2 - x_1)(t_k - t_1)}{t_2 - t_1}\right)\prod_{k=1}^{N}h_{i_k}(\boldsymbol{\theta}).$$

Here the functions $g_N(y; \boldsymbol{\theta})$ are defined via generating functions as

$$\mathsf{F}_a(w(y; \boldsymbol{\theta})) - \mathsf{F}_a(w(y; \boldsymbol{\theta}) - z) = \sum_{N=1}^{\infty}\frac{z^N}{N!}g_N(y; \boldsymbol{\theta}), \tag{4.24}$$

where $a$ is the quasi-particle type associated to $\boldsymbol{\theta} = (\theta, a)$, and where the free energy function $F_a(w)$ is given in (2.7). In (4.23), the delta-functions on the right-hand side constrain the equalities $(x_k - x_1)/(t_k - t_1) = (x_2 - x_1)/(t_2 - t_1)$ (for all $k$), and thus $(x_k - x_j)/(t_k - t_j) = (x_2 - x_1)/(t_2 - t_1)$ (for all $j \neq k$), which equal $v^{\mathrm{gr}}(\boldsymbol{\theta})$. Therefore, we may replace the argument $(x_1 t_2 - x_2 t_1)/(t_2 - t_1)$ of the function $g_N(\cdot; \boldsymbol{\theta})$ by $(x_j t_k - x_k t_j)/(t_k - t_j)$ or by $x_k - v^{\mathrm{gr}}(\boldsymbol{\theta}) t_k$ for any $j \neq k$.

In fact, all Euler-scale correlation functions, for $N = 1, 2, 3, 4, \ldots$, can be obtained formally by using generating functionals over the generating parameters $\varepsilon_k(x)$ via

$$
\left\langle \exp\left[\sum_k \int_{\mathbb{R}} \mathrm{d}x \, \varepsilon_k(x) \mathfrak{q}_{i_k}(x, t_k)\right] - 1 \right\rangle_{[n_0]}^{\mathrm{Eul}} \tag{4.25}
$$
$$
= \int_{\mathcal{S}} \frac{\mathrm{d}\boldsymbol{\theta}}{2\pi} p'(\boldsymbol{\theta}) \int_{\mathbb{R}} \mathrm{d}u \left( F_a(w(u; \boldsymbol{\theta})) - F_a\left(w(u; \boldsymbol{\theta}) - \sum_k \varepsilon_k(u + v^{\mathrm{gr}}(\boldsymbol{\theta}) t_k) h_{i_k}(\boldsymbol{\theta})\right)\right),
$$

where on the right-hand side $\boldsymbol{\theta} = (\theta, a)$.

Conjecturally, correlation functions involving currents are obtained by replacing factors $h_{i_k}(\boldsymbol{\theta})$ by $v^{\mathrm{gr}}(\boldsymbol{\theta}) h_{i_k}(\boldsymbol{\theta})$.

We note that (4.19) and (4.23) give general, explicit expressions for Euler-scale $N$-point correlation functions of conserved densities in free theories. There is no integral over spectral parameters: for every particle type, there is a single velocity that contributes to the connected Euler-scale correlation, which is the velocity of the particle propagating from the initial to the final point. For similar reasons, correlation functions for $N \geq 3$ have a delta-function structure which imposes colinearity of all space-time positions. Connected Euler-scale correlations can only arise from single quasi-particles travelling through each of the space-time points. Due to this, all correlation functions depend on the initial state only through the local state at a single position: the position, at time 0, crossed by the single ray passing through all space-time points (this is $y$ in (4.19) and more generally $x_k - v^{\mathrm{gr}}(\boldsymbol{\theta}) t_k$ in (4.23)). Therefore, the only effect of the weak inhomogeneity is to give a dependence on the state via this single position. All these properties are expected to be broken in inhomogeneous states of interacting models. The dependence is not solely on the state at a single position, as the knowledge of the state at other positions is necessary in order to evaluate the effect of a disturbance on the quasi-particle trajectories (hence to evaluate the response function). Similarly, we do not expect a delta-function structure for higher-point functions in interacting models.

## 5 Discussion and analysis

### 5.1 Interpretation of the general formulae

Formulae (3.16)-(3.19) can be given a relatively clear interpretation. A correlation function is expressed as an integral over all spectral parameters of the product of the quantity of charge of the first observable, $h_i^{\mathrm{dr}}(x, t, \boldsymbol{\theta})$, carried by the spectral parameter $\boldsymbol{\theta}$ and dressed with respect to the the local bath $n_t(x)$, times the propagation to the point $(x, t)$, of the quantity of charge of the second observable $h_j^{\mathrm{dr}}(0, y; \boldsymbol{\alpha})$ dressed by the local bath $n_0(y)$. The propagation factor is $\Gamma_{(y,0) \to (x,t)}(\boldsymbol{\theta}, \boldsymbol{\alpha}) \rho_{\mathrm{p}}(x, t; \boldsymbol{\theta}) f(x, t; \boldsymbol{\theta})$. It includes the propagator itself $\Gamma_{(y,0) \to (x,t)}(\boldsymbol{\theta}, \boldsymbol{\alpha})$, representing the effect of quasi-particles ballistically propagating from $(y, 0)$ to $(x, t)$, as well as the density $\rho_{\mathrm{p}}(x, t; \boldsymbol{\theta})$, which weigh this effect with the quantity of quasi-particles actually propagating. It also includes the factor $f(x, t; \boldsymbol{\theta})$, which modulates the weight according the quasi-particle statistics; for instance, for fermions, this factor forbids the entry of a new quasi-particles if the local occupation function is saturated to $n_t(x; \boldsymbol{\theta}) = 1$, making the correlation effect of this quasi-particle vanish. The same structure occurs for more general local

observables in (3.40), where the only complication is in the dressing by the local bath, which involves a sum over all form factors. The nontrivial physics of the propagation of correlations – the response of the operator at $(x, t)$ to a disturbance by the observable at $(y, 0)$ – is fully encoded within the propagator.

There is an apparent asymmetry between the initial position $(y, 0)$ and the final position $(x, t)$, as the factors $\rho_p(x, t; \boldsymbol{\theta}) f(x, t; \boldsymbol{\theta})$ are only present for the latter position. However we note that the points $(y, 0)$ and $(x, t)$ are not independent, they are related by the evolution equation (2.22): thus every quasi-particle at $(x, t)$ has an antecedent at $(y, 0)$. In nontrivial (inhomogeneous, interacting) cases, asymmetry is also explicit in the equation defining the propagator (3.12), where quantities pertaining to the initial state density appear naturally. In these cases, the propagator is not an intrinsic, state-independent property of quasi-particle propagation: it is affected by the initial state in nontrivial ways. It is possible to write the two-point functions in more symmetric ways, such as in (3.26), but the choice (3.7) for the propagator has the advantage that (i) it specializes to delta-functions in simple cases (3.13), (3.14), (3.15), and (ii) its defining integral equation (3.12) only involves quantities that are explicitly non-divergent in any GGE.

The propagator is composed of two elements, as per (3.20). The first, the direct propagator, comes from the direct propagation of the disturbance of the initial state due to the observable at $(y, 0)$. At the Euler scale, this travels with quasi-particles along their characteristics described by the function $u(x, t; \boldsymbol{\theta})$. Only particles with just the right spectral parameter will travel from $(y, 0)$ to $(x, t)$, and thus this element should indeed give a delta-function contribution to the propagator.

The second element, the indirect propagator $\Delta_{(y,0)\to(x,t)}(\boldsymbol{\theta}, \boldsymbol{\alpha})$, is more subtle. It comes from the *change of trajectories* of quasi-particles due to the disturbance at $(y, 0)$. In the explicit calculation in Appendix B.1, it is seen as the change of the characteristics $u(x, t; \boldsymbol{\theta})$ upon differentiation with respect to the Lagrange parameter $\beta_j(0)$. The indirect propagator $\Delta_{(y,0)\to(x,t)}(\boldsymbol{\theta}, \boldsymbol{\alpha})$ is still applied to the local dressed quantity $h_j^{dr}(0, y; \boldsymbol{\alpha})$, as it is this quantity that is to travel on the slightly modified trajectory in order to create correlations. However all spectral parameters $\boldsymbol{\alpha}$ may generically participate, instead of a single one, because all are involved in determining the trajectory.

As has been noted above, the indirect propagator vanishes in homogeneous states and in free-particle models (see (3.13) and (3.14)). The above interpretation makes these fact clear: in homogeneous states, it does not matter if the quasi-particle trajectories are modified, as the state is everywhere the same; and in free models, the trajectories do not depend on the local states, thus are not affected by the disturbance at $(y, 0)$.

One can see that the indirect propagator is largely controlled by the effective acceleration $a_{[n_0]}^{eff}(u(x, t; \boldsymbol{\theta}); \boldsymbol{\theta})$, and in particular that it vanishes if the latter does. Recall that the effective acceleration was initially introduced in order to describe force terms due to external, space-dependent fields (that is, weakly inhomogeneous evolution Hamiltonians) [12]. Here, it instead encodes the (weak) spatial inhomogeneity of the initial state. The space-dependent GGEs in the fluid cells of the initial fluid state are associated with an inhomogeneous "Hamiltonian" $\sum_i \beta_i(x) Q_i$, and it would be an evolution with respect to this that would generate force terms controlled by the acceleration field $a_{[n_0]}^{eff}(z; \boldsymbol{\theta})$. Here we see that the effective acceleration instead determines, in part, the way in which characteristics are modified due to disturbances.

It is in principle possible to numerically evaluate the expressions (3.16)-(3.19). Recall that the exact solution (2.23) can be solved very efficiently by iteration, as explained in [33]. Therefore, we may assume $n_0(z; \boldsymbol{\theta})$, $n_t(z; \boldsymbol{\theta})$ and $u(z, t; \boldsymbol{\theta})$ to be readily numerically available for all $z, \boldsymbol{\theta}$. It is then straightforward to evaluate dressed quantities, which can be done by solving (2.3) either by iteration, or by discretizing the linear integral equation and inverting

the resulting matrix $1 - Tn$. Therefore, the only ingredient in (3.16)-(3.19) that is not readily numerically available from previous works is the propagator $\Gamma_{(y,0)\to(x,t)}(\boldsymbol{\theta}, \boldsymbol{\alpha})$. For this, we write it in the form (3.20). We can then evaluate the indirect propagator by solving (3.21), where the function $g(\boldsymbol{\theta})$ is chosen as either $h_j^{\mathrm{dr}}(y, 0; \boldsymbol{\theta})$ or $v^{\mathrm{eff}}(y, 0; \boldsymbol{\theta})h_j^{\mathrm{dr}}(y, 0; \boldsymbol{\theta})$ depending on the correlator sought for. The source term can be evaluated from the quantities already numerically available, and (3.21) is a linear integral equation which can be solved, for instance, by iterations. One difficulty might lie in the evaluation of derivatives, for instance $u'(x, t; \boldsymbol{\gamma})$. One might find it more efficient to differentiate the integral equation (2.23) and solve for $u'(x, t; \boldsymbol{\gamma})$ instead of directly taking the derivative numerically.

## 5.2 Partitioning protocol (domain wall initial condition)

Consider the evolution from an initial density operator

$$\exp\left[ -\int_{-\infty}^{0} \mathrm{d}x \sum_i \beta_i^L \mathsf{q}_i(x) - \int_{0}^{\infty} \mathrm{d}x \sum_i \beta_i^R \mathsf{q}_i(x) \right],$$

where the state is spatially separated between two different homogeneous states on the left and right. This is referred to as the partitioning or cut-and-glue protocol, or as the evolution with domain wall initial condition, and has been studied extensively, see the review [72]. Even though the initial condition is not smooth, as the initial generalized temperatures display an abrupt jump at the origin, profiles quickly smooth out and the fluid approximation is very accurate after a small relaxation time. The GHD solution [7, 8], obtained with initial fluid state of the form

$$n_0(x; \boldsymbol{\theta}) = n_{\mathrm{L}}(\boldsymbol{\theta})\Theta(-x) + n_{\mathrm{R}}(\boldsymbol{\theta})\Theta(x), \tag{5.1}$$

gives extremely accurate predictions, as verified in the XXZ model [8] and in the hard rod gas [29]. The solution is a set of ray dependent states

$$n_t(x; \boldsymbol{\theta}) = n(\xi; \boldsymbol{\theta}) = n_{\mathrm{L}}(\boldsymbol{\theta})\Theta(v^{\mathrm{eff}}(\xi; \boldsymbol{\theta}) - \xi) + n_{\mathrm{R}}(\boldsymbol{\theta})\Theta(\xi - v^{\mathrm{eff}}(\xi; \boldsymbol{\theta})), \tag{5.2}$$

where $\xi = x/t$. Below we will denote

$$\langle \mathcal{O}(x, t)\mathcal{O}'(y, 0)\rangle_{[n_{\mathrm{L}}, n_{\mathrm{R}}]}^{\mathrm{Eul}}$$

scaled correlation functions in this protocol.

Let us analyze certain correlation functions in this setup. Naively, one might think that scaled two-point correlation functions (3.16)-(3.19) for two fields lying on the same ray should equal those in the homogeneous state of this ray, as obtained using (3.13): correlations should be carried by particles traveling along this ray alone. This is however incorrect. In order to see this, we consider two different situations. For simplicity we concentrate on charge-charge correlations (3.16), but a similar analysis holds in other cases. See Appendix D for a study of the characteristics in the partitioning protocol.

### 5.2.1 Correlations on a ray away from connection time

Consider the initial domain-wall state to be at time $-t_0 < 0$ (so $n_0$ is the fluid state after the evolution by $t_0$ from the domain wall), and let $(x, t)$ and $(y, 0)$ lie on the same ray emanating from $(-t_0, 0)$, that is $\xi = x/(t + t_0) = y/t_0$. Then the fluid state is the same at $(x, t)$ and at $(y, 0)$. Thus we have

$$\langle \mathsf{q}_i(x, t)\mathsf{q}_j(y, 0)\rangle_{[n_{\mathrm{L}}, n_{\mathrm{R}}]}^{\mathrm{Eul}} = \int_{\mathcal{S}} \mathrm{d}\boldsymbol{\theta} \int_{\mathcal{S}} \mathrm{d}\boldsymbol{\alpha}\, \Gamma_{(y,0)\to(x,t)}(\boldsymbol{\theta}, \boldsymbol{\alpha})\rho_{\mathrm{p}}(\xi; \boldsymbol{\theta})f(\xi; \boldsymbol{\theta})h_i^{\mathrm{dr}}(\xi; \boldsymbol{\theta})h_j^{\mathrm{dr}}(\xi; \boldsymbol{\alpha}).$$
$$\tag{5.3}$$

First let us look at the contribution from the direct propagator $\delta(y - u(x, t; \boldsymbol{\theta}))\delta_S(\boldsymbol{\alpha} - \boldsymbol{\theta})$ (see (3.20)). For $(x, t)$ and $(y, 0)$ on the same ray, the only solutions $\boldsymbol{\theta}$ to $y = u(x, t; \boldsymbol{\theta})$ are the solutions to $y = x - v^{\text{eff}}(\boldsymbol{\theta})t$. However, we cannot replace $\delta(y - u(x, t; \boldsymbol{\theta}))$ by $\delta(x - y - v^{\text{eff}}(\xi; \boldsymbol{\theta})t)$, as would be required to reproduce the homogeneous correlators according to (3.13). Indeed, the variation, with respect to $\theta$, of $u(x, t; \boldsymbol{\theta})$ is not the same as that of $v^{\text{eff}}(\xi; \boldsymbol{\theta})t$, due to the second equation in (D.14). Instead, the direct-propagator contribution to the two-point function is

$$\int_S \mathrm{d}\boldsymbol{\theta} \, \frac{\delta(x - y - v^{\text{eff}}(\xi; \boldsymbol{\theta})t)}{V(\boldsymbol{\theta})} \rho_{\text{p}}(\xi; \boldsymbol{\theta}) f(\xi; \boldsymbol{\theta}) h_i^{\text{dr}}(\xi; \boldsymbol{\theta}) h_j^{\text{dr}}(\xi; \boldsymbol{\theta}), \tag{5.4}$$

where $V(\boldsymbol{\theta})$ is defined in (D.15).

The contribution from the indirect propagator $\Delta_{(y,0)\to(x,t)}(\boldsymbol{\alpha}; \boldsymbol{\theta})$ gives an additional correction. This contribution is generically nonzero, in particular the state at $t = 0$ is not homogeneous and thus the effective acceleration $a_{[n_0]}^{\text{eff}}(x; \boldsymbol{\theta})$ is nonzero.

Therefore, as compared to the homogeneous correlator obtained using (3.13) in (3.16), there are two corrections: the factor $1/V(\boldsymbol{\theta})$ in the direct-propagator contribution (5.4), and the indirect-propagator contribution. We have not shown that these two corrections don't cancel each other, but this seems unlikely. Both corrections are due to the fact that the insertion of an observable in a correlation function perturbs the state as seen by other observables, and that due to the nonlinearity of GHD, this perturbation generically affects the trajectories of quasi-particles. Thus other rays are explored, and the two-point function is not that in the homogeneous state of a single ray.

### 5.2.2 Correlations with one observable at connection time

Second, consider the initial domain wall to be at $t = 0$. In this case, the state is locally homogeneous at $(y, 0)$ for any $y \in \mathbb{R} \setminus \{0\}$, therefore $a_{[n_0]}^{\text{eff}}(y; \boldsymbol{\theta}) = 0$. As a consequence only the direct propagator remains,

$$\Gamma_{(y,0)\to(x,t)}(\boldsymbol{\theta}, \boldsymbol{\alpha}) = \delta(y - u(x, t; \boldsymbol{\theta}))\delta_S(\boldsymbol{\alpha} - \boldsymbol{\theta}). \tag{5.5}$$

The expression for the scaled two-point function simplifies to a finite sum, as per (3.25). Then we have, for any $x, t, y$ and with $\xi = x/t$,

$$\langle \mathfrak{q}_i(x, t)\mathfrak{q}_j(y, 0)\rangle_{n_L, n_R}^{\text{Eul}} = \frac{1}{t} \sum_{\gamma \in \theta_\star(x, t; y)} \frac{\rho_{\text{p}}(\xi; \gamma) f(\xi; \gamma)}{|\partial_\gamma \tilde{u}(\xi; \gamma)|} h_i^{\text{dr}}(\xi; \gamma) h_j^{\text{dr}}(\text{sgn}(y) \infty; \gamma), \tag{5.6}$$

where $\tilde{u}(\xi; \boldsymbol{\theta}) = u(x, t; \boldsymbol{\theta})/t$ (see Appendix D). Taking $y \to 0^\pm$, we can use again (D.14) and the argument above to obtain

$$\lim_{y \to 0^\pm} \langle \mathfrak{q}_i(x, t)\mathfrak{q}_j(y, 0)\rangle_{[n_L, n_R]}^{\text{Eul}} = \int_S \mathrm{d}\boldsymbol{\theta} \, \frac{\delta(x - v^{\text{eff}}(\xi; \boldsymbol{\theta})t)}{V(\boldsymbol{\theta})} \rho_{\text{p}}(\xi; \boldsymbol{\theta}) f(\xi; \boldsymbol{\theta}) h_i^{\text{dr}}(\xi; \boldsymbol{\theta}) h_j^{\text{dr}}(\pm\infty; \boldsymbol{\theta}). \tag{5.7}$$

This again looks very similar to the two-point function in a homogeneous state, except for two differences: the factor $V(\boldsymbol{\theta})$, and the fact that the state at $(y = 0^\pm, 0)$ is not equal to that on the ray $\xi$ that emanates from the origin: it is instead the initial condition, equal to the state at $\xi = \pm\infty$. Thus, again, the two-point function on a ray is not that in the homogeneous state of that ray.

The question of the two-point function with $y = 0$, that is, with one observable within the original discontinuity, is more subtle and answered below.

## 5.3 Long-time asymptotics

Consider an initial state $n_0(x; \boldsymbol{\theta})$. Suppose it has well-defined asymptotic behavior at large distances, where it becomes homogeneous:

$$\lim_{x \to \pm\infty} n_0(x; \boldsymbol{\theta}) = n_0^{\pm}(\boldsymbol{\theta}). \tag{5.8}$$

In particular, we suppose that $x_0$ can be set to $-\infty$ in (2.23). Suppose also that the asymptotic is uniform enough, so that the following integrals converge absolutely:

$$\int_x^{\infty} dz \left( \rho_s(z, t; \boldsymbol{\theta}) - \rho_s^+(\boldsymbol{\theta}) \right) < \infty, \quad \int_{-\infty}^x dz \left( \rho_s(z, t; \boldsymbol{\theta}) - \rho_s^-(\boldsymbol{\theta}) \right) < \infty \qquad \forall\, x \in \mathbb{R} \tag{5.9}$$

(here and below we denote by $\rho_s^{\pm}(\boldsymbol{\theta}) = \lim_{x \to \pm\infty} \rho_s(x, 0; \boldsymbol{\theta})$ the asymptotic forms of the initial state density). For instance, the initial state could be a state that varies nontrivially only on some finite region. Consider the long-time limit $t \to \infty$ of scaled two-point functions (3.16)-(3.19), along rays $x = \xi t$ with $y, \xi$ fixed. By a simple scaling argument, they should decay like $1/t$. We provide a derivation of the coefficient of this decay:

$$\langle \mathsf{q}_i(\xi t, t) \mathsf{q}_j(y, 0) \rangle_{[n_0]}^{\mathrm{Eul}} \sim \frac{A_{ij}(\xi; y)}{t} \qquad (t \to \infty). \tag{5.10}$$

Again we concentrate on the charge-charge two-point function as the derivation and result is easy to generalize to the currents.

In order to derive this result, we further assume that in the limit $t \to \infty$ along any ray $x = \xi t$, one obtains the state $n(\xi; \boldsymbol{\theta})$, given by (5.2), of the partitioning protocol with initial condition specified by $n_0^{\pm}(\boldsymbol{\theta})$:

$$\lim_{t \to \infty} n_t(\xi t; \boldsymbol{\theta}) = n(\xi; \boldsymbol{\theta}), \qquad n(\xi; \boldsymbol{\theta}) \text{ from the partitioning protocol with } n_{\mathrm{R,L}}(\boldsymbol{\theta}) = n_0^{\pm}(\boldsymbol{\theta}). \tag{5.11}$$

We provide in Appendix E a proof under certain more basic assumptions of uniform convergence. Here and below, for lightness of notation, we take the convention that GHD functions explicitly evaluated on a ray, say $\xi$, instead of a space-time doublet $(x, t)$, are understood as the functions obtained in this limit, for instance $\rho_s(\xi; \boldsymbol{\theta}) = \lim_{t \to \infty} \rho_s(\xi t, t; \boldsymbol{\theta})$. These are set by the solution to the partitioning protocol (5.2), see also Appendix D. From this viewpoint, we note that the exact initial condition $n_0(x; \boldsymbol{\theta})$ provides a regularization of the initial discontinuity at $x = 0$ of the partitioning protocol.

An important observation of the result below is that $A_{ij}(\xi; y)$ is *not* determined solely by the partitioning protocol; in particular it is not the coefficient obtained in either of the two situations studied in Subsection 5.2, and it depends on the point $y$ and on the details of $n_0(x; \boldsymbol{\theta})$. What this means is that, from the viewpoint of the partitioning protocol, correlation functions on a single ray, with one space-time point being at the initial time $t = 0$ and lying on the initial discontinuity of the protocol, explicitly depend on the regularization $n_0(x; \boldsymbol{\theta})$ of this initial discontinuity, and on the exact position $y$, within the regularized region, of the observable at initial time.

Consider the following quantity, which encodes the difference between the regularized initial condition $n_0(x; \boldsymbol{\theta})$ and the discontinuous one determined by $n_{\mathrm{R,L}}(\boldsymbol{\theta})$. Given a point $y \in \mathbb{R}$, a ray $\xi$ and a time $t$, we look for the spectral parameters $\boldsymbol{\theta}$ of quasi-particles starting at $y$ that reaches the position $\xi t$ at time $t$, under the full initial condition $n_0(x; \boldsymbol{\theta})$. If the effective velocity is monotonic with respect to the rapidity, then thanks to (2.28), this is unique once the quasi-particle type is determined. In general, we simply consider the set of such $\boldsymbol{\theta}$. We then look for the position $r$ of a quasi-particle $\boldsymbol{\theta}$ that would reach the same point $(xt, t)$, but in the partitioning protocol. See Fig. 1. Finally we take the limit $t \to \infty$ of this position. In

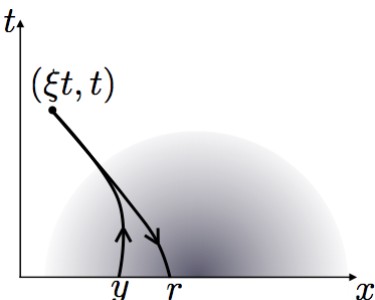

Figure 1: A pictorial representation of how to evaluate the quantity $r$, given $\xi, t, y$. Start at the point $y$, and find the quasi-particle's rapidity which is such that its trajectory joins $y$ with $(\xi t, t)$, in the full problem with initial condition $n_0(x; \boldsymbol{\theta})$. Then, using the same quasi-particle type and rapidity, evaluate the backward trajectory from $(\xi t, t)$ in the partitioning protocol. The value of $r$ is the position obtained at time 0. In this picture, the shade indicates the space-time region where the fluid states in the full problem and in the partitioning protocol are substantially different, thus affecting the trajectories.

this limit, $\boldsymbol{\theta} \in \boldsymbol{\theta}_\star(\xi)$ (that is $v^{\text{eff}}(\xi, \boldsymbol{\theta}) = \xi$). Given this value of $\boldsymbol{\theta}$, the ray $\xi$ is known uniquely (see Appendix D), and thus it fully encodes the ray $\xi$. The result is $r(y; \boldsymbol{\theta})$, which depends on both $y$ and on this limiting value of $\boldsymbol{\theta}$. This is defined for all values of $\boldsymbol{\theta}$ (there is always a solution to $v^{\text{eff}}(\xi, \boldsymbol{\theta}) = \xi$). In formulae, this is expressed as follows in terms of the function $\tilde{u}(\xi; \boldsymbol{\theta})$ of the partitioning protocol, which has the explicit form (D.9). We define $\boldsymbol{\theta}_t$ (whose depence on $\xi, y$ we keep implicit) as $u(\xi t, t; \boldsymbol{\theta}_t) = y$, and then $r(y; \boldsymbol{\theta}_\infty) = \lim_{t \to \infty} t\tilde{u}(\xi; \boldsymbol{\theta}_t)$ with $\boldsymbol{\theta}_\infty = \lim_{t \to \infty} \boldsymbol{\theta}_t \in \boldsymbol{\theta}_\star(\xi)$.

The above defines $r(y; \boldsymbol{\theta})$ in a very delicate way, that involves the full time evolution: one needs to evaluate the finite difference between the end-points of two trajectories that start far in time and stay near to each other for a long time. In order to go further, we need to make certain assumptions about $r(y; \boldsymbol{\theta})$, which appear to be natural but which we do not know how to verify explicitly. The main assumption is simply that $r(y; \boldsymbol{\theta})$ is finite. This seems natural if the space-time region where the effects of the regularized partitioning is felt, is of finite extent, as pictorially suggested in Fig. 1. Other more subtle assumptions relate to the exchange of $y$-derivative and large-time limit, see Appendix E.2.

Under these assumptions, we show in Appendix E.2 that the following integral equation holds:

$$\int_{-\infty}^{\xi_\star(\boldsymbol{\theta})} d\eta \sum_{\gamma \in \boldsymbol{\theta}_\star(\eta)} \frac{\rho_s(\eta; \gamma) T^{\text{dr}}(\eta; \boldsymbol{\theta}, \gamma)}{V(\gamma)|(v^{\text{eff}})'(\eta; \gamma)|} \int_{\mathbb{R}} dz \left| \frac{\partial r(z, \gamma)}{\partial z} \right| \left( n_0(z; \gamma) - n_0^{\text{sgn}(r(z; \gamma))} \right)$$
$$= \int_{-\infty}^{r(y; \boldsymbol{\theta})} dz \left( \rho_s(z, 0; \boldsymbol{\theta}) - \rho_s^{\text{sgn}(z)}(\boldsymbol{\theta}) \right) + \int_{r(y; \boldsymbol{\theta})}^{y} dz \, \rho_s(z, 0; \boldsymbol{\theta}).$$

$$(5.12)$$

Recall the dressed scattering operator (3.23). Equation (5.12) is a powerful result, as it determines $r(y; \boldsymbol{\theta})$ entirely in terms of initial data, without the need for time evolution. Even more powerful is the fact that, although the left-hand side depends on $\boldsymbol{\theta}$, it is independent of $y$. Thus, by uniform convergence (5.9), we must have

$$r(y; \boldsymbol{\theta}) \sim y \qquad (|y| \to \infty). \qquad (5.13)$$

This is simply saying that for $y$ far from the regularization region, there is no difference with

the partitioning protocol. Further, differentiating with respect to $y$, we obtain

$$\frac{\partial r(y;\boldsymbol{\theta})}{\partial y} = \frac{\rho_s(y,0;\boldsymbol{\theta})}{\rho_s^{\text{sgn}(r(y;\boldsymbol{\theta}))}(\boldsymbol{\theta})} > 0, \tag{5.14}$$

thus $r(y;\boldsymbol{\theta})$ is monotonic in $y$. This means that $r(y;\boldsymbol{\theta})$ has a unique zero $y_\star(\boldsymbol{\theta})$, which determines its sign:

$$r(y_\star(\boldsymbol{\theta});\boldsymbol{\theta}) = 0, \qquad r(y;\boldsymbol{\theta}) \gtrless 0 \quad \text{if} \quad y \gtrless y_\star(\boldsymbol{\theta}). \tag{5.15}$$

It is this zero that plays a fundamental role for the long-time asymptotics of correlation functions. Consider the sign function

$$\sigma(y;\boldsymbol{\theta}) = \text{sgn}(y - y_\star(\boldsymbol{\theta})) = \text{sgn}(r(y;\boldsymbol{\theta})). \tag{5.16}$$

An equation determining this zero is inferred from (5.12):

$$\int_{-\infty}^{\xi_\star(\boldsymbol{\theta})} d\eta \sum_{\boldsymbol{\gamma}\in\boldsymbol{\theta}_\star(\eta)} \frac{\rho_s(\eta;\boldsymbol{\gamma})\varphi^{\text{dr}}(\eta;\boldsymbol{\theta},\boldsymbol{\gamma})}{V(\boldsymbol{\gamma})|(v^{\text{eff}})'(\eta;\boldsymbol{\gamma})|} \int_{\mathbb{R}} dz \frac{\rho_s(z,0;\boldsymbol{\gamma})}{\rho_s^{\sigma(z;\boldsymbol{\gamma})}(\boldsymbol{\gamma})} \left(n_0(z;\boldsymbol{\gamma}) - n_0^{\sigma(z;\boldsymbol{\gamma})}\right)$$
$$= \int_{-\infty}^{0} dz \left(\rho_s(z,0;\boldsymbol{\theta}) - \rho_s^{-}(\boldsymbol{\theta})\right) + \int_{0}^{y_\star(\boldsymbol{\theta})} dz \, \rho_s(z,0;\boldsymbol{\theta}). \tag{5.17}$$

The function $r(y;\boldsymbol{\theta})$ then takes the simple form

$$r(y;\boldsymbol{\theta}) = \frac{1}{\rho_s^{\pm}(\boldsymbol{\theta})} \int_{y_\star(\boldsymbol{\theta})}^{y} dz \, \rho_s(z,0;\boldsymbol{\theta}) \qquad \text{for } y \gtrless y_\star(\boldsymbol{\theta}). \tag{5.18}$$

We show in Appendix E.3 that:

$$A_{ij}(\xi;y) = \sum_{\boldsymbol{\gamma}\in\boldsymbol{\theta}_\star(\xi)} \frac{\rho_s(\xi;\boldsymbol{\gamma})h_i^{\text{dr}}(\xi;\boldsymbol{\gamma})}{\rho_s^{\sigma(y;\boldsymbol{\gamma})}(\boldsymbol{\gamma})V(\boldsymbol{\gamma})|(v^{\text{eff}})'(\xi;\boldsymbol{\gamma})|} \left[\rho_p(y,0;\boldsymbol{\gamma})f(y,0;\boldsymbol{\gamma})h_j^{\text{dr}}(y,0;\boldsymbol{\gamma})\right.$$
$$\left. + \left(n_0(y;\boldsymbol{\gamma}) - n_0^{\sigma(y;\boldsymbol{\gamma})}(\boldsymbol{\gamma})\right)\left(\rho_s(y,0)f(y,0)h_j^{\text{dr}}(y,0)\right)^{*\text{dr}}(y,0;\boldsymbol{\gamma})\right]. \tag{5.19}$$

This provides the long-time asymptotic coefficient explicitly in terms of initial data. In the special case where both sides have the same asymptotics,

$$n_0^{+}(\boldsymbol{\theta}) = n_0^{-}(\boldsymbol{\theta}) = n(\boldsymbol{\theta}), \tag{5.20}$$

the partitioning protocol is homogeneous, and the formula simplifies to:

$$A_{ij}(\xi;y) = \sum_{\boldsymbol{\gamma}\in\boldsymbol{\theta}_\star(\xi)} \frac{h_i^{\text{dr}}(\boldsymbol{\gamma})}{|(v^{\text{eff}})'(\boldsymbol{\gamma})|} \left[\rho_p(y,0;\boldsymbol{\gamma})f(y,0;\boldsymbol{\gamma})h_j^{\text{dr}}(y,0;\boldsymbol{\gamma})\right.$$
$$\left. + \left(n_0(y;\boldsymbol{\gamma}) - n(\boldsymbol{\gamma})\right)\left(\rho_s(y,0)f(y,0)h_j^{\text{dr}}(y,0)\right)^{*\text{dr}}(y,0;\boldsymbol{\gamma})\right], \tag{5.21}$$

where GHD quantities depending only on the spectral variable are to be evaluated in the asymptotic GGE state $n(\boldsymbol{\theta})$. Here we have used the fact that $V(\boldsymbol{\theta}) = 1$ in the homogeneous case, as $v^{\text{eff}}(\boldsymbol{\theta}) = \xi_\star(\boldsymbol{\theta})$ (see (D.15)).

As a check, we can verify that the limit $|y| \to \infty$ of (5.21) gives the two-point correlation function in the homogeneous case (3.16) with (3.13) (whose full dependence on time is in the factor $t^{-1}$):

$$\lim_{|y| \to \infty} \frac{A_{ij}(\xi; y)}{t} = \langle \mathtt{q}_i(\xi t, t) \mathtt{q}_j(0, 0) \rangle^{\mathrm{Eul}}_{[n]} \qquad \text{(same left and right asymptotics).} \tag{5.22}$$

Indeed, this follows from (using (5.8)):

$$\lim_{|y| \to \infty} n_0(y; \boldsymbol{\theta}), \rho_{\mathrm{s}}(y, 0; \boldsymbol{\theta}), h_j^{\mathrm{dr}}(y, 0; \boldsymbol{\theta}) = n(\boldsymbol{\theta}), \rho_{\mathrm{s}}(\boldsymbol{\theta}), h_j^{\mathrm{dr}}(\boldsymbol{\theta}). \tag{5.23}$$

We see that the homogeneous correlation function is at the point $y = 0$: this is natural, as on the left-hand side, the limit $t \to \infty$, $x = \xi t$ is taken before $|y| \to \infty$.

We can similarly check that the limit $y \to \infty$ of (5.19) gives the the two-point correlation function

$$\lim_{y \to \pm\infty} \frac{A_{ij}(\xi; y)}{t} = \lim_{y \to 0^{\pm}} \langle \mathtt{q}_i(\xi t, t) \mathtt{q}_j(y, 0) \rangle^{\mathrm{Eul}}_{[n_0^+, n_0^-]} \qquad \text{(different left and right asymptotics)} \tag{5.24}$$

in the partitioning protocol, see (5.7). We therefore find the natural result that the limit $|y| \to \infty$ of the regularized partitioning protocol, where $y$ starts within the inhomogeneous region that regularizes the discontinuity and goes away from it, exactly agrees with the limit $y \to 0^{\pm}$ of the exact partitioning protocol, where $y$ starts within the homogeneous region and goes towards the discontinuity.

**Remark.** We note a somewhat surprising result that is derived in Appendix E.3, and that leads to the particular form of the results expressed above. It can be expressed equivalently as a "sum rule"

$$\int_{-\infty}^{\infty} \mathrm{d}z \, \frac{\rho_{\mathrm{p}}(z, 0; \boldsymbol{\theta}) f(z, 0; \boldsymbol{\theta})}{\rho_{\mathrm{s}}^{\sigma(z; \boldsymbol{\theta})}(\boldsymbol{\theta})} a_{[n_0]}^{\mathrm{eff}}(z; \boldsymbol{\theta}) = 0, \tag{5.25}$$

or as an "occupation equipartition" relation,

$$n_0(y_{\star}(\boldsymbol{\theta}); \boldsymbol{\theta}) = \frac{n_0^-(\boldsymbol{\theta}) \rho_{\mathrm{s}}^+(\boldsymbol{\theta}) - n_0^+(\boldsymbol{\theta}) \rho_{\mathrm{s}}^-(\boldsymbol{\theta})}{\rho_{\mathrm{s}}^+(\boldsymbol{\theta}) - \rho_{\mathrm{s}}^-(\boldsymbol{\theta})}. \tag{5.26}$$

The latter relation is extremely nontrivial, as it relates the zero $y_{\star}(\boldsymbol{\theta})$ to state properties at the asymptotics and at the point $y_{\star}(\boldsymbol{\theta})$ only, while (5.17) defines the zero in terms of states at other points as well. Also, it is not *a priori* obvious that (5.26) has a solution at all. The derivation we provide in Appendices E.2 and E.3 imply that there is at least one solution. We believe this is deeply related to the requirements of finiteness of $r(y; \boldsymbol{\theta})$ and the possibility of exchanging $y$-derivative and large-$t$ limit.

## 6 Conclusion

In this paper we have obtained exact expressions for dynamical connected correlation functions at the Euler scale in non-equilibrium integrable models. These represent correlations obtained under unitary time evolution from inhomogeneous density matrices in quantum models, or deterministic evolution from random initial configurations in classical models. The time evolution is taken to be the homogeneous evolution of the integrable model (and thus this excludes the cases of evolutions in external potentials). The results are expressed solely in terms of quantities that are available within the thermodynamic Bethe ansatz framework. They are

valid at the Euler scale, where variations of averages of local fields occur on very large scales. Interestingly, this shows that hydrodynamic ideas provide, in principle, all large-scale correlation functions. The range of applicability of our results is the same as that of GHD, and thus includes a wide variety of integrable models. Our derivation is based on a natural Euler-scale fluctuation-dissipation principle combined with the exact GHD general solution found in [33]. We showed that our results agree with the general principles of the hydrodynamic projection theory, with in particular the hydrodynamic operators found in [13].

We also showed how two-point functions of arbitrary observables can be obtained from the knowledge of their one-point functions using the hydrodynamic projection theory. From the Leclair-Mussardo formula, valid for one-point functions, we therefore obtained Euler-scale two-point functions as infinite form factor series. This formula is new both in the inhomogeneous case, and in homogeneous GGEs. We also remark that recently, an exact recursion relation was obtained for expectation values of vertex operators of the form $e^{a\phi}$ in the sinh-Gordon model [50–52]. This can be used to extract some of the spectral function $V^{e^{a\phi}}$, and deduce their Euler-scale two-point functions using (3.35).

The general Euler-scale hydrodynamic argument presented in this paper supports the assumption that $N$-point correlation functions vanish, under scaling by $\lambda$, as $\lambda^{1-N}$, as per the formula (1.5). In particular, two-point functions vanish as $1/t$ at large times. This is clear in various formulae established, for instance in the homogeneous case (3.36), in free models (4.19), in the partitioning protocol (5.6), and in the long-time asymptotics (5.10). However, in these formulae, the quantity $|\partial_\theta v^{\mathrm{eff}}(\theta)|$ appears in denominators, evaluated in particular states and at particular values of $\theta$ (for instance, $|\partial_\theta v^{\mathrm{eff}}(\xi, \theta)|$ in the state at ray $\xi$ of the partitioning protocol, evaluated at $\theta$ such that $v^{\mathrm{eff}}(\xi; \theta) = \xi$). This quantity may vanish if the effective velocity is not strictly monotonic with respect to the rapidity, and may thus lead to singularities (except, in some cases, if there is zero density of quasi-particles at this rapidity, or, in fermionic systems, of quasi-holes). In such situations, the asymptotic formulae we show do not apply, and we expect a modification of the large-time limit, naively as $1/\sqrt{t}$. The physical intuition is that, if there is a finite quasi-particle density at a rapidity for which $\partial_\theta v^{\mathrm{eff}}(\theta) = 0$, then, as the effective velocity is stationary in $\theta$, there is an accumulation of quasi-particles around this effective velocity. If, for instance, the observation ray in the partitioning protocol is along this velocity, then this accumulation may increase the correlation. This might happen at the boundary of the "light cone" emanating from the connection point, if there is a maximal velocity. Similar effects might appear in fully inhomogeneous situations if the rapidity derivative of the characteristic function $u(x, t; \theta)$ vanishes, as again singularities may occur for instance in (3.22) and (3.25). It would be interesting to further investigate this aspect.

Comparing exact hydrodynamic predictions for two-point functions with numerics is a very important problem. Steps forwards are made in this direction in [16], where the classical sinh-Gordon model is studied, both for one-point functions in the partitioning protocol and correlation functions in GGEs. In particular, the classical spectral functions for an infinite family of vertex operators are evaluated, and numerical comparisons are made. Comparison with quantum field theories are however more challenging.

It would be interesting to investigate if hydrodynamic ideas provide more than the Euler-scale part of correlation functions, the least decaying part found along ballistic rays. For instance, the recent works [76–79] suggest that it is possible to combine hydrodynamics with a more detailed knowledge of local observables in order to go further.

Other ways of deriving Euler-scale correlation functions in homogeneous cases are based on form factors. This was done in [69] based on form factors obtained in [93]. The form factor techniques of [53, 56] might also be applicable as explained in the Remark in Subection 3.4. We note that using the general results of [58] for space-like two-point functions in arbitrary homogeneous GGEs, one may combine this with the spectral function method of subsection

3.4 in order to get various configurations of dynamical higher-point functions. It would be interesting to see if form factors can be used to derive results in the inhomogeneous situations considered here. It would also be interesting to obtain correlations in situations with evolution in weakly varying external potentials or temperature fields. The GHD theory for such situations was developed in [12], however the equivalent of the solution by characteristics (2.23) has not yet been written. We leave this for future works.

## Acknowledgments

I am grateful to Alvise Bastianello, Olalla Castro Alvaredo, Jacopo De Nardis, Jérôme Dubail, Balázs Pozsgay, Herbert Spohn, Gerard Watts and Takato Yoshimura for discussions, comments and encouragement, and for collaborations on related aspects. I am grateful to the Institut d'Étude Scientifique de Cargèse, France and the Perimeter Institute, Waterloo, Canada for hospitality during completion of this work. I thank the Centre for Non-Equilibrium Science (CNES).

## A   Invertibility of the function $u(x, t; \theta)$

Here we show that $u'(x, t; \theta) < 0$ if the effective velocity is monotonic in the rapidity. This is natural: recall that the function $u(x, t; \theta)$ represents the position, at time 0, from where a quasi-particle trajectory of spectral parameter $\theta$ would reach the position $x$ at time $t$. Therefore, since the effective velocity is monotonic with the rapidity of quasi-particles, a positive change of the velocity associated to $\theta$ will occasion a negative change of the initial position of the trajectory that reaches the space-time point $(x, t)$.

More precisely, for the formal proof, we assume that the effective velocity is monotonic $(v^{\text{eff}})'(\theta) > 0$ (this is the case in many situations in QFT for instance, see [7]), and that (2.23) has a solution. We also recall that $u(x, 0; \theta) = x$.

Let $t \mapsto (x(t; \theta), t)$ be a $\theta$-trajectory: $\partial_t x(t; \theta) = v^{\text{eff}}(x(t; \theta), t; \theta)$. Consider $u'(x, t; \theta)$. Differentiating (2.24) with respect to $\theta$, this satisfies the equation

$$\partial_t u'(x, t; \theta) + v^{\text{eff}}(x, t; \theta)\, \partial_x u'(x, t; \theta) = -(v^{\text{eff}})'(x, t; \theta)\, \partial_x u(x, t; \theta). \tag{A.1}$$

Evaluated on the trajectory $x(t; \theta)$, we therefore have

$$\partial_t u'(x(t; \theta), t; \theta) = -(v^{\text{eff}})'(x(t, \theta), t; \theta)\, \partial_x u(x, t; \theta). \tag{A.2}$$

By assumption $(v^{\text{eff}})'(x, t; \theta) > 0$, and (2.27) says that $\partial_x u(x, t; \theta) > 0$. Therefore $\partial_t u'(x(t; \theta), t; \theta) < 0$. Since $u'(x, 0; \theta) = 0$, we conclude that $u'(x, t; \theta) < 0$ for all $x$ and all $t > 0$.

## B   Propagator and derivation of two-point function formulae

### B.1   Main formulae

Here we present the derivation of formulae (3.16) and (3.17) using the technique explained in Subsection 3.1. Formulae (3.18) is obtained by symmetry, and we note that (3.17) and (3.18) agree with hydrodynamic projection principles. Formula (3.19) is then obtained by using hydrodynamic projections as explained in the Subsection 3.3.

We start with the one-point function. We use the general formula (3.5). Differentiating with respect to $\beta_j(y)$, we find

$$\langle \mathsf{q}_i(x,t)\mathsf{q}_j(y,0)\rangle^{\mathrm{Eul}}_{[n_0]} = \int_{\mathcal{S}} \frac{\mathrm{d}\boldsymbol{\theta}}{2\pi} (p')^{\mathrm{dr}}(x,t;\boldsymbol{\theta})\, \eta_j(x,t;y;\boldsymbol{\theta})\, h_i^{\mathrm{dr}}(x,t;\boldsymbol{\theta}), \tag{B.1}$$

where

$$\eta_j(x,t;y;\boldsymbol{\theta}) = -\frac{\delta}{\delta\beta_j(y)} n_t(x;\boldsymbol{\theta}). \tag{B.2}$$

In order to evaluate $\eta_j(x,t;y;\boldsymbol{\theta})$, we use the solution (2.23). Two terms occur: the first is the derivative of $n_0(u;\boldsymbol{\theta})$ with respect to $\beta_j(y)$ at $u$ fixed, the second involves the derivative of $u(x,t;\boldsymbol{\theta})$ with respect to $\beta_j(y)$. Using (2.18), the first term gives

$$\delta\big(y - u(x,t;\boldsymbol{\theta})\big) h_j^{\mathrm{dr}}(y,0,\boldsymbol{\theta})\, n_t(x;\boldsymbol{\theta}) f(x,t;\boldsymbol{\theta}). \tag{B.3}$$

The second term is evaluated using (3.10), giving

$$n_t(x;\boldsymbol{\theta}) f(x,t;\boldsymbol{\theta}) (p')^{\mathrm{dr}}(u(x,t;\boldsymbol{\theta}),0;\boldsymbol{\theta}) a^{\mathrm{eff}}_{[n_0]}(u(x,t;\boldsymbol{\theta});\boldsymbol{\theta}) \left(-\frac{\delta u(x,t;\boldsymbol{\theta})}{\delta\beta_j(y)}\right). \tag{B.4}$$

In this latter expression, the derivative of $u(x,t;\boldsymbol{\theta})$ occurs. This cannot be evaluated explicitly, but we can obtain an integral equation involving it by differentiating the second equation of (2.23):

$$-\frac{\delta u(x,t;\boldsymbol{\theta})}{\delta\beta_j(y)} (p')^{\mathrm{dr}}(u(x,t;\boldsymbol{\theta}),0;\boldsymbol{\theta}) = \int_{x_0}^{u(x,t;\boldsymbol{\theta})} \mathrm{d}z\, \frac{\delta(p')^{\mathrm{dr}}(z,0;\boldsymbol{\theta})}{\delta\beta_j(y)} - \int_{x_0}^{x} \mathrm{d}z\, \frac{\delta(p')^{\mathrm{dr}}(z,t;\boldsymbol{\theta})}{\delta\beta_j(y)}. \tag{B.5}$$

Let $\mu$ be again a generic parameter on which a GGE state $n$ may depend. Then the following general formula holds, for spectral functions $g(\boldsymbol{\theta})$:

$$\partial_\mu g^{\mathrm{dr}} = (1 - Tn)^{-1} T\partial_\mu n(1 - Tn)^{-1} g = \big(T\,\partial_\mu n\, g^{\mathrm{dr}}\big)^{\mathrm{dr}}. \tag{B.6}$$

Therefore, using (2.18),

$$\frac{\delta(p')^{\mathrm{dr}}(z,0)}{\delta\beta_j(y)} = -\delta(y - z)\big(T\, h_j^{\mathrm{dr}}(y,0)\, n_0(y) f(y,0)\, (p')^{\mathrm{dr}}(y,0)\big)^{\mathrm{dr}}(y,0), \tag{B.7}$$

and definition (B.2) implies

$$-\frac{\delta(p')^{\mathrm{dr}}(z,t)}{\delta\beta_j(y)} = \big(T\, \eta_j(z,t;y)\, (p')^{\mathrm{dr}}(z,t)\big)^{\mathrm{dr}}(z,t). \tag{B.8}$$

Combining (B.3), (B.4), (B.5), (B.7) and (B.8), we have

$$\begin{aligned}
\eta_j&(x,t;y;\boldsymbol{\theta}) \\
&= \delta(y - u) h_j^{\mathrm{dr}}(y,0;\boldsymbol{\theta})\, n_t(x;\boldsymbol{\theta}) f(x,t;\boldsymbol{\theta}) + n_t(x;\boldsymbol{\theta}) f(x,t;\boldsymbol{\theta}) a^{\mathrm{eff}}_{[n_0]}(u;\boldsymbol{\theta}) \times \\
&\quad \times \bigg(-\Theta(u - y)\big(T\, h_j^{\mathrm{dr}}(y,0)\, n_0(y) f(y,0)\, (p')^{\mathrm{dr}}(y,0)\big)^{\mathrm{dr}}(y,0;\boldsymbol{\theta}) + \\
&\qquad\quad + \int_{x_0}^{x} \mathrm{d}z\, \big(T\, \eta_j(z,t;y)\, (p')^{\mathrm{dr}}(z,t)\big)^{\mathrm{dr}}(z,t;\boldsymbol{\theta})\bigg),
\end{aligned} \tag{B.9}$$

where $u = u(x, t; \boldsymbol{\theta})$. Replacing $\eta_j(x, t; y; \boldsymbol{\theta}) = n_t(x; \boldsymbol{\theta})f(x, t; \boldsymbol{\theta})\big(\Gamma_{(y,0)\to(x,t)}h_j^{\mathrm{dr}}(y, 0)\big)(\boldsymbol{\theta})$, which follows from the definitions (3.7) and (B.2), along with the chain rule and (2.18), we obtain the defining integral equation (3.12).

Formula (3.17) is obtained in a similar way starting from the one-point function $\langle \mathrm{j}_i \rangle_{[n]}$, giving

$$\langle \mathrm{j}_i(x, t)\mathrm{q}_j(y, 0)\rangle_{[n_0]}^{\mathrm{Eul}} = \int_{\mathcal{S}} \frac{\mathrm{d}\boldsymbol{\theta}}{2\pi} (E')^{\mathrm{dr}}(x, t; \boldsymbol{\theta})\, \eta_j(x, t; y; \boldsymbol{\theta})\, h_i^{\mathrm{dr}}(x, t; \boldsymbol{\theta}). \tag{B.10}$$

## B.2 Indirect propagator

We show the integral equation (3.21) for the indirect propagator $\big(\Delta_{(y,0)\to(x,t)}g\big)(\boldsymbol{\theta})$. According to the second term on the left-hand side of (3.12), we need to evaluate the star-dressing of the function $\rho_{\mathrm{s}}(z, t; \boldsymbol{\theta})f(z, t; \boldsymbol{\theta})\delta(y - u(z, t; \boldsymbol{\theta}))g(\boldsymbol{\theta})$ as a function of $\boldsymbol{\theta}$. For this purpose, we note that the application $Tn_t(z)$ on it, which is required as per definition (3.9), gives

$$\int_{\mathcal{S}} \frac{\mathrm{d}\boldsymbol{\alpha}}{2\pi} \varphi(\boldsymbol{\theta}, \boldsymbol{\alpha})n_t(z; \boldsymbol{\alpha})\rho_{\mathrm{s}}(z, t; \boldsymbol{\alpha})f(z, t; \boldsymbol{\alpha})\delta(y - u(z, t; \boldsymbol{\alpha}))g(\boldsymbol{\alpha})$$

$$= \sum_{\boldsymbol{\alpha} \in \boldsymbol{\theta}_\star(z, t; y)} \frac{\varphi(\boldsymbol{\theta}, \boldsymbol{\alpha})n_t(z; \boldsymbol{\alpha})\rho_{\mathrm{s}}(z, t; \boldsymbol{\alpha})f(z, t; \boldsymbol{\alpha})g(\boldsymbol{\alpha})}{2\pi|u'(z, t; \boldsymbol{\alpha})|}, \quad \text{(B.11)}$$

where the root set $\boldsymbol{\theta}_\star(z, t; y)$ is defined in (3.24). This then needs to be dressed, but the only dependence in $\boldsymbol{\theta}$ is via the differential scattering phase $\varphi(\boldsymbol{\theta}, \boldsymbol{\alpha})$. Consider the dressed scattering operator (3.23). In components, it is

$$T^{\mathrm{dr}}(\boldsymbol{\theta}, \boldsymbol{\alpha}) = T(\boldsymbol{\theta}, \boldsymbol{\alpha}) + \int_{\mathcal{S}} \frac{\mathrm{d}\boldsymbol{\gamma}}{2\pi}\varphi(\boldsymbol{\theta}, \boldsymbol{\gamma})n(\boldsymbol{\gamma})T(\boldsymbol{\gamma}, \boldsymbol{\alpha}) +$$
$$+ \int_{\mathcal{S}} \frac{\mathrm{d}\boldsymbol{\gamma}_1\mathrm{d}\boldsymbol{\gamma}_2}{(2\pi)^2}\varphi(\boldsymbol{\theta}, \boldsymbol{\gamma}_1)n(\boldsymbol{\gamma}_1)\varphi(\boldsymbol{\gamma}_1, \boldsymbol{\gamma}_2)n(\boldsymbol{\gamma}_2)T(\boldsymbol{\gamma}_2, \boldsymbol{\alpha}) + \dots \tag{B.12}$$

Combining, we obtain (3.21) with

$$\big(\mathsf{W}_{(y,0)\to(x,t)}g\big)(\boldsymbol{\theta}) = \int_{x_0}^{x} \mathrm{d}z \sum_{\boldsymbol{\gamma} \in \boldsymbol{\theta}_\star(z, t; y)} \frac{\rho_{\mathrm{p}}(z, t; \boldsymbol{\gamma})f(z, t; \boldsymbol{\gamma})}{|u'(z, t; \boldsymbol{\gamma})|}T^{\mathrm{dr}}(z, t; \boldsymbol{\theta}, \boldsymbol{\gamma})g(\boldsymbol{\gamma})$$
$$- \Theta(u(x, t; \boldsymbol{\theta}) - y)\big(\rho_{\mathrm{s}}(y, 0)f(y, 0)g\big)^{*\mathrm{dr}}(y, 0; \boldsymbol{\theta}). \tag{B.13}$$

Using (2.23), this can be simplified slightly to (3.22).

## C Verification of the Leclair-Mussardo spectral function

In this appendix we verify that the Leclair-Mussardo spectral function (3.39) indeed reproduces the conserved-density and conserved-current spectral functions (3.34), when the diagonal matrix elements involved in the Leclair-Mussardo formula are specialized to those of conserved densities and currents. Here for simplicity we specialize to the sinh-Gordon model with unit mass. We refer to the explanations in [43, 44, 94] for the initial studies, and to [7, App D] for the explicit diagonal matrix elements af all conserved densities and currents. The results are

$$F_k^{\mathrm{q}_i}(\theta_1, \dots, \theta_k) = \varphi(\theta_{1,2})\cdots\varphi(\theta_{k-1,k})h_i(\theta_1)\cosh(\theta_k) + \text{permutations}, \tag{C.1}$$

and

$$F_k^{\mathrm{j}_i}(\theta_1, \dots, \theta_k) = \varphi(\theta_{1,2})\cdots\varphi(\theta_{k-1,k})h_i(\theta_1)\sinh(\theta_k) + \text{permutations}, \tag{C.2}$$

where $\theta_{j,k} = \theta_j - \theta_k$. We need to evaluate

$$\sum_{k=0}^{\infty} \frac{1}{k!} \int_{\mathbb{R}^k} \prod_{j=1}^{k} \left( \frac{d\theta_j}{2\pi} n(\theta_j) \right) (2\pi\rho_s(\theta))^{-1} F_{k+1}^{q_i}(\theta_1, \dots, \theta_k, \theta) \tag{C.3}$$

and similarly for $j_i$. Explicitly, we have

$$\sum_{k=0}^{\infty} \frac{1}{k!} \int_{\mathbb{R}^k} \prod_{j=1}^{k} \left( \frac{d\theta_j}{2\pi} n(\theta_j) \right) F_{k+1}^{q_i}(\theta_1, \dots, \theta_k, \theta)$$

$$= h_i(\theta)\cosh(\theta) +$$

$$+ \sum_{k=1}^{\infty} \int_{\mathbb{R}^k} \prod_{j=1}^{k} \left( \frac{d\theta_j}{2\pi} n(\theta_j) \right) \varphi(\theta_{1,2}) \cdots \varphi(\theta_{k-1,k}) \varphi(\theta_k - \theta) \big( h_i(\theta_1)\cosh(\theta) + \cosh(\theta_1)h_i(\theta) \big) +$$

$$+ \sum_{k=2}^{\infty} \int_{\mathbb{R}^k} \prod_{j=1}^{k} \left( \frac{d\theta_j}{2\pi} n(\theta_j) \right) \sum_{\ell=1}^{k-1} \varphi(\theta_{1,2}) \cdots \varphi(\theta_\ell - \theta)\varphi(\theta - \theta_{\ell+1}) \cdots \varphi(\theta_{k-1,k}) h_i(\theta_1)\cosh(\theta_k).$$

Identifying the *-dressing operation, this is

$$\sum_{k=0}^{\infty} \frac{1}{k!} \int_{\mathbb{R}^k} \prod_{j=1}^{k} \left( \frac{d\theta_j}{2\pi} n(\theta_j) \right) F_{k+1}^{q_i}(\theta_1, \dots, \theta_k, \theta)$$

$$= h_i(\theta)\cosh(\theta) + h_i^{*dr}(\theta)\cosh(\theta) + h_i(\theta)\cosh^{*dr}(\theta) + h_i^{*dr}(\theta)\cosh^{*dr}(\theta)$$

$$= h_i^{dr}(\theta)\cosh^{dr}(\theta). \tag{C.4}$$

Therefore, using $\cosh^{dr}(\theta) = 2\pi\rho_s(\theta)$, we find that (C.3) reproduces the first equation of (3.34). A similar calculation reproduces the second.

# D  The characteristics in the partitioning protocol

Recall the partitioning protocol, and in particular (5.1) and (5.2). In this case, the solution (5.2) is obtained without the use of characteristics (2.23). Nevertheless, it is useful, when analyzing correlations, to have an understanding of the function $u(x, t; \boldsymbol{\theta})$.

First, we show that we can re-write

$$u(x, t; \boldsymbol{\theta}) = t\tilde{u}(\xi; \boldsymbol{\theta}), \tag{D.1}$$

where the function $\tilde{u}$ depends on $x, t$ only through the ratio $\xi = x/t$. Indeed, thanks to the exact solution (5.2) it is clear that the effective velocity $v^{eff}(\xi; \boldsymbol{\theta})$ likewise only depends on $\xi$. With the change of variable, the differential equation in (2.24) leads to

$$(\xi - v^{eff}(\xi; \boldsymbol{\theta}))\partial_\xi \tilde{u} = \tilde{u}. \tag{D.2}$$

Further, the initial condition becomes the asymptotic condition $\lim_{t\to 0}(t/x)\tilde{u} = 1$,

$$\tilde{u} \sim \xi \qquad (\xi \to \pm\infty). \tag{D.3}$$

Since the equation and asymptotic condition only involve the variable $\xi$, the solution likewise only do.

Next, we may solve exactly this equation. Integrating over $\xi$, we obtain

$$\tilde{u}(\xi; \boldsymbol{\theta}) = \tilde{u}(\xi_0; \boldsymbol{\theta}) \exp \int_{\xi_0}^{\xi} \frac{d\eta}{\eta - v^{eff}(\eta; \boldsymbol{\theta})}. \tag{D.4}$$

This can be re-written, for any $A(\boldsymbol{\theta})$ and $B(\boldsymbol{\theta})$, as

$$\tilde{u}(\xi;\boldsymbol{\theta}) = \tilde{u}(\xi_0;\boldsymbol{\theta}) \left( \frac{\xi - B(\boldsymbol{\theta})}{\xi_0 - B(\boldsymbol{\theta})} \right)^{A(\boldsymbol{\theta})} \exp \int_{\xi_0}^{\xi} \mathrm{d}\eta \left[ \frac{1}{\eta - v^{\mathrm{eff}}(\eta;\boldsymbol{\theta})} - \frac{A(\boldsymbol{\theta})}{\eta - B(\boldsymbol{\theta})} \right]. \tag{D.5}$$

The solution of course does not depend on $\xi_0$. We may therefore take the limit $\xi_0 \to -\infty$ on the right-hand side. Choosing $A(\boldsymbol{\theta}) = 1$, we may use (D.3) as well as the fact that $\lim_{\eta \to -\infty} v^{\mathrm{eff}}(\eta;\boldsymbol{\theta}) = v_{\mathrm{L}}^{\mathrm{eff}}(\boldsymbol{\theta})$ (that is, the limit is finite) in order to see that the right-hand side has a limit that gives

$$\tilde{u}(\xi;\boldsymbol{\theta}) = (\xi - B(\boldsymbol{\theta})) \exp \int_{-\infty}^{\xi} \mathrm{d}\eta \left[ \frac{1}{\eta - v^{\mathrm{eff}}(\eta;\boldsymbol{\theta})} - \frac{1}{\eta - B(\boldsymbol{\theta})} \right]. \tag{D.6}$$

Since $u(x,t;\boldsymbol{\theta})$ is strictly increasing with $x$ (see (2.27)), it has at most a single zero as function of $x$. Comparing the exact solution (5.2), (5.1) with the first equation in (2.23), we find that this zero must be at the solution to the equation $x/t = v^{\mathrm{eff}}(x/t;\boldsymbol{\theta})$. This has the simple physical interpretation that the quasi-particle whose trajectory reaches the point $(x,t)$ at an effective velocity equal to the ray $\xi$, is the one that goes along the ray $\xi$ and thus originates from $x = 0$ at $t = 0$. Let us define the function $\xi_\star(\boldsymbol{\theta})$ by this solution

$$\xi_\star(\boldsymbol{\theta}) \;:\; v^{\mathrm{eff}}(\xi_\star(\boldsymbol{\theta});\boldsymbol{\theta}) = \xi_\star(\boldsymbol{\theta}). \tag{D.7}$$

Therefore

$$\tilde{u}(\xi;\boldsymbol{\theta}) = 0 \;\Leftrightarrow\; \xi = \xi_\star(\boldsymbol{\theta}). \tag{D.8}$$

Choosing $B(\boldsymbol{\theta}) = \xi_\star(\boldsymbol{\theta})$, we then have

$$\tilde{u}(\xi;\boldsymbol{\theta}) = (\xi - \xi_\star(\boldsymbol{\theta})) \exp \int_{-\infty}^{\xi} \mathrm{d}\eta \left[ \frac{1}{\eta - v^{\mathrm{eff}}(\eta;\boldsymbol{\theta})} - \frac{1}{\eta - \xi_\star(\boldsymbol{\theta})} \right]. \tag{D.9}$$

This gives a convenient explicit form of the function $\tilde{u}(\xi;\boldsymbol{\theta})$.

Note that the integrand on the right-hand side of (D.9) is composed of two terms both of which have a unique pole at the same position. Since $\partial_\xi \tilde{u}(\xi;\boldsymbol{\theta})$ exists and is (finite and) nonzero by (2.27), then $\tilde{u}(\xi;\boldsymbol{\theta})$ must have a simple zero at $\xi = \xi_\star(\boldsymbol{\theta})$. This implies that the poles of the two terms in the integrand cancel each other. Therefore

$$\frac{\partial}{\partial \eta}(\eta - v^{\mathrm{eff}}(\eta;\boldsymbol{\theta}))\Big|_{\eta=\xi_\star(\boldsymbol{\theta})} = \frac{\partial}{\partial \eta}(\eta - \xi_\star(\boldsymbol{\theta}))\Big|_{\eta=\xi_\star(\boldsymbol{\theta})} = 1. \tag{D.10}$$

This implies

$$\frac{\partial v^{\mathrm{eff}}(\eta;\boldsymbol{\theta})}{\partial \eta}\Bigg|_{\eta=\xi_\star(\boldsymbol{\theta})} = 0. \tag{D.11}$$

Taking the $\theta$-derivative of (D.7), we conclude that

$$\frac{\mathrm{d}\xi_\star(\boldsymbol{\theta})}{\mathrm{d}\theta} = \frac{\partial v^{\mathrm{eff}}(\eta;\boldsymbol{\theta})}{\partial \theta}\Bigg|_{\eta=\xi_\star(\boldsymbol{\theta})}. \tag{D.12}$$

We thus arrive at the following conclusions:

$$\tilde{u}(\xi;\boldsymbol{\theta}) = 0 \quad \text{iff} \quad v^{\mathrm{eff}}(\xi;\boldsymbol{\theta}) = \xi \tag{D.13}$$

and

$$\frac{1}{V(\boldsymbol{\theta})} \frac{\partial \tilde{u}(\xi;\boldsymbol{\theta})}{\partial \xi} \overset{v^{\mathrm{eff}}(\xi;\boldsymbol{\theta})=\xi}{=} 1, \qquad \frac{1}{V(\boldsymbol{\theta})} \frac{\partial \tilde{u}(\xi;\boldsymbol{\theta})}{\partial \theta} \overset{v^{\mathrm{eff}}(\xi;\boldsymbol{\theta})=\xi}{=} -\frac{\partial v^{\mathrm{eff}}(\xi;\boldsymbol{\theta})}{\partial \theta}, \tag{D.14}$$

where

$$V(\boldsymbol{\theta}) = \exp \int_{-\infty}^{\xi_\star(\boldsymbol{\theta})} \mathrm{d}\eta \left[ \frac{1}{\eta - v^{\mathrm{eff}}(\eta;\boldsymbol{\theta})} - \frac{1}{\eta - \xi_\star(\boldsymbol{\theta})} \right]. \tag{D.15}$$

# E   Long time limit

## E.1   A proof of the emergence of the partitioning solution

We make the assumptions stated in the first paragraph of Subsection 5.3, and only assume, instead of those made in the second paragraph, that the limit $\lim_{t\to\infty} n_t(\xi t; \boldsymbol{\theta})$ exists and is of the form $n(\xi; \boldsymbol{\theta})$. We also assume that the state density is uniformly bounded away from zero and infinity in space time, and we assume that the integral

$$\int_{-\infty}^{\xi} d\zeta \left( \rho_s(\zeta t, t; \boldsymbol{\theta}) - \rho_s(\zeta t, 0; \boldsymbol{\theta}) \right) \tag{E.1}$$

converges to that of the pointwise limit of its integrand at large $t$. We show (5.11) as follows. Consider the integral equation (2.23). Using the stated assumptions, we have that $\lim_{t\to\infty} \rho_s(\zeta t, t; \boldsymbol{\theta})$ is of the form $\rho_s(\zeta; \boldsymbol{\theta})$, and that $\lim_{t\to\infty} \rho_s(\zeta t, 0; \boldsymbol{\theta}) = \rho_s^{\mathrm{sgn}(\zeta)}(\boldsymbol{\theta})$ and we find, to leading order in $t$,

$$t\left( \int_{-\infty}^{\xi} d\zeta \left( \rho_s(\zeta; \boldsymbol{\theta}) - \rho_s^{\mathrm{sgn}(\zeta)}(\boldsymbol{\theta}) \right) - v^{\mathrm{eff}}(-\infty, 0; \boldsymbol{\theta}) \rho_s(-\infty, 0; \boldsymbol{\theta}) \right) = \int_{t\xi}^{u(\xi t, t; \boldsymbol{\theta})} dy\, \rho_s(y, 0; \boldsymbol{\theta}), \tag{E.2}$$

where $\xi = x/t$. Since the state density is (uniformly) positive, in order for the equality to hold $u(\xi t, t; \boldsymbol{\theta})$ must scale proportionally to $t$ at large $t$, except for the possible values of $(\xi, \boldsymbol{\theta})$ where the left-hand side vanishes. Given a $\xi$, only a finite number of values of $\boldsymbol{\theta}$ might make this happen. Since these do not affect spectral integrals, they do not affect the evaluation of the state density from the occupation function. Therefore, using (5.9) we find

$$\left( \int_{-\infty}^{\xi} d\zeta \left( \rho_s(\zeta; \boldsymbol{\theta}) - \rho_s^{\mathrm{sgn}(\zeta)}(\boldsymbol{\theta}) \right) - v^{\mathrm{eff}}(-\infty, 0; \boldsymbol{\theta}) \rho_s(-\infty, 0; \boldsymbol{\theta}) \right) = \int_{\xi}^{\tilde{u}(\xi; \boldsymbol{\theta})} d\zeta\, \rho_s^{\mathrm{sgn}(\zeta)}(\boldsymbol{\theta}), \tag{E.3}$$

where $\tilde{u}(\xi; \boldsymbol{\theta}) = \lim_{t\to\infty} u(\xi t, t; \boldsymbol{\theta})/t$. Clearly, we also have

$$n(\xi; \boldsymbol{\theta}) = n_0^{\mathrm{sgn}(\tilde{u}(\xi; \boldsymbol{\theta}))}(\boldsymbol{\theta}). \tag{E.4}$$

Equations (E.3) and (E.4) are exactly the equations (2.23) for the partitioning protocol.

## E.2   The function $r(y; \theta)$ and its integral equation

Choose $\xi = \xi_\star(\boldsymbol{\theta})$, for spectral parameter $\boldsymbol{\theta} = (\theta, a)$. We assume that $(v^{\mathrm{eff}})'(\xi; \boldsymbol{\theta}) \neq 0$. Define $\boldsymbol{\theta}_t(y; \boldsymbol{\theta}) = (\theta_t(y; \boldsymbol{\theta}), a)$ some element in $\boldsymbol{\theta}_\star(\xi t, t; y)$; if the effective velocity is monotonic with respect to the rapidity, then this is the unique element with particle type $a$; but otherwise it is an element which continuously depends on $t$, and which, given $\boldsymbol{\theta}$, can generically be made unique for large enough $t$ by its large-time limit. That is, we have $u(\xi_\star(\boldsymbol{\theta})t, t; \boldsymbol{\theta}_t(y; \boldsymbol{\theta})) = y$. Taking the large-$t$ limit, we get $\tilde{u}(\xi; \boldsymbol{\theta}_\infty) = 0$, and thus $\boldsymbol{\theta}_\infty = \lim_{t\to\infty} \boldsymbol{\theta}_t(y; \boldsymbol{\theta}) = \boldsymbol{\theta}$, and the choice of $\boldsymbol{\theta}$ determines (generically) the element $\boldsymbol{\theta}_t(t; \boldsymbol{\theta})$. The function $r(y; \boldsymbol{\theta})$ is defined as

$$r(y; \boldsymbol{\theta}) = \lim_{t\to\infty} t\tilde{u}(\xi_\star(\boldsymbol{\theta}); \boldsymbol{\theta}_t(y; \boldsymbol{\theta})). \tag{E.5}$$

We assume that the limit defining $r(y; \boldsymbol{\theta})$ exists and is finite, and that it is differentiable with respect to $y$. Clearly $u'(\xi t, t; \boldsymbol{\theta}_t) = (\partial \theta_t / \partial y)^{-1}$. Since $r(y; \boldsymbol{\theta}_\infty)$ is finite, and since, by the assumption that $(v^{\mathrm{eff}})'(\xi; \boldsymbol{\theta}) \neq 0$, the function $\tilde{u}(\xi; \boldsymbol{\theta})$ has simple zeroes in $\theta$, then $\theta_t$ approaches the zero $\theta_\infty$ with corrections of order $t^{-1}$:

$$\theta_t(y; \boldsymbol{\theta}) = \theta + t^{-1} \left( \tilde{u}'(\xi_\star(\boldsymbol{\theta}); \boldsymbol{\theta}) \right)^{-1} r(y; \boldsymbol{\theta}) + o(t^{-1}). \tag{E.6}$$

Let us assume that the corrections $o(t^{-1})$ are smooth enough in $y$. Then the term displayed gives the correct variation of $\theta_t(y; \boldsymbol{\theta})$ with respect to $y$ at fixed $\boldsymbol{\theta}$ to leading order in $t^{-1}$:

$$\frac{\partial \theta_t(y; \boldsymbol{\theta})}{\partial y} = -t^{-1} \frac{\partial r(y; \boldsymbol{\theta})}{\partial y} \Big/ \left( V(\boldsymbol{\theta})(v^{\text{eff}})'(\xi_\star(\boldsymbol{\theta}); \boldsymbol{\theta}) \right) + o(t^{-1}), \tag{E.7}$$

where we used (D.14).

Now consider the integral equation (2.23). Subtracting that for $t\tilde{u}(\xi; \boldsymbol{\theta}_t)$ from that for $u(\xi t, t; \boldsymbol{\theta}_t)$, and using the fact that the states $n(z, t; \boldsymbol{\theta}_t)$ and $n(z/t; \boldsymbol{\theta}_t)$ have the same asymptotic at large distances, we get

$$\int_{-\infty}^{\xi_\star(\boldsymbol{\theta})t} dz \left( \rho_s(z, t; \boldsymbol{\theta}_t(y; \boldsymbol{\theta})) - \rho_s(z/t; \boldsymbol{\theta}_t(y; \boldsymbol{\theta})) \right)$$
$$= \int_{-\infty}^{t\tilde{u}(\xi; \boldsymbol{\theta})} dz \left( \rho_s(z, 0; \boldsymbol{\theta}_t) - \rho_s^{\text{sgn}(z)}(\boldsymbol{\theta}_t) \right) + \int_{t\tilde{u}(\xi; \boldsymbol{\theta})}^{y} dz\, \rho_s(z, 0; \boldsymbol{\theta}_t). \tag{E.8}$$

We may take the large-$t$ limit. Assuming that the initial state densities are continuous in rapidity, we therefore find (changing integration variable on the left-hand side)

$$\lim_{t \to \infty} t \int_{-\infty}^{\xi_\star(\boldsymbol{\theta})} d\eta \left( \rho_s(\eta t, t; \boldsymbol{\theta}_t(y; \boldsymbol{\theta})) - \rho_s(\eta; \boldsymbol{\theta}_t(y; \boldsymbol{\theta})) \right)$$
$$= \int_{-\infty}^{r(y; \boldsymbol{\theta})} dz \left( \rho_s(z, 0; \boldsymbol{\theta}) - \rho_s^{\text{sgn}(z)}(\boldsymbol{\theta}) \right) + \int_{r(y; \boldsymbol{\theta})}^{y} dz\, \rho_s(z, 0; \boldsymbol{\theta}). \tag{E.9}$$

Here we have been careful not to simply replace, on the left-hand side, the integrand by its limit. The physical meaning of the left-hand side of (E.9) is as follows. First observe that $\int_{-\infty}^{\infty} dz\, \rho_s(z, t; \boldsymbol{\theta})$ depends on $t$ only via the linear dependence $t(v^{\text{eff}}(\infty, 0; \boldsymbol{\theta}) - v^{\text{eff}}(-\infty, 0; \boldsymbol{\theta}))$, thanks to the conservation equation $\partial_t \rho_s(x, t; \boldsymbol{\theta}) + \partial_x(v^{\text{eff}}(x, t; \boldsymbol{\theta})\rho_s(x, t; \boldsymbol{\theta})) = 0$ [7, 8]. This represents the inflow and outflow at the asymptotic boundaries of the system. The same $t$-dependence occur for $\int_{-\infty}^{\infty} dz\, \rho_s(z/t; \boldsymbol{\theta})$, as the partitioning protocol is based on the same states at its asymptotic boundaries. Therefore $\int_{-\infty}^{\infty} dz \left( \rho_s(z, t; \boldsymbol{\theta}) - \rho_s(z/t; \boldsymbol{\theta}) \right)$ does not depend on $t$. Taking $t \to 0$, we see that it is simply equal to the total difference between the initial state density, which effectively regularizes the partitioning protocol, and the partitioned initial state with a discontinuity at the origin; this difference is finite by (5.9). Taking the large-$t$ limit, it is clear that we cannot simply take the limit on the integrand, as we would get zero. As time evolves, the difference between the two initial conditions is redistributed in space: the difference between the integrands goes to zero, but the integrated difference does not. The limit in (5.12) measures how much of the initial state density difference has been transferred to the left of the ray $\xi$. This quantity should become constant in time: we would expect it to be the portion of the initial, finite difference carried to the left of $\xi$ by all quasi-particles of effective velocities allowing them to cross the ray. It is a nontrivial quantity, as it depends on the details of the initial condition $n_0(x; \boldsymbol{\theta})$.

In order to evaluate this quantity, we consider the defining relation (2.6) for the state densities. Since $\rho_s(\eta t, t; \boldsymbol{\theta})$ approaches $\rho_s(\eta; \boldsymbol{\theta})$, we may write

$$\rho_s(\eta t, t; \boldsymbol{\theta}) = \rho_s(\eta; \boldsymbol{\theta}) + \delta\rho_s(\eta, t; \boldsymbol{\theta}). \tag{E.10}$$

We expect $\delta\rho_s(\eta, t; \boldsymbol{\theta})$ to decay proportionally to $t^{-1}$ at large $t$. Consider also

$$n(\eta t, t; \boldsymbol{\theta}) = n(\eta; \boldsymbol{\theta}) + \delta n(\eta, t; \boldsymbol{\theta}). \tag{E.11}$$

Clearly

$$\delta n(\eta, t; \boldsymbol{\theta}) = n_0(u(\eta t, t; \boldsymbol{\theta}); \boldsymbol{\theta}) - n_0^{\mathrm{sgn}(\tilde{u}(\eta; \boldsymbol{\theta}))}, \tag{E.12}$$

and thus this is not small at large $t$. However, since $u(\eta t, t; \boldsymbol{\theta})$ grows linearly with $t$ for generic $\boldsymbol{\theta}$, we find that $\delta n(\eta, t; \boldsymbol{\theta})$ is effectively supported, as a function of $\boldsymbol{\theta}$, on small rapidity intervals of order $t^{-1}$. Thus, under integration with smooth functions of rapidity, its contribution is of order $t^{-1}$. Using (2.6), we therefore find, to leading order at large $t$,

$$\delta \rho_s(\eta, t; \boldsymbol{\theta}) = \int_{\mathcal{S}} \frac{\mathrm{d}\boldsymbol{\alpha}}{2\pi} \varphi(\boldsymbol{\theta}, \boldsymbol{\alpha}) n(\eta; \boldsymbol{\alpha}) \delta \rho_s(\eta, t; \boldsymbol{\alpha}) + \int_{\mathcal{S}} \frac{\mathrm{d}\boldsymbol{\alpha}}{2\pi} \varphi(\boldsymbol{\theta}, \boldsymbol{\alpha}) \delta n(\eta, t; \boldsymbol{\alpha}) \rho_s(\eta; \boldsymbol{\alpha}). \tag{E.13}$$

In order to evaluate the contribution of the second term on the right-hand side, consider changing the variable of integration, concentrating on the region of support of the integrand. This can be achieved by setting $\boldsymbol{\alpha} = \boldsymbol{\theta}_t(z; \boldsymbol{\gamma})$, and integrating over $z$ and summing over $\boldsymbol{\gamma} \in \boldsymbol{\theta}_\star(\eta)$:

$$\int_{\mathcal{S}} \mathrm{d}\boldsymbol{\alpha} = \sum_{\boldsymbol{\gamma} \in \boldsymbol{\theta}_\star(\eta)} \int_{\mathbb{R}} \mathrm{d}z \left| \frac{\partial \boldsymbol{\theta}_t(z; \boldsymbol{\gamma})}{\partial z} \right|. \tag{E.14}$$

This change of variable is permitted in order to evaluate the large-$t$ limit assuming that, at large $t$, the function $\boldsymbol{\theta}_t(z; \boldsymbol{\gamma})$ is monotonic with $z$. Thanks to (E.7), this is the case if $r(z; \boldsymbol{\gamma})$ is itself monotonic. Here we simply assume this is the case, and show that this is a consistent assumption. The factors $\varphi(\boldsymbol{\theta}, \boldsymbol{\alpha})$ and $\rho_s(\eta; \boldsymbol{\alpha})$ are smooth in $\boldsymbol{\alpha}$, and thus can be evaluated at $\boldsymbol{\gamma}$. With (E.7), this gives

$$\begin{aligned}
&\int_{\mathcal{S}} \frac{\mathrm{d}\boldsymbol{\alpha}}{2\pi} \varphi(\boldsymbol{\theta}, \boldsymbol{\alpha}) \delta n(\eta, t; \boldsymbol{\alpha}) \rho_s(\eta; \boldsymbol{\alpha}) \\
&= \quad t^{-1} \sum_{\boldsymbol{\gamma} \in \boldsymbol{\theta}_\star(\eta)} \frac{\varphi(\boldsymbol{\theta}, \boldsymbol{\gamma}) \rho_s(\eta; \boldsymbol{\gamma})}{2\pi V(\boldsymbol{\gamma}) |(v^{\mathrm{eff}})'(\eta; \boldsymbol{\gamma})|} \int_{\mathbb{R}} \mathrm{d}z \left| \frac{\partial r(z; \boldsymbol{\gamma})}{\partial z} \right| \delta n(\eta, t; \boldsymbol{\theta}_t(z; \boldsymbol{\gamma})) + o(t^{-1}).
\end{aligned} \tag{E.15}$$

Now we use (E.12), and we can take the large-$t$ limit:

$$\lim_{t \to \infty} \delta n(\eta, t; \boldsymbol{\theta}_t(z; \boldsymbol{\gamma})) = n_0(z; \boldsymbol{\gamma}) - n_0^{\mathrm{sgn}(r(z; \boldsymbol{\gamma}))}(\boldsymbol{\gamma}), \tag{E.16}$$

where we use continuity in rapidity of $n_0$. Putting together (E.13), (E.15) and (E.16), and using the dressing operation, we identify

$$\delta \rho_s(\eta, t; \boldsymbol{\theta}) = t^{-1} \sum_{\boldsymbol{\gamma} \in \boldsymbol{\theta}_\star(\eta)} \frac{T^{\mathrm{dr}}(\eta; \boldsymbol{\theta}, \boldsymbol{\gamma}) \rho_s(\eta; \boldsymbol{\gamma})}{V(\boldsymbol{\gamma}) |(v^{\mathrm{eff}})'(\eta; \boldsymbol{\gamma})|} \int_{\mathbb{R}} \mathrm{d}z \left| \frac{\partial r(z; \boldsymbol{\gamma})}{\partial z} \right| \left( n_0(z; \boldsymbol{\gamma}) - n_0^{\mathrm{sgn}(r(z; \boldsymbol{\gamma}))}(\boldsymbol{\gamma}) \right), \tag{E.17}$$

where the dressed scattering operator (3.23) is involved, as this is the only $\boldsymbol{\theta}$ dependence in the driving term of (E.13). Combining with (E.9), we may now take the large-$t$ limit, and – assuming that the neglected terms $o(t^{-1})$ indeed don't contribute finitely to the integral – we obtain (5.12).

As explained in the main text, this the implies (5.14), which states monotonicity of $r(y; \boldsymbol{\theta})$; thus the assumption is indeed consistent.

## E.3   Derivation of the main result

There are two contributions to $A_{ij}(\xi; y)$: from the direct and the indirect propagators. Consider first that from the direct propagator, the first term on the right-hand side of (3.25).

We need to evaluate the long-time asymptotic of the spectral derivative of $u(x, t; \boldsymbol{\theta})$. By (D.1), $u(x, t; \boldsymbol{\theta})$ grows with $t$ as

$$u(\xi t, t; \boldsymbol{\theta}) \sim t \tilde{u}(\xi; \boldsymbol{\theta}) \qquad (t \to \infty) \tag{E.18}$$

for any $\boldsymbol{\theta}$ such that $\tilde{u}(\xi; \boldsymbol{\theta}) \neq 0$. This simply means that the point $y = u(x, t; \boldsymbol{\theta})$ from which a quasi-particle reaches $x$ after a long time $t$ is generically very far away from the origin. Thus the spectral derivative should also grow with $t$. However, we cannot simply take the $\theta$ derivative of (E.18) in order to obtain the large-$t$ asymptotic of $u'(\xi t, t; \boldsymbol{\theta})$. This is because the finite correction to (E.18) also has a very large derivative. This finite correction occurs when the spectral parameter $\boldsymbol{\theta}$ is very near to that of a quasi-particle traveling along the ray $\xi$ in the partitioning protocol; that is, very near to satisfying $v^{\mathrm{eff}}(\xi; \boldsymbol{\theta}) = \xi$, equivalently $\tilde{u}(\xi; \boldsymbol{\theta}) = 0$, or according to the notations introduced, $\boldsymbol{\theta} \in \boldsymbol{\theta}_\star(\xi; 0) = \boldsymbol{\theta}_\star(\xi)$. If $\boldsymbol{\theta}$ approaches such a point as time grows, the point $y$ may stay finite. But at long times, a small change of $\theta$ (of order $1/t$) will occasion a large change of $y$ (of order 1), because the trajectory depends on the precise structure of the state in finite regions around the origin, and a finite region is spanned by a small change of $\theta$. This contribution is in fact immediate to evaluate form the result of the Appendix E.2. Indeed, from (E.7) we have

$$u'(\xi t, t; \boldsymbol{\theta}_t(y; \boldsymbol{\theta})) = -t \frac{V(\boldsymbol{\theta})(v^{\mathrm{eff}})'(\xi; \boldsymbol{\theta})}{\partial r(y; \boldsymbol{\theta})/\partial y} + o(t), \tag{E.19}$$

where $\xi = \xi_\star(\boldsymbol{\theta})$.

In order to go further, we need to understand how other functions behave when evaluated at $(\xi t, t; \boldsymbol{\theta}_t(y; \boldsymbol{\theta}))$. Clearly, the occupation function $n_t(\xi t; \boldsymbol{\theta}_t(y, \boldsymbol{\theta}))$ does not tend to its partitioning value $n(\xi; \boldsymbol{\theta}) = n_0^{\mathrm{sgn}(\tilde{u}(\xi; \boldsymbol{\theta}))}(\boldsymbol{\theta})$, but rather equals $n_0(y; \boldsymbol{\theta})$. However, the principle we will use below is that any dressed quantity, $h^{\mathrm{dr}}(\xi t, t; \boldsymbol{\theta}_t(y; \boldsymbol{\theta}))$, tends to its partitioning value $h^{\mathrm{dr}}(\xi; \boldsymbol{\theta})$ at large $t$. This is because dressing involves spectral integrals of $n_t(\xi t; \boldsymbol{\alpha})$, and the set of values of $\boldsymbol{\alpha}$ around $\boldsymbol{\theta}_t(y; \boldsymbol{\theta})$ for which $n_t(\xi t; \boldsymbol{\alpha})$ is significantly different from its partitioning value becomes of measure zero, at large $t$, under the $d\boldsymbol{\alpha}$ measure. This is the same effect as that explained in Appendix E.2, and the above principle was used there for the state density $\rho_s = (p')^{\mathrm{dr}}/(2\pi)$.

Therefore, combining (E.19) with (5.14), the first term in (3.25) gives the following contribution to $A_{ij}(\xi; y)$:

$$\left(A_{ij}(\xi; y)\right)_1 = \sum_{\boldsymbol{\gamma} \in \boldsymbol{\theta}_\star(\xi)} \frac{\rho_s(\xi; \boldsymbol{\gamma}) \rho_p(y, 0; \boldsymbol{\gamma}) f(y, 0; \boldsymbol{\gamma})}{\rho_s^{\sigma(y; \boldsymbol{\gamma})}(\boldsymbol{\gamma}) V(\boldsymbol{\gamma}) |(v^{\mathrm{eff}})'(\xi; \boldsymbol{\gamma})|} h_i^{\mathrm{dr}}(\xi; \boldsymbol{\gamma}) h_j^{\mathrm{dr}}(y, 0; \boldsymbol{\gamma}). \tag{E.20}$$

Remark that this can be seen as coming from the integral

$$\int_S d\boldsymbol{\theta}\, \delta(x - v^{\mathrm{eff}}(\xi; \boldsymbol{\theta}) t) \frac{\rho_s(\xi; \boldsymbol{\theta}) \rho_p(y, 0; \boldsymbol{\theta}) f(y, 0; \boldsymbol{\theta})}{\rho_s^{\sigma(y; \boldsymbol{\theta})}(\boldsymbol{\theta}) V(\boldsymbol{\theta})} h_i^{\mathrm{dr}}(\xi; \boldsymbol{\theta}) h_j^{\mathrm{dr}}(y, 0; \boldsymbol{\theta}).$$

Consider now the contribution from the indirect propagator: the second term in (3.25). In order to have an intuition of its contribution, recall that the effective acceleration $a_{[n_0]}^{\mathrm{eff}}(z; \boldsymbol{\theta})$ is zero, except for $z$ in a region around the origin which can roughly be considered as finite. With an argument similar to that made above and in Appendix E.2, since $u(\xi t, t; \boldsymbol{\theta})$ diverges proportionally to $t$ as per (E.18), it is clear that $a_{[n_0]}^{\mathrm{eff}}(u(\xi t, t; \boldsymbol{\theta}); \boldsymbol{\theta})$ vanishes for almost all values of $\boldsymbol{\theta}$, except for a small region, whose extent decreases as $t^{-1}$, around the zeroes of $\tilde{u}(\xi; \boldsymbol{\theta})$. Therefore, any integral of the form $\int_S d\boldsymbol{\theta}\, a_{[n_0]}^{\mathrm{eff}}(u(\xi t, t; \boldsymbol{\theta}); \boldsymbol{\theta}) g(\boldsymbol{\theta})$, for bounded spectral function $g(\boldsymbol{\theta})$, decreases proportionally to $t^{-1}$ and is supported on the point set $\boldsymbol{\theta}_\star(\xi)$. Thanks to (3.21), the indirect propagator has an overall factor $a_{[n_0]}^{\mathrm{eff}}(u(\xi t, t; \boldsymbol{\theta}); \boldsymbol{\theta})$, wherefore the indirect propagator contribution – the second term in (3.25) – is an integral of the above form. Thus, in order to obtain the leading $t^{-1}$ decay of the correlation function, we only need to keep terms that stay finite at large $t$ on the right-hand side of the integral equation (3.21).

In order to determine the finite contribution on the right-hand side of (3.21), consider first the source term (3.22). It has itself two contributions. For the first, we do the change of variable $z = \eta t$ to write it as

$$t \int_{-\infty}^{\xi} d\eta \sum_{\gamma \in \theta_\star(\eta t, t; y)} \frac{\rho_s(\eta t, t; \gamma) n_0(y; \gamma) f(y, 0; \gamma)}{|u'(\eta t, t; \gamma)|} T^{dr}(\eta t, t; \theta, \gamma) g(\gamma). \tag{E.21}$$

In the integrand, all factors converge at large $t$ except for $u'(\eta t, t; \gamma)$, which diverges linearly as per (E.19). Therefore we can directly use the result (E.20) (with appropriate choice of $h_i^{dr}$ and $h_j^{dr}$) to obtain

$$\int_{-\infty}^{\xi} d\eta \sum_{\gamma \in \theta_\star(\eta)} \frac{\rho_s(\eta; \gamma) \rho_p(y, 0; \gamma) f(y, 0; \gamma)}{\rho_s^{\sigma(y; \gamma)}(\gamma) V(\gamma) |(v^{eff})'(\eta; \gamma)|} T^{dr}(\eta; \theta, \gamma) g(\gamma). \tag{E.22}$$

The second term in the source (3.22) is clearly finite.

Next, consider the second term in (3.21), the integral of a star-dressed quantity involving the indirect propagator itself. From the definition (3.9), a star-dressed quantity can be written as a series of terms each involving at least one spectral integral. Since, as argued above, the indirect propagator has an overall factor $a_{[n_0]}^{eff}(u(\xi t, t; \theta); \theta)$, and spectral integrals involving such factors decrease as $t^{-1}$, we conclude that the second term in (3.21) also decreases as $t^{-1}$. Therefore, for the purpose of evaluating the leading decaying term of the correlation function, we may make the replacement

$$\begin{aligned}
\left(\Delta_{(y,0)\to(x,t)} g\right)(\theta) &\mapsto 2\pi a_{[n_0]}^{eff}(u(\xi t, t; \theta); \theta) \times \\
&\left( \int_{-\infty}^{\xi} d\eta \sum_{\gamma \in \theta_\star(\eta)} \frac{\rho_s(\eta; \gamma) \rho_p(y, 0; \gamma) f(y, 0; \gamma)}{\rho_s^{\sigma(y; \gamma)}(\gamma) V(\gamma) |(v^{eff})'(\eta; \gamma)|} T^{dr}(\eta; \theta, \gamma) g(\gamma) \right. \\
&\left. - \Theta(u(\xi t, t; \theta) - y)\left(\rho_s(y, 0) f(y, 0) g\right)^{*dr}(y, 0; \theta) \right).
\end{aligned} \tag{E.23}$$

In order to evaluate the contribution from the second term on the right-hand side of (3.25), let us examine more precisely how spectral integrals involving the factor

$$a_{[n_0]}^{eff}(u(\xi t, t; \theta); \theta) \Theta(u(\xi t, t; \theta) - y) n(\xi t, t; \theta) f(\xi t, t; \theta)$$

decay at large $t$. We evaluate such integrals by changing variable to $z$ via $\theta = \theta_t(z; \gamma)$ and summing over $\gamma \in \theta_\star(\xi)$, similarly to (E.14). Consider some spectral function $g_t(\theta)$, and assume that it is not only bounded, but also that the limit $\lim_{t\to\infty} g_t(\theta_t(z; \gamma)) = g_\infty(\gamma)$ exists and is independent of $z$. Then,

$$\int_{\mathcal{S}} d\theta \, a_{[n_0]}^{eff}(u(\xi t, t; \theta); \theta) \Theta(u(\xi t, t; \theta) - y) n(\xi t, t; \theta) f(\xi t, t; \theta) g_t(\theta) \tag{E.24}$$

$$= t^{-1} \sum_{\gamma \in \theta_\star(\xi)} \frac{g_\infty(\gamma)}{V(\gamma) |(v^{eff})'(\xi; \gamma)|} \int_y^\infty dz \frac{\partial r(z; \gamma)}{\partial z} a_{[n_0]}^{eff}(z; \gamma) n_0(z; \gamma) f(z, 0; \gamma) + o(t^{-1}).$$

The $z$ integral in the above expression can be performed as follows. Using (5.14) and (3.10), we have

$$\int_y^\infty dz \frac{\partial r(z; \theta)}{\partial z} a_{[n_0]}^{eff}(z; \theta) n_0(z; \theta) f(z, 0; \theta) = \int_y^\infty \frac{dz}{2\pi} \frac{\partial_z n_0(z; \theta)}{\rho_s^{\sigma(z; \theta)}(\theta)}, \tag{E.25}$$

giving the result

$$\frac{1}{2\pi}I(y;\boldsymbol{\theta}), \quad I(y;\boldsymbol{\theta}) = \begin{cases} \dfrac{n_0^+(\boldsymbol{\theta}) - n_\star(\boldsymbol{\theta})}{\rho_s^+(\boldsymbol{\theta})} + \dfrac{n_\star(\boldsymbol{\theta}) - n_0(y;\boldsymbol{\theta})}{\rho_s^-(\boldsymbol{\theta})} & (y_\star(\boldsymbol{\theta}) > y) \\[3mm] \dfrac{n_0^+(\boldsymbol{\theta}) - n_0(y;\boldsymbol{\theta})}{\rho_s^+(\boldsymbol{\theta})} & (y_\star(\boldsymbol{\theta}) < y). \end{cases} \tag{E.26}$$

Here $n_\star(\boldsymbol{\theta}) = n_0(y_\star(\boldsymbol{\theta}); \boldsymbol{\theta})$ and $y_\star(\boldsymbol{\theta})$ is the unique zero of $r(y;\boldsymbol{\theta})$, which satisfies (5.17). We will also denote

$$I(\boldsymbol{\theta}) = \lim_{y \to -\infty} I(y;\boldsymbol{\theta}) = \frac{n_0^+(\boldsymbol{\theta}) - n_\star(\boldsymbol{\theta})}{\rho_s^+(\boldsymbol{\theta})} + \frac{n_\star(\boldsymbol{\theta}) - n_0^-(\boldsymbol{\theta})}{\rho_s^-(\boldsymbol{\theta})}. \tag{E.27}$$

Combining (E.23) with the second term on the right-hand side of (3.25), we obtain the indirect propagator contribution to the long-time asymptotics. There are two terms. The first corresponds to choosing, in (E.24), the value $y = -\infty$, and then the function

$$g_t(\boldsymbol{\theta}) = 2\pi \rho_s(\xi t, t; \boldsymbol{\theta}) h_i^{\mathrm{dr}}(\xi t, t; \boldsymbol{\theta}) \times$$
$$\times \int_{-\infty}^{\xi} \mathrm{d}\eta \sum_{\boldsymbol{\gamma} \in \boldsymbol{\theta}_\star(\eta)} \frac{\rho_s(\eta; \boldsymbol{\gamma}) \rho_p(y, 0; \boldsymbol{\gamma}) f(y, 0; \boldsymbol{\gamma})}{\rho_s^{\sigma(y;\boldsymbol{\gamma})}(\boldsymbol{\gamma}) V(\boldsymbol{\gamma}) |(v^{\mathrm{eff}})'(\eta; \boldsymbol{\gamma})|} T^{\mathrm{dr}}(\eta; \boldsymbol{\theta}, \boldsymbol{\gamma}) h_j^{\mathrm{dr}}(y, 0; \boldsymbol{\gamma}).$$

The other corresponds to keeping $y$, and choosing the function

$$g_t(\boldsymbol{\theta}) = -2\pi \rho_s(\xi t, t; \boldsymbol{\theta}) h_i^{\mathrm{dr}}(\xi t, t; \boldsymbol{\theta}) \big(\rho_s(y, 0) f(y, 0) h_j^{\mathrm{dr}}(y, 0)\big)^{*\mathrm{dr}}(y, 0; \boldsymbol{\theta}).$$

We therefore get the following contribution to $A_{ij}(\xi; y)$:

$$\big(A_{ij}(\xi; y)\big)_2$$
$$= \sum_{\boldsymbol{\gamma} \in \boldsymbol{\theta}_\star(\xi)} \frac{\rho_s(\xi; \boldsymbol{\gamma})}{V(\boldsymbol{\gamma}) |(v^{\mathrm{eff}})'(\xi; \boldsymbol{\gamma})|} |h_i^{\mathrm{dr}}(\xi; \boldsymbol{\gamma}) \times$$
$$\times \bigg( I(\boldsymbol{\gamma}) \int_{-\infty}^{\xi} \mathrm{d}\eta \sum_{\boldsymbol{\alpha} \in \boldsymbol{\theta}_\star(\eta)} \frac{\rho_s(\eta; \boldsymbol{\alpha}) \rho_p(y, 0; \boldsymbol{\alpha}) f(y, 0; \boldsymbol{\alpha})}{\rho_s^{\sigma(y;\boldsymbol{\alpha})}(\boldsymbol{\alpha}) V(\boldsymbol{\alpha}) |(v^{\mathrm{eff}})'(\eta; \boldsymbol{\alpha})|} T^{\mathrm{dr}}(\eta; \boldsymbol{\gamma}, \boldsymbol{\alpha}) h_j^{\mathrm{dr}}(y, 0; \boldsymbol{\alpha})$$
$$- I(y; \boldsymbol{\gamma}) \big(\rho_s(y, 0) f(y, 0) h_j^{\mathrm{dr}}(y, 0)\big)^{*\mathrm{dr}}(y, 0; \boldsymbol{\gamma}) \bigg). \tag{E.28}$$

We can then sum this contribution with (E.20) to get the full coefficient. Before going further, however, let us note the apparent lack of space parity symmetry in the expression (E.28), both in the integral $I(y; \boldsymbol{\gamma})$, and in the integral $\int_{-\infty}^{\xi} \mathrm{d}\eta$. We do not assume the *model* to be parity symmetric, however what we note here is that the *expression* treats left and right regions of space differently, independently of the properties of the model. This is due to the lack of a manifest parity symmetry in the solution by characteristics (2.23), where the integral is chosen to start at a left asymptotic stationary point. As mentioned in [33], this is a conventional choice, and a similar formula can be obtained by integrating towards a right asymptotic stationary point instead. It is not too difficult to obtain the result with this different choice:

$$\big(A_{ij}(\xi; y)\big)_2$$
$$= \sum_{\boldsymbol{\gamma} \in \boldsymbol{\theta}_\star(\xi)} \frac{\rho_s(\xi; \boldsymbol{\gamma})}{V(\boldsymbol{\gamma}) |(v^{\mathrm{eff}})'(\xi; \boldsymbol{\gamma})|} h_i^{\mathrm{dr}}(\xi; \boldsymbol{\gamma}) \times$$
$$\times \bigg( -I(\boldsymbol{\gamma}) \int_{\xi}^{\infty} \mathrm{d}\eta \sum_{\boldsymbol{\alpha} \in \boldsymbol{\theta}_\star(\eta)} \frac{\rho_s(\eta; \boldsymbol{\alpha}) \rho_p(y, 0; \boldsymbol{\alpha}) f(y, 0; \boldsymbol{\alpha})}{\rho_s^{\sigma(y;\boldsymbol{\alpha})}(\boldsymbol{\alpha}) V(\boldsymbol{\alpha}) |(v^{\mathrm{eff}})'(\eta; \boldsymbol{\alpha})|} T^{\mathrm{dr}}(\eta; \boldsymbol{\gamma}, \boldsymbol{\alpha}) h_j^{\mathrm{dr}}(y, 0; \boldsymbol{\alpha})$$
$$+ (I(\boldsymbol{\gamma}) - I(y; \boldsymbol{\gamma})) \big(\rho_s(y, 0) f(y, 0) h_j^{\mathrm{dr}}(y, 0)\big)^{*\mathrm{dr}}(y, 0; \boldsymbol{\gamma}) \bigg). \tag{E.29}$$

while (E.20) stays unchanged. We subtract (E.28) from (E.29), and we take the functional derivative with respect to $h_i^{\mathrm{dr}}(\xi, \boldsymbol{\gamma})$ in order to isolate the terms within the large parentheses. The result is

$$
\begin{aligned}
0 = I(\boldsymbol{\gamma}) \bigg( & \int_{-\infty}^{\infty} \mathrm{d}\eta \sum_{\boldsymbol{\alpha} \in \boldsymbol{\theta}_\star(\eta)} \frac{\rho_{\mathrm{s}}(\eta; \boldsymbol{\alpha}) \rho_{\mathrm{p}}(y, 0; \boldsymbol{\alpha}) f(y, 0; \boldsymbol{\alpha})}{\rho_{\mathrm{s}}^{\sigma(y; \boldsymbol{\alpha})}(\boldsymbol{\alpha}) V(\boldsymbol{\alpha}) |(v^{\mathrm{eff}})'(\eta; \boldsymbol{\alpha})|} T^{\mathrm{dr}}(\eta; \boldsymbol{\gamma}, \boldsymbol{\alpha}) h_j^{\mathrm{dr}}(y, 0; \boldsymbol{\alpha}) \\
& - \big(\rho_{\mathrm{s}}(y, 0) f(y, 0) h_j^{\mathrm{dr}}(y, 0)\big)^{*\mathrm{dr}}(y, 0; \boldsymbol{\gamma}) \bigg).
\end{aligned}
\tag{E.30}
$$

The functional derivative with respect to $h_j^{\mathrm{dr}}(y, 0; \boldsymbol{\alpha})$ then gives, after dividing by $\rho_{\mathrm{s}}(y, 0; \boldsymbol{\alpha}) f(y, 0; \boldsymbol{\alpha})$ (assuming without loss of generality the generic case $f(y, 0; \boldsymbol{\alpha}) \neq 0$),

$$
0 = I(\boldsymbol{\gamma}) \bigg( \frac{\rho_{\mathrm{s}}(\eta_\star(\boldsymbol{\alpha}); \boldsymbol{\alpha}) n_0(y; \boldsymbol{\alpha}) T^{\mathrm{dr}}(\eta_\star(\boldsymbol{\alpha}); \boldsymbol{\gamma}, \boldsymbol{\alpha})}{\rho_{\mathrm{s}}^{\sigma(y; \boldsymbol{\alpha})}(\boldsymbol{\alpha}) V(\boldsymbol{\alpha}) |(v^{\mathrm{eff}})'(\eta_\star(\boldsymbol{\alpha}); \boldsymbol{\alpha})|} - ((1 - T n_0(y))^{-1} T n_0(y))(\boldsymbol{\gamma}, \boldsymbol{\alpha}) \bigg).
\tag{E.31}
$$

If the asymptotics $n_0^\pm(\boldsymbol{\theta})$ are different, then the expression within the large parentheses cannot be zero: as a function of $y$, the first terms has a jump at $y = y_\star(\boldsymbol{\alpha})$, while the second term is continuous. Therefore we conclude that

$$
I(\boldsymbol{\gamma}) = 0.
\tag{E.32}
$$

Since $I(\boldsymbol{\gamma})$ is continuous as a function of the asymptotics $n_0^\pm(\boldsymbol{\theta})$, the case of equal asymptotics is obtained by taking the limit, thus also giving 0.

Using this important simplification, we find

$$
\big(A_{ij}(\xi; y)\big)_2
\tag{E.33}
$$
$$
= -\sum_{\boldsymbol{\gamma} \in \boldsymbol{\theta}_\star(\xi)} \frac{\rho_{\mathrm{s}}(\xi; \boldsymbol{\gamma})}{V(\boldsymbol{\gamma}) |(v^{\mathrm{eff}})'(\xi; \boldsymbol{\gamma})|} h_i^{\mathrm{dr}}(\xi; \boldsymbol{\gamma}) I(y; \boldsymbol{\gamma}) \big(\rho_{\mathrm{s}}(y, 0) f(y, 0) h_j^{\mathrm{dr}}(y, 0)\big)^{*\mathrm{dr}}(y, 0; \boldsymbol{\gamma}).
$$

We also note that $I(\boldsymbol{\theta}) = 0$ implies

$$
I(y; \boldsymbol{\theta}) = \frac{n_0^\pm(\boldsymbol{\theta}) - n_0(y; \boldsymbol{\theta})}{\rho_{\mathrm{s}}^\pm(\boldsymbol{\theta})} \qquad \text{for} \qquad y \gtrless y_\star(\boldsymbol{\theta})
\tag{E.34}
$$

as well as the relations (5.26) and (5.25). We put together (E.20) and (E.33) to obtain (5.19).

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
