# Peer review of "Exact large-scale correlations in integrable systems out of equilibrium"

_SciPost Physics, doi:SciPost Phys. 5, 054 (2018)_

## Round 2 · Referee Report · Anonymous (Referee 1) · 2018-5-16

Strengths

1- Important advances on a very interesting problem; 2-Timely;

Weaknesses

  • See the report

Report

The paper studies nonequilibrium dynamics of integrable systems in inhomogeneous settings focussing on the determination of dynamical correlation functions. Specifically, the author considers situations treatable using the recently introduced theory of generalised hydrodynamics and proposes new formulae for the dynamical connected two-point functions of generic local observables in the ``Eulerian scaling limit" of large distances and times. In this limit, the state of system can be thought of as a collection of stationary states one at each space-time point. States on a given time slice are uncorrelated, however, non trivial correlations can be observed considering observables at different times. The author combines a generalised fluctuation dissipation theorem with the a "non-linear method of characteristics" and some Thermodynamic Bethe ansatz identities to determine new formulae for the connected two-point functions of charge densities and currents in the scaling limit. The latter are used to determine two point functions of generic operators in the framework of "hydrodynamic projection theory".
Two point-functions of charge densities and currents have a relatively simple expression, and require the solution of a linear integral equation. Instead, two-point functions of generic observables also depend on a function written in terms of an infinite form-factor series. In some special cases, using some known results for one-point functions in homogeneous settings, this function can be written in a simpler way in terms of the solution of an integral equation. The author also outlines a general inductive strategy to determine $N$-point functions and carries it out explicitly in the non-interacting case. Finally, he applies his results to the case of partitioning protocol and finds that the two point function $\langle{{q}_i(\xi t,t)q_j(y,0)}\rangle$, where $t=0$ is the initial time and $y=0$ is the junction, depends on how precisely the two states are connected.

The examples given in the paper are centred on the case of integrable quantum field theories but the treatment is kept at a general level and it is applicable also to classical integrable field theories and, with some caveats (see below), to integrable quantum spin chains.

I think that the paper is very interesting, it provides novel and highly non-trivial results further expanding the generalised hydrodynamics theory, and the derivation is mathematically sound. Therefore, I recommend the publication of this paper in SciPost. Before publication, however, the author should improve some aspects. First of all assumption (i) at page 12 (namely $p'(\theta)>0$ and $(v^{eff})'(\theta)>0$) is not fulfilled in integrable quantum spin chains (in short-ranged quantum spin chains both the momentum and the effective velocities are non monotonic functions of the rapidity, this is the case for the XY and the XXZ models for example). This means that all the results obtained on the basis of this assumption are not immediately applicable to integrable quantum spin chains but need some modification. This point should be clearly stressed and the text modified accordingly. Moreover, even if the author moved most of the technical parts in the appendices, some passages are difficult to read (see the detailed points below).

Another point that the author might want to consider is to add some further numerical checks, complementing those of [16], at least in the case of non-interacting systems. Such checks would in my opinion improve the paper.

Requested changes

1- In the introduction I suggest to move the paragraph

"Here we use a continuous space notation x, and the trace notation Tr. This is for convenience, and the problem is posed in its most general setting, for classical (where the trace means a summation over classical configurations) or quantum models, on a one-dimensional infinite space that can be continuous or discrete."

after Eq. (1.1), as the notation described is introduced there.

2- When discussing the various spectral expansions for GGE two-point functions (or Gibbs two-point functions that can be extended to GGE) I suggest to add also "Essler and Konik, J. Stat. Mech. (2009) P09018" to Refs.[53-56];

3- Can the author explain why the averaging in Eq. (1.5) is not expected to be necessary for one point functions?

4- What do the author mean with "involving TBA strings if the fundamental scattering is nondiagonal" at page 7? TBA strings are interpreted as bound states, they are not related non-diagonal scattering.

5- I suggest to give more explanation on the physical meaning of vector and scalar fields in Sec 2.1.

6- For consistency, in the discussion after Eq. (2.14) I suggest to use $\boldsymbol \theta$ also for the argument of $n_t(x,\boldsymbol \theta)$.

7- I find the discussion after Eq. (2.2) confusing. I suggest of using something on the lines of "the space of pseudolocal charges" instead of "the space spanned by $h_i(\boldsymbol\theta)$" because this gives the wrong impression that the author considers $h_i(\boldsymbol\theta)$ observables in the Hilbert space.

8 - In the discussion above equation 3.2 I suggest adding "of charge densities and currents", as the previous discussion was generic.

9- It seems that there is some confusion in the referencing of equation 3.4. In several cases (e.g. above Eq. 3.5 and in Appendix B1) what is referenced as Eq. 3.4 is the unnumbered equation above.

10- At the top of page 17 I suggest to replace "these ingredients" with " Eq. (3.11)" as that is the only ingredient needed. Moreover, I suggest to include Eq. (3.11) among the main results of the paper stressed in the discussion after Eq. (3.18).

11- I find Sec. 3.3 difficult to read. First of all, when talking about the hydrodynamic projection theory, I suggest to give the main ideas of the theory. For example the author could move there the brief discussion which is now at the beginning of Sec. 3.4. Second, Eqs. (3.28) and (3.29) appear to me as a rewriting of (3.16) - (3.18) and not a derivation of them as stated at the beginning of the sub-section. In my understanding such re-derivation is carried out in the remark. In summary: I suggest the author to reorganise this sub-section to improve readability.

12- I think that Eq. (3.34) should be explained in more detail. In particular it could be helpful for the reader to stress that the particular form of the resolution of the identity used is due to the non-orthogonality of the basis. Moreover, the author could add another step between the second and the third line.

---

## Round 2 · Referee Report · Anonymous (Referee 2) · 2018-5-22

Strengths

1- New analytic findings for space-time averaged dynamical correlation functions in integrable systems out of equilibrium 2- The results sound general 3- There are explicit examples

Weaknesses

1- The notations are not always standard, and readability is compromised by the large number of definitions 2- The analytic results are not checked against numerics

Report

This paper addresses the problem of computing correlation functions in inhomogeneous states that time evolve under the Hamiltonian of an integrable system. The author identifies and compute some quantities that can be accessed within the framework of the so-called ``generalized hydrodynamics'', which is a theory recently developed to deal with inhomogeneities in integrable systems. In particular, the author exhibits analytic expressions for space-time averaged dynamical correlation functions (he calls it ``Eulerian scaling limit for correlation functions'').
The paper is very long and rather technical, but, undoubtedly, the author made an effort to present the results in a simple way. Considering also their generality, the results are extremely interesting, therefore I strongly recommend this paper for publication in Scipost after minor revision, detailed in ``Requested changes''.

Requested changes

1- In the middle of page 4, the author writes In integrable quantum spin chains, two-point functions in Gibbs state have been calculated [63,64], but it is unclear how to extend to GGEs''. I think that the situation is less obscure than it is presented. Indeed, also Ref. [83] is a generalization of [63,64] to GGEs; as far as I know, the first papers generalizing [63,64] to GGEs were [] B. Pozsgay, J. Stat. Mech. (2013) P07003; [] M. Fagotti and F.H.L. Essler, J. Stat. Mech. (2013) P07012. 2- I think that there is a typo in the definition of $T^T$ just below (2.4). 3- The author presents the theory in a very general way. There are however equations that could be less general than expected, and I wonder whether such a general presentation is really worth. For example, I'm not completely sure that the first equation in (2.6) holds true also in the gapless XXZ model, where the equation could be correct only up to the sign. Can the sign be simplified by redefining the various quantities? 4- I think that I understand the logic behind (2.19) and (2.20), however I'm wondering whether the limiting procedure could have subtleties. In (1.5) the operators are averaged over a a space-time region whose extent scales as $\lambda^\nu$. If, in (2.19), one considers the case $x_n-x_m\sim O(\lambda^\beta)$, the validity of (2.19) could depend on how big $\beta$ is with respect to $\nu$, couldn't it? 5- The variable $y$ in the definition (2.25) is not defined. 6- Two lines below (2.26),... are monotonically increasing functions of the velocity.'' Is ''velocity'' a typo? If not, which velocity? 7- Could the author explain the comment above (3.1), that is to say, In quantum models, terms coming from nontrivial commutators between local conserved densities are negligible in the Euler scale, contributing only to higher-order derivatives''? 8- I'm not sure to understand the physical meaning ofobservables perturb the state'' at the end of section 5.2.1.

---

## Round 3 · Referee Report · Anonymous (Referee 1) · 2018-9-23

Strengths

-

Weaknesses

-

Report

The author addressed almost all of the points I raised in my previous report and substantially improved the readability of the paper. I am, however, not satisfied by his response to my first point of the previous report for the part concerning the derivative of the effective velocity with respect to the rapidity $\theta$. Specifically, the author writes

"As for $v^{{\rm eff},\prime}(\theta)>0$ this is the case in most states that I have seen (including thermal, and non-equilibrium steady states, in XY and in XXZ and other models; indeed this is something that is important in the solution of the equations defining the nonequilibrium steady state in [7,8]), but it might be broken in certain special states. Indeed, in such cases, some formulae will have to be modified. For simplicity I keep this assumption here. Comments added to clarify this."

I disagree. Take for example the XX spin-1/2 model: in this case $v^{{\rm eff}}( \theta)\propto\sin \theta$, which is clearly a non-monotonic function of the rapidity (here I am using as rapidity the standard quasi-momentum $\theta=[-\pi,\pi]$). This conclusion holds for every stationary state (the velocity does not depend on the state as the model is free). The same happens also for interacting spin-chains with local interactions, for example the XXZ spin-1/2 chain, for all the stationary states I have ever encountered (including thermal and non-equilibrium steady states). Finally, as far as I can see, the condition $v^{{\rm eff},\prime}(\theta)>0$ is not the relevant one for the solution of the equations defining the nonequilibrium steady state in [7,8]. For example, in the partitioning protocol what is needed is that $\xi-v^{{\rm eff}}(\xi, \theta)=0$ has unique solution in $\xi$ for every fixed $\theta$.

This point should be fixed before publication.

Requested changes

Additional Typos: 1- Page 6 first paragraph: Pozsgai -> Pozsgay 2- Page 6 second paragraph: characteresitics -> characteristics 2- Page 6: Section 4 still not mentioned in the description of the paper's organization (see previous report) 3- Page 14 first paragraph: obtain -> obtains/can obtain 4- Page 16 last paragraph: time-evolution operator -> time evolved occupation function 6- Some of the $i$'s are missing from the equations of Appendix C 7- In Appendix C I suggest to mention $\cosh^{\rm dr} \theta= 2\pi \rho_s(\theta)$

---

## Round 3 · Referee Report · Anonymous (Referee 2) · 2018-10-9

Strengths

-

Weaknesses

-

Report

The new version of the paper addresses almost all my criticisms. I still think, however, that there is an issue related to the generality of the assumptions. For example, the assumption of the effective velocity being a monotonically increasing function of the rapidity is not even verified in noninteracting chains, where $2\pi$-periodicity and locality of the Hamiltonian imply that the velocity is a continuous function of the momentum with at least a maximum and a minimum.

Requested changes

1- I think that the author should lessen some claims of generality.

---

## Round 3 · Author Response

First, I apologise for the long delay in replying. This is because of a misunderstanding: for some reason, I have not received the email from SciPost telling me that an editorial decision had been made and that I should now provide answers to referee and make minor modifications. I have seen this by luck, as I went to the SciPost page of the article because I was wondering why it was taking so long.

I thank both referees for their careful reading of the manuscript. I have made most of the changes asked for: all small adjustments, and additional small explanations and additional refs, re-organisation of section 3.3, and clarification of the notation in section 2.1, and of the assumptions in section 2.3. In addition: (1) I have changed the sign of the “free energy function” that I was using (now calling it $\mathsf{F}_a$), in order to make it really a free energy type of function and correcting a sign mistake in what I said was the free energy; (2) I have replaced “theorem” by “principle”: Euler-scale fluctuation-dissipation principle, (3) on page 5 (just above eq. 1.5) I have made the condition on the fluid cell more stringent, as I believe the exponent 1/2 is necessary in order to avoid diffusion effect, and higher exponents when super-diffusion occurs, and (4) I have updated the references.

I have not added additional numerics, because I think the paper is already rather long as it is, and because the calculations in ref [16] provide sufficient evidence for the most nontrivial formulae, especially in interacting models (in fact, there are also free-field calculations in [16] that further confirm some formulae). I’d like to keep a more in-depth numerical and analycitcal study of the free field cases to another paper.

In the "list of changes" are the referees' reports, with my answers marked by ">>>" and the corresponding changes made.

---

## Round 3 · List of Changes

Warnings issued while processing user-supplied markup:

  • Inconsistency: plain/Markdown and reStructuredText syntaxes are mixed. Markdown will be used.
    Add "#coerce:reST" or "#coerce:plain" as the first line of your text to force reStructuredText or no markup.
    You may also contact the helpdesk if the formatting is incorrect and you are unable to edit your text.

=======

Report

This paper addresses the problem of computing correlation functions in inhomogeneous states that time evolve under the Hamiltonian of an integrable system. The author identifies and compute some quantities that can be accessed within the framework of the so-called generalized hydrodynamics'', which is a theory recently developed to deal with inhomogeneities in integrable systems. In particular, the author exhibits analytic expressions for space-time averaged dynamical correlation functions (he calls it Eulerian scaling limit for correlation functions''). The paper is very long and rather technical, but, undoubtedly, the author made an effort to present the results in a simple way. Considering also their generality, the results are extremely interesting, therefore I strongly recommend this paper for publication in Scipost after minor revision, detailed in Requested changes''.

Requested changes

1- In the middle of page 4, the author writes In integrable quantum spin chains, two-point functions in Gibbs state have been calculated [63,64], but it is unclear how to extend to GGEs''. I think that the situation is less obscure than it is presented. Indeed, also Ref. [83] is a generalization of [63,64] to GGEs; as far as I know, the first papers generalizing [63,64] to GGEs were [] B. Pozsgay, J. Stat. Mech. (2013) P07003; [] M. Fagotti and F.H.L. Essler, J. Stat. Mech. (2013) P07012.

Thank you - the papers pointed out indeed generalise some of these techniques to GGEs, although some of the “two-point functions” really are one-point functions (i.e. short range) - the asymptotic at large space-time are not studied.

2- I think that there is a typo in the definition of TT just below (2.4).

Corrected - thank you

3- The author presents the theory in a very general way. There are however equations that could be less general than expected, and I wonder whether such a general presentation is really worth. For example, I'm not completely sure that the first equation in (2.6) holds true also in the gapless XXZ model, where the equation could be correct only up to the sign. Can the sign be simplified by redefining the various quantities?

I think all equations have quite wide applicability, including in the XXZ model. In eq 2.6, the first equation is correct and I believe general. The choice of the spectral parameter $\theta$ is arbitrary, and its relation to TBA quantities is effected by the function $p(\theta)$. If one gives the density of state as function of some parametrisation $\theta$, then eq 2.6 can be seen as the definition of $p(\theta)$. If in some papers a sign difference arises in eq 2.6, then this is just a redefinition of $p(\theta)$; here I choose the sign indicated. Footnote 2 on p 8 adds a bit of explanation.

4- I think that I understand the logic behind (2.19) and (2.20), however I'm wondering whether the limiting procedure could have subtleties. In (1.5) the operators are averaged over a a space-time region whose extent scales as λν. If, in (2.19), one considers the case xn−xm∼O(λβ), the validity of (2.19) could depend on how big β is with respect to ν, couldn't it?

Yes this is correct - if one makes the $x_n$’s dependent on $\lambda$ when taking the large $\lambda$ limit, then the right hand side of 2.19 can be modified. But this would not be the Euler scaling limit anymore. In 2.19 we use the Euler scaling limit, and in this limit the $x_k$’s and $t_k$’s are kept fixed (eq 1.5). Then the r.h.s. of 2.19 is expected. I have added “for fixed $x_k$’s and $t_k$’s” below eq 1.5.

5- The variable y in the definition (2.25) is not defined.

It is the starting point of the trajectory, comment added

6- Two lines below (2.26), ... are monotonically increasing functions of the velocity.'' Is ''velocity'' a typo? If not, which velocity?

Yes typo - I meant the spectral rapidity $\theta$ (the spectral parameter is seen here as a doublet of a rapidity and a particle type, see paragraph above eq 2.1).

7- Could the author explain the comment above (3.1), that is to say, In quantum models, terms coming from nontrivial commutators between local conserved densities are negligible in the Euler scale, contributing only to higher-order derivatives''?

These commutators only give rise to derivatives of local operators, see eqs. 91-93 of [12], which can be neglected in Eulerian correlation functions (sentence adjusted, footnote added). But such terms are expected to contribute to the higher-derivative corrections to Euler hydrodynamics, perhaps diffusive terms (second order in derivatives) or at higher orders.

8- I'm not sure to understand the physical meaning of observables perturb the state'' at the end of section 5.2.1.

Sentence clarified: “ the insertion of an observable in a correlation function perturbs the state as seen by other observables, “

===

Report

The paper studies nonequilibrium dynamics of integrable systems in inhomogeneous settings focussing on the determination of dynamical correlation functions. Specifically, the author considers situations treatable using the recently introduced theory of generalised hydrodynamics and proposes new formulae for the dynamical connected two-point functions of generic local observables in the Eulerian scaling limit" of large distances and times. In this limit, the state of system can be thought of as a collection of stationary states one at each space-time point. States on a given time slice are uncorrelated, however, non trivial correlations can be observed considering observables at different times. The author combines a generalised fluctuation dissipation theorem with the a "non-linear method of characteristics" and some Thermodynamic Bethe ansatz identities to determine new formulae for the connected two-point functions of charge densities and currents in the scaling limit. The latter are used to determine two point functions of generic operators in the framework of "hydrodynamic projection theory". Two point-functions of charge densities and currents have a relatively simple expression, and require the solution of a linear integral equation. Instead, two-point functions of generic observables also depend on a function written in terms of an infinite form-factor series. In some special cases, using some known results for one-point functions in homogeneous settings, this function can be written in a simpler way in terms of the solution of an integral equation. The author also outlines a general inductive strategy to determine N-point functions and carries it out explicitly in the non-interacting case. Finally, he applies his results to the case of partitioning protocol and finds that the two point function ⟨qi(ξt,t)qj(y,0)⟩, where t=0 is the initial time and y=0 is the junction, depends on how precisely the two states are connected.

The examples given in the paper are centred on the case of integrable quantum field theories but the treatment is kept at a general level and it is applicable also to classical integrable field theories and, with some caveats (see below), to integrable quantum spin chains.

I think that the paper is very interesting, it provides novel and highly non-trivial results further expanding the generalised hydrodynamics theory, and the derivation is mathematically sound. Therefore, I recommend the publication of this paper in SciPost. Before publication, however, the author should improve some aspects. First of all assumption (i) at page 12 (namely p′(θ)>0 and (veff)′(θ)>0 ) is not fulfilled in integrable quantum spin chains (in short-ranged quantum spin chains both the momentum and the effective velocities are non monotonic functions of the rapidity, this is the case for the XY and the XXZ models for example). This means that all the results obtained on the basis of this assumption are not immediately applicable to integrable quantum spin chains but need some modification. This point should be clearly stressed and the text modified accordingly. Moreover, even if the author moved most of the technical parts in the appendices, some passages are difficult to read (see the detailed points below). 

Another point that the author might want to consider is to add some further numerical checks, complementing those of [16], at least in the case of non-interacting systems. Such checks would in my opinion improve the paper.

First point made: Thank you for pointing out these subtleties. The condition $p’(\theta)>0$ is a required property of the parametrisation chosen, not a property of any model under consideration. For instance, one could choose $\theta$ to be the momentum itself, in which case $p’(\theta)=1$. This is true in any model. In particular, the quantity $\theta$ is not necessarily the “natural” (in whatever way) Bethe ansatz rapidity. If for a choice of $\theta$ we have that $p(\theta)$ is not monotonic, then this means there are two values of theta giving the same momentum, and within the formalism that I am introducing here, these would be two different particle types. The condition $p’(\theta)>-0$ is only an expression of the faithfulness of the parametrisation. This point has been clarified at the beginning of section 2.1, the end of section 2 has been changed accordingly. As for ${v^{\rm eff}}’(\theta)>0$ this is the case in most states that I have seen (including thermal, and non-equilibrium steady states, in XY and in XXZ and other models; indeed this is something that is important in the solution of the equations defining the nonequilibrium steady state in [7,8]), but it might be broken in certain special states. Indeed, in such cases, some formulae will have to be modified. For simplicity I keep this assumption here. Comments added to clarify this.

Second point made: I have addressed the comments below, which I hope will make the discussion clearer.

Third point made: Yes it would be nice to have additional numerics, but I feel this is starting to be beyond the scope of this paper, which is already quite lengthy. I also think that the numerics provided in [16] is rather strong, and more interesting than any numerics one may do in free models, as it involves truly interacting models. Nevertheless, I do hope to provide more numerical evidence in the future.

Requested changes

1- In the introduction I suggest to move the paragraph 

"Here we use a continuous space notation x, and the trace notation Tr. This is for convenience, and the problem is posed in its most general setting, for classical (where the trace means a summation over classical configurations) or quantum models, on a one-dimensional infinite space that can be continuous or discrete."

after Eq. (1.1), as the notation described is introduced there.

done 

2- When discussing the various spectral expansions for GGE two-point functions (or Gibbs two-point functions that can be extended to GGE) I suggest to add also "Essler and Konik, J. Stat. Mech. (2009) P09018" to Refs.[53-56];
 added - indeed this was an important reference missing. 
3- Can the author explain why the averaging in Eq. (1.5) is not expected to be necessary for one point functions? 
 I don’t have a good, principle-based explanation for this - but from observations of numerical and exact calculations found in the literature, this seems to be the case. I’ve changed the sentence accordingly. 
4- What do the author mean with "involving TBA strings if the fundamental scattering is nondiagonal" at page 7? TBA strings are interpreted as bound states, they are not related non-diagonal scattering. 
 Indeed - probably the particles emerging after diagonalising the scattering are called “magnons” in the literature, in any case I’ve just removed the sentence in the parenthesis, which was not that useful. 
5- I suggest to give more explanation on the physical meaning of vector and scalar fields in Sec 2.1.

The first few paragraphs of section 2.1 have been modified to clarify the situation.

6- For consistency, in the discussion after Eq. (2.14) I suggest to use θ also for the argument of nt(x,θ).

In square bracket [n_t(x)] this means that the quantity depends on the whole function $\theta\mapsto n_t(x;\theta)$, not on $n_t(x;\theta)$ for any particular $\theta$. But it’s true that this was a bit unclear and not a usual notation. I’ve changed this by introducing the notation $n_{x,t}$ for the state at $x,t$, which I think makes things more clear. See explanations in this paragraph and the relation at the beginning of section 2.2 for the two notations used for the state function.

7- I find the discussion after Eq. (2.2) confusing. I suggest of using something on the lines of "the space of pseudolocal charges" instead of "the space spanned by hi(θ)" because this gives the wrong impression that the author considers hi(θ) observables in the Hilbert space.

I see the confusion; the use of “the Hilbert space” can be a bit misleading, as there are many Hilbert spaces, not just those of the states of the original quantum problem (and in any case, the paper applies as well to classical problems). I’ve adjusted the sentence. 
8 - In the discussion above equation 3.2 I suggest adding "of charge densities and currents", as the previous discussion was generic.
 done 
9- It seems that there is some confusion in the referencing of equation 3.4. In several cases (e.g. above Eq. 3.5 and in Appendix B1) what is referenced as Eq. 3.4 is the unnumbered equation above.
 Thank you - corrected 
10- At the top of page 17 I suggest to replace "these ingredients" with " Eq. (3.11)" as that is the only ingredient needed. Moreover, I suggest to include Eq. (3.11) among the main results of the paper stressed in the discussion after Eq. (3.18).
 done 
11- I find Sec. 3.3 difficult to read. First of all, when talking about the hydrodynamic projection theory, I suggest to give the main ideas of the theory. For example the author could move there the brief discussion which is now at the beginning of Sec. 3.4. Second, Eqs. (3.28) and (3.29) appear to me as a rewriting of (3.16) - (3.18) and not a derivation of them as stated at the beginning of the sub-section. In my understanding such re-derivation is carried out in the remark. In summary: I suggest the author to reorganise this sub-section to improve readability.
 The section has been clarified. Indeed the main goal was simply re-writing of results obtained to make the connection with hydrodynamic projection clear - this has now been clarified at the beginning of the section. And indeed the remark was to be seen as a re-derivation. I’d like to keep the structure as is; however I’ve clarified the logic and made two sub-subsections instead of a main text and a remark. 
12- I think that Eq. (3.34) should be explained in more detail. In particular it could be helpful for the reader to stress that the particular form of the resolution of the identity used is due to the non-orthogonality of the basis. Moreover, the author could add another step between the second and the third line.

Explanations added, and the additional calculation line also added.

---

## Round 4 · Author Response

I thank the referees for their careful consideration of the manuscript, and especially for both pointing out that the assumption about monotonicity of the effective velocity in rapidity is too strong; I was clearly too fast in making the assertion. [I think in the free-chain examples that both referees gave, the assumption *is* satisfied, with an appropriate choice of spectral space: one may divide the momenta into two regions, in such a way that within each region the velocity is monotonic, and one can see each regions as corresponding to a different particle type. However I don't think such a construction can be done generically in interacting systems.]

Indeed as pointed out the assumption is not necessary for the solution to the partitioning protocol. In fact, I realised that it was not necessary for any result I have presented - it was just simplifying my life in characterising the solutions to certain equations, but is in fact not strictly needed. Thus I have modified the discussion of this assumption on page 13, making it a remark only, and I have make appropriate modifications throughout in order to account for this: all places where the derivative of the effective velocity appeared through Jacobian I have added absolute values; in sections 5.3 and E.2 I have taken away the requirement of the monotonicity assumption, and I have adjusted the sentence between eq 3.24 and 3.25 on p 19.

However, perhaps the most interesting realisation from thinking about this is that in general, the rapidity derivative of the effective velocity may vanish. In this case, some large-time asymptotics, at certain rays for instance in the partitioning protocol (e.g. near the maximal velocity), may be modified. I think this is a potentially very interesting effect, which I keep for future works. I have added a paragraph about this in the conclusion, and also a short comment in the Remark on page 13.

I have also corrected all typos found by referee 2.

---

## Round 4 · List of Changes

Absolute value for derivative of effective velocity in eqs. 3.36, 4.19, 4.23, 5.12, 5.17, 5.19, 5.21, E.15, E.17, E.20, E.22, E.23, E.24, E.28 and eq above - E.31, E.33

paragraph added in conclusion

discussion adjusted in section 5.3 (p35) and E.2 (p47)

discussion adjusted and remark added p13

adjusted the sentence between eq 3.24 and 3.25 on p 19

---

## Editorial Decision

published